# On the infinite-depth limit of finite-width neural networks

**Soufiane Hayou**  *hayou@nus.edu.sg*
*Department of Mathematics*
*National University of Singapore*

**Reviewed on OpenReview:** *https: // openreview. net/ forum? id=RbLsYz1Az9*

## Abstract

In this paper, we study the infinite-depth limit of finite-width residual neural networks with random Gaussian weights. With proper scaling, we show that by fixing the width and taking the depth to infinity, the pre-activations converge in distribution to a zero-drift diffusion process. Unlike the infinite-width limit where the pre-activation converge weakly to a Gaussian random variable, we show that the infinite-depth limit yields different distributions depending on the choice of the activation function. We document two cases where these distributions have closed-form (different) expressions. We further show an intriguing change of regime phenomenon of the post-activation norms when the width increases from 3 to 4. Lastly, we study the sequential limit infinite-depth-then-infinite-width and compare it with the more commonly studied infinite-width-then-infinite-depth limit.

## 1 Introduction

The empirical success of over-parameterized neural networks has sparked a growing interest in the theoretical understanding of these models. The large number of parameters – millions if not billions – and the complex (non-linear) nature of the neural computations (presence of non-linearities) make this hypothesis space highly non-trivial. However, in certain situations, increasing the number of parameters has the effect of 'placing' the network in some 'average' regime that simplifies the theoretical analysis. This is the case with the infinite-width asymptotics of random neural networks. The infinite-width limit of neural network architectures has been extensively studied in the literature (e.g. Neal (1995); Schoenholz et al. (2017); Yang (2020); Poole et al. (2016); Arora et al. (2019); Hayou et al. (2019a)), and has led to many interesting theoretical and algorithmic innovations (see Appendix A for a comprehensive discussion). However, most works on this limit consider a fixed depth network. *What about infinite-depth?* Existing works on the infinite-depth limit can generally be divided into three categories:

- *Infinite-width-then-infinite-depth limit*: in this case, the width is taken to infinity first, then the depth is take to infinity. This is the infinite-depth limit of infinite-width neural networks. This limit was particularly used to derive the Edge of Chaos initialization scheme (Schoenholz et al., 2017; Poole et al., 2016), study the impact of the activation function (Hayou et al., 2019a), the behaviour of the NTK (Hayou et al., 2020; Xiao et al., 2020) etc.

- *The joint infinite-width-and-depth limit*: in this case, the depth-to-width ratio is fixed, and therefore, the width and depth are jointly taken to infinity at the same time. There are few works that study the joint width-depth limit. For instance, in (Li et al., 2021), the authors showed that for a special form of residual neural networks (ResNet), the network output exhibits a (scaled) log-normal behaviour in this joint limit. This is different from the sequential limit where width is taken to infinity first, followed by the depth, in which case the distribution of the network output is asymptotically normal (Schoenholz et al., 2017; Hayou et al., 2019a). In (Li et al., 2022), the authors studied the covariance kernel of an MLP in the joint limit, and showed that it converges weakly to the solution of Stochastic Differential Equation (SDE). In Hanin & Nica (2020), the authors showed that in the joint limit case, the NTK of an MLP

remains random when the width and depth jointly go to infinity. This is different from the deterministic limit of the NTK where the width is taken to infinity before depth (Hayou et al., 2020). More recently, Hanin (2022) explored the impact of the depth-to-width ratio on the correlation kernel and the gradient norms in the case of an MLP architecture, and showed that this ratio can be interpreted as an effective network depth.

- *Infinite-depth limit of finite-width neural networks*: in both previous limits (infinite-width-then-infinite-depth limit, and the joint infinite-width-depth limit), the width goes to infinity. Naturally, one might ask what happens if width is fixed and depth goes to infinity? What is the limiting distribution of the network output at initialization? In (Hanin, 2019), the author showed that neural networks with bounded width are still universal approximators, which motivates the study of finite-width large depth neural networks. In (Peluchetti & Favaro, 2020), the authors showed that the pre-activations of a particular ResNet architecture converge weakly to a diffusion process in the infinite-depth limit. This is the result of the fact that ResNet can be seen as discretizations of SDEs (see Section 2).

In the present paper, we study the infinite-depth limit of finite-width ResNet with random Gaussian weights (an architecture that is different from the one studied in (Peluchetti & Favaro, 2020)). We are particularly interested in the *asymptotic behaviour of the pre/post-activation values*. Our contributions are four-fold:

1. Unlike the infinite-width limit, we show that the distribution of the pre-activations in the infinite-depth limit is not necessarily Gaussian. In the simple case of networks of width 1, we study two cases where we obtain known but completely different distributions by carefully choosing the activation function.

2. For ReLU activation function, we introduce and discuss the phenomenon of *network collapse*. This phenomenon occurs when the pre-activations in some hidden layer have all non-positive values which results in zero post-activations. This leads to a stagnant network where increasing the depth beyond a certain level has no effect on the network output. For any fixed width, we show that in the infinite-depth limit, network collapse is a zero-probability event, meaning that almost surely, all post-activations in the network are non-zero.

3. For general width networks, the distribution of the pre-activations is generally intractable. We focus on the norm of the post-activations with ReLU activation function, and show that this norm has approximately a Geometric Bronwian Motion (GBM) dynamics. We call this Quasi-GBM. We also shed light on a change of regime phenomenon that occurs when the width $n$ increases from 3 to 4. For width $n \leq 3$, resp. $n \geq 4$, the logarithmic growth factor of the post-activations is , resp. positive.

4. We study the sequential limit infinite-depth-then-infinite-width, which is the converse of the more commonly studied infinite-width-then-infinite-depth limit, and show some key similarities between these two limits. We particularly show that the pre-activations converge to the solution of a Mckean-Vlasov process, which has marginal Gaussian distributions, and thus we recover the Gaussian behaviour in this limit.

All the proofs are provided in the appendix and referenced in the main text. Empirical evaluations of these theoretical findings are also provided.

## 2   The infinite-depth limit

Hereafter, we denote the width, resp. depth, of the network by $n$, resp. $L$. We also denote the input dimension by $d$. Let $d, n, L \geq 1$, and consider the following ResNet architecture of width $n$ and depth $L$

$$
\begin{aligned}
Y_0 &= W_{in}x, \quad x \in \mathbb{R}^d \\
Y_l &= Y_{l-1} + \frac{1}{\sqrt{L}}W_l\phi(Y_{l-1}), \quad l = 1, \ldots, L,
\end{aligned}
\tag{1}
$$

where $\phi : \mathbb{R} \to \mathbb{R}$ is the activation function, $L \geq 1$ is the network depth, $W_{in} \in \mathbb{R}^{n \times d}$, and $W_l \in \mathbb{R}^{n \times n}$ is the weight matrix in the $l^{th}$ layer. We assume that the weights are randomly initialized with *iid* Gaussian

variables $W_l^{ij} \sim \mathcal{N}(0, \frac{1}{n})$, $W_{in}^{ij} \sim \mathcal{N}(0, \frac{1}{d})$. For the sake of simplification, we only consider networks with no bias, and we omit the dependence of $Y_l$ on $n$ in the notation. While the activation function is only defined for real numbers, we will abuse the notation and write $\phi(z) = (\phi(z^1), \dots, \phi(z^k))$ for any $k$-dimensional vector $z = (z^1, \dots, z^k) \in \mathbb{R}^k$ for any $k \geq 1$. We refer to the vectors $\{Y_l, l = 0, \dots, L\}$ by the *pre-activations* and the vectors $\{\phi(Y_l), l = 0, \dots, L\}$ by the *post-activations*. Hereafter, $x \in \mathbb{R}^d$ is fixed, and we assume that $x \neq 0$.

The $1/\sqrt{L}$ scaling in Eq. (1) is not arbitrary. This specific scaling was shown to stabilize the norm of $Y_l$ as well as gradient norms in the large depth limit (e.g. Hayou et al. (2021); Marion et al. (2022)). In the next result (which has been shown for the single input case in Peluchetti & Favaro (2020)), we show that the infinite depth limit of Eq. (1) (in the weak sense) exists and has the same distribution of the solution of a stochastic differential equation. For the sake of simplicity, we only state the result for the single input case; the multiple inputs case is provided in in Appendix B.

**Proposition 1.** *Assume that the activation function $\phi$ is Lipschitz on $\mathbb{R}^n$. Then, in the limit $L \to \infty$, the process $X_t^L = Y_{\lfloor tL \rfloor}$, $t \in [0, 1]$, converges in distribution to the solution of the following SDE*

$$dX_t = \frac{1}{\sqrt{n}} \|\phi(X_t)\| dB_t, \quad X_0 = W_{in} x, \tag{2}$$

*where $(B_t)_{t \geq 0}$ is a Brownian motion (Wiener process), independent from $W_{in}$. Moreover, if the activation function $\phi$ is only locally Lipschitz, then $X_t^L$ converges locally to $X_t$. More precisely, for any fixed $r > 0$, we consider the stopping times $\tau^L = \inf\{t \geq 0 : |X_t^L| \geq r\}$, and $\tau = \inf\{t \geq 0 : |X_t| \geq r\}$, then the stopped process $X_{t \wedge \tau^L}^L$ converges in distribution to the stopped solution $X_{t \wedge \tau}$ of the above SDE.*

The proof of Proposition 1 is provided in Appendix B.6. We use classical results on the numerical approximations of SDEs. Proposition 1 shows that the infinite-depth limit of finite-width ResNet (Eq. (1)) has a similar behaviour to the solution of the SDE given in Eq. (8). In this limit, $Y_{\lfloor tL \rfloor}$ converges in distribution to $X_t$. Hence, properties of the solutions of Eq. (8) should theoretically be 'shared' by the pre-activations $Y_{\lfloor tL \rfloor}$ when the depth is large. For the rest of the paper, we study some properties of the solutions of Eq. (8). This requires the definition of filtered probability spaces which we omit here. All the technical details are provided in Appendix B. We compare the theoretical findings with empirical results obtained by simulating the pre/post-activations of the original network Eq. (1). We refer to $X_t$, the solution of Eq. (8), by the *infinite-depth network*.

The distribution of $X_1$ (the last layer in the infinite-depth limit) is generally intractable, unlike in the infinite-width-then-infinite-depth limit (Gaussian, Hayou et al. (2021)) or joint infinite-depth-and-width limit (involves a log-normal distribution in the case of an MLP architecture, Li et al. (2021)). Intuitively, one should not expect a universal behaviour (e.g. the Gaussian behaviour in the infinite-width case) of the solution of Eq. (8) as this latter is highly sensitive to the choice of the activation function, and different activation functions might yield completely different distributions of $X_1$. We demonstrate this in the next section by showing that we can recover closed-form distributions by carefully choosing the activation function. The main ingredient is the use of Itô 's lemma. See Appendix B for more details.

## 3 Different behaviours depending on the activation function

In this section, we restrict our analysis to a width-1 ResNet with one-dimensional inputs, where each layer consists of a single neuron, i.e. $d = n = 1$. In this case, the process $(X_t)_{0 \leq t \leq 1}$ is one-dimensional and is solution of the following SDE

$$dX_t = |\phi(X_t)| dB_t, \quad X_0 = W_{in} x.$$

We can get rid of the absolute value in the equation above since the process $X_t$ has the same distribution as $\tilde{X}_t$, the solution of the SDE $d\tilde{X}_t = \phi(\tilde{X}_t) dB_t$. The intuition behind this is that the infinitesimal random variable '$dB_t$' is Gaussian distributed with zero mean and variance $dt$. Hence, it is a symmetric random variable and can absorb the sign of $\phi(X_t)$. The rigorous justification of this fact is provided in Theorem 7 in the Appendix. Hereafter in this section, we consider the process $X$, solution of the SDE

$$dX_t = \phi(X_t) dB_t, \quad X_0 = W_{in} x. \tag{3}$$

Given a function $g \in \mathcal{C}^2(\mathbb{R})$[1], we use Itô 's lemma (Lemma 4 in the appendix) to derive the dynamics of the process $g(X_t)$. We obtain,

$$dg(X_t) = \underbrace{\phi(X_t)g'(X_t)}_{\sigma(X_t)}\,dB_t + \underbrace{\frac{1}{2}\phi(X_t)^2 g''(X_t)}_{\mu(X_t)}\,dt. \tag{4}$$

In financial mathematics nomenclature, the function $\mu$ is called the *drift* and $\sigma$ is called the *volatility* of the diffusion process. Itô 's lemma is a valuable tool in stochastic calculus and is often used to transform and simplify SDEs to better understand their properties. It can also be used to find candidate functions $g$ and activation functions $\phi$ such that the SDE Eq. (4) admits solutions with known distributions, which yields a closed-form distribution for $X_t$. We consecrate the rest of this section to this purpose.

## 3.1 ReLU activation

ReLU is a piece-wise linear activation function. Let us first deal with the simpler case of linear activation functions. In the next result, we show that linear activation functions yield log-normal distributions. In this case, the process $X_t$ follows the Geometric Brownian motion dynamics. Later in this section, we show that this result can be adapted to the case of the ReLU activation function given by $\phi(x) = \max(x, 0)$.

**Proposition 2.** *Let $x \in \mathbb{R}$ such that $x \neq 0$. Consider a linear activation function $\phi(y) = \alpha y + \beta$, where $\alpha > 0, \beta \in \mathbb{R}$ are constants. Let $\sigma > 0$ and define the function $g$ by $g(y) = (\alpha y + \beta)^\gamma$, where $\gamma = \sigma \alpha^{-1}$. Consider the stochastic process $X_t$, solution of Eq. (3). Then, the process $g(X_t)$ is a solution of the SDE*

$$dg(X_t) = ag(X_t)dt + \sigma g(X_t)dB_t,$$

*where $a = \frac{1}{2}\sigma^2 \gamma^{-1}(\gamma - 1)$. As a result, we have that for all $t \in [0, 1]$,*

$$g(X_t) \sim g(X_0)\exp\left(\left(a - \frac{1}{2}\sigma^2\right)t + \sigma B_t\right).$$

The proof of Proposition 2 is provided in Appendix E, and consists of using Itô lemma and solving a differential equation. When the activation function is ReLU, we still obtain a log-normal distribution conditionally on the event that the initial value $X_0$ is positive.

**Proposition 3.** *Let $x \in \mathbb{R}$ such that $x \neq 0$, and let $\phi$ be the ReLU activation function given by $\phi(z) = \max(z, 0)$ for all $z \in \mathbb{R}$. Consider the stochastic process $X_t$, the solution of Eq. (3). Then, the process $X$ is a mixture of a Geometric Brownian motion and a constant process. More precisely, we have for all $t \in [0, 1]$*

$$X_t \sim \mathbb{1}_{\{X_0 > 0\}} X_0 \exp\left(-\frac{1}{2}t + B_t\right) + \mathbb{1}_{\{X_0 \leq 0\}} X_0.$$

*Hence, given a fixed $X_0 > 0$, the process $X$ is a Geometric Brownian motion.*

The proof of Proposition 3 is provided in Appendix F. We show that conditionally on $X_0 > 0$, with probability 1, the process $X_t$ is positive for all $t \in [0, 1]$[2]. When $X_t > 0$, the ReLU activation is just the identity function, which justifies the similarity between this result and the one obtained with linear activations (Proposition 2). Conversely, if $X_0 < 0$, the process is constant equal to $X_0$ since the updates '$dX_t$' are equal to zero in this case. A rigorous justification of this is given for general width $n$ later in the paper (Lemma 1). An empirical verification of Proposition 2 is provided in Fig. 1 where we compare the theoretical results to simulations of the *neural paths* $(Y_l)_{1 \leq l \leq L}$ and $(\log(Y_l))_{1 \leq l \leq L}$ from the original (finite-depth) ResNet given by Eq. (1). We observe an excellent match with theoretical predictions for depths $L = 50$ and $L = 100$. In the case of a small depth ($L = 5$), the theoretical distribution does not fit well the empirical one (obtained by simulations),

---

[1] Here $\mathcal{C}^2(\mathbb{R})$ refers to the vector space of functions $g : \mathbb{R} \to \mathbb{R}$ that are twice differentiable and their second derivatives are continuous.

[2] In Appendix F, we show that the stopping $\tau = \inf\{t \geq 0 : \text{ s.t. } X_t \leq 0\}$ is infinite almost surely, which is stronger that what we need. This is a classic result in stochastic calculus.

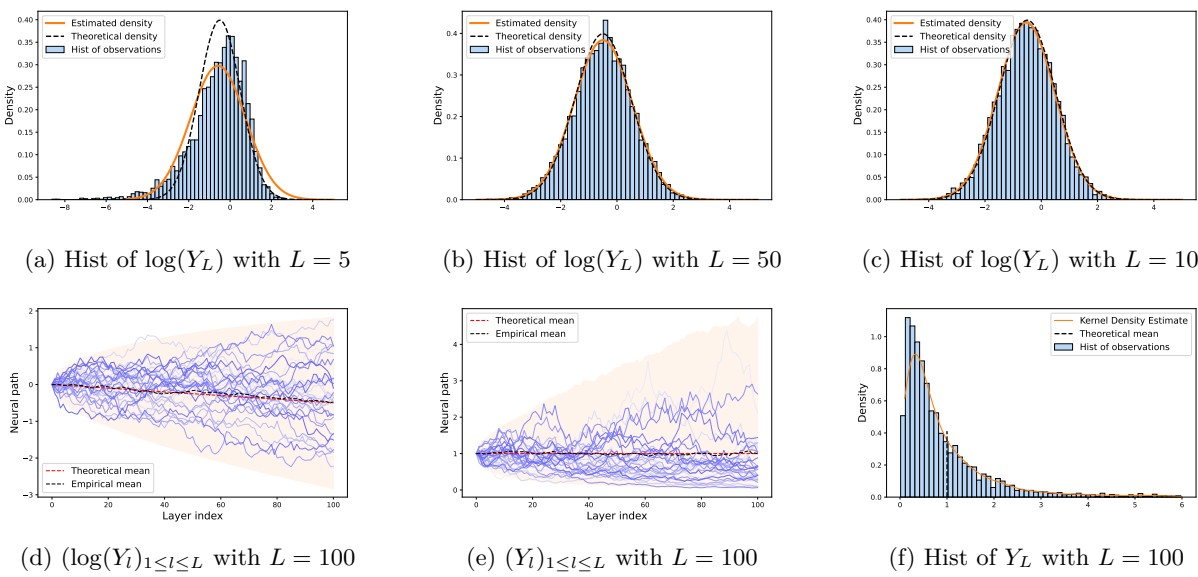

(a) Hist of $\log(Y_L)$ with $L = 5$     (b) Hist of $\log(Y_L)$ with $L = 50$     (c) Hist of $\log(Y_L)$ with $L = 100$

(d) $(\log(Y_l)_{1 \leq l \leq L}$ with $L = 100$     (e) $(Y_l)_{1 \leq l \leq L}$ with $L = 100$     (f) Hist of $Y_L$ with $L = 100$

Figure 1: Empirical verification of Proposition 2. **(a), (b), (d)** Histograms of $\log(Y_L)$ and based on $N = 5000$ simulations for depths $L \in \{5, 50, 100\}$ with $Y_0 = 1$. Estimated density (Gaussian kernel estimate) and theoretical density (Gaussian) are illustrated on the same graphs. **(c), (e)** 30 Simulations of the sequence $(\log(Y_l))_{l \leq L}$ (c) and the sequence $(Y_l)_{l \leq L}$ (e). We call such sequences Neural paths. The results are reported for depth $L = 100$, with $Y_0 = 1$, $\phi$ being the ReLU activation. The theoretical mean of $\log(Y_l)$ is given by $m(l) = -\frac{l}{2L}$ and that of $Y_l$ is equal to $Y_0 = 1$. We also illustrate the 99% confidence intervals, based on the theoretical prediction for $\log(Y_l)$ (Proposition 2), and the empirical Quantiles for $Y_l$. **(f)** Histogram of $Y_L$ based on $N = 5000$ simulations for depth $L = 100$.

which is expected since the dynamics of $X$ describe (only) the infinite-depth limit of the ResNet. More figures are provided in Appendix L.

*Remark:* notice that the log-normal behaviour is a result of the fact that we only consider the case $n = 1$ (width one). Indeed, the single neuron case forces ReLU to act like a linear activation when $X_0 > 0$, and like a 'zero' activation when $X_0 \leq 0$. For general width $n \geq 1$, such behaviour does not hold in general, and usually some coordinates of $X_t$ will be negative while others are non-negative, which implies that the volatility term $\|\phi(X_t)\|$ has non-trivial dependence on $X_t$. We discuss this in more details in Section 4. In the next section, we illustrate a case of an exotic (non-standard) activation function that yields a completely different closed-form distribution of $X_t$.

## 3.2 Exotic activation

With a particular choice of the activation function $\phi$ and mapping $g$, the stochastic process $g(X_t)$ is the solution of well-known type of SDEs known as the Ornstein-Uhlenbeck SDEs. In this case, the activation function is non-standard and involves the inverse of the imaginary error function, a variant of the error function.

**Proposition 4** (Ornstein-Uhlenbeck neural networks). *Let $x \in \mathbb{R}$ such that $x \neq 0$. Consider the following activation function $\phi$*

$$\phi(y) = \exp(h^{-1}(\alpha y + \beta)^2),$$

*where $\alpha, \beta \in \mathbb{R}$ are constants and $h^{-1}$ is the inverse function of the imaginary error function given by $h(z) = \frac{2}{\sqrt{\pi}} \int_0^z e^{t^2} dt$. Let $g$ be the function defined by*

$$g(y) = \alpha\sqrt{\pi} h^{-1}(\alpha y + \beta).$$

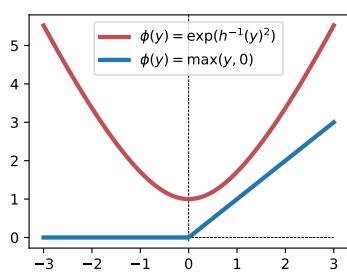

Figure 2: Exotic activation

*Let the process $X_t$ be the solution of Eq. (3)[3]. Then, the stochastic process $g(X_t)$ follows the Ornstein-Uhlenbeck dynamics on $(0, 1]$ given by*

$$dg(X_t) = ag(X_t)dt + 2adB_t, \quad g(X_0) = g(W_{in}x),$$

*where $a = \frac{\pi\alpha^2}{4}$. As a result, conditionally on $X_0$ (fixed $X_0$), we have that for all $t \in [0, 1]$,*

$$g(X_t) \sim \mathcal{N}\left(g(X_0)e^{-at}, \frac{\pi}{2}(1 - e^{-2at})\right),$$

*and the process $X_t$ is distributed as $X_t \sim \alpha^{-1}(h(\alpha^{-1}\pi^{-1/2}\mathcal{N}\left(g(X_0)e^{-at}, \frac{\pi}{2}(1 - e^{-2at})\right)) - \beta)$.*

Fig. 2 shows the graph of the function $\phi(y) = \exp(h^{-1}(y)^2)$ mentioned in Proposition 4 with $\alpha = 1$ and $\beta = 0$. With this choice of the activation function, the infinite-depth network output $X_1$ has the distribution $g^{-1}\left(\mathcal{N}\left(g(X_0)e^{-at}, 2(1 + e^{-2at})\right)\right)$ (conditionally on $X_0$), where $g$ is given in the statement of the proposition. This distribution, although easy to simulate, is different from both the Gaussian distribution that we obtain in the infinite-width limit and the log-normal distribution associated with ReLU activation. This confirms that not only do neural networks exhibit completely different behaviours when the ratio depth-to-width is large, but in this case, that their behaviour is very sensitive to the choice of the activation function. The results of Proposition 4 are empirically confirmed in Fig. 3. The original ResNet given by Eq. (8) with depth $L = 100$ exhibit very similar behaviour to that of the SDE.

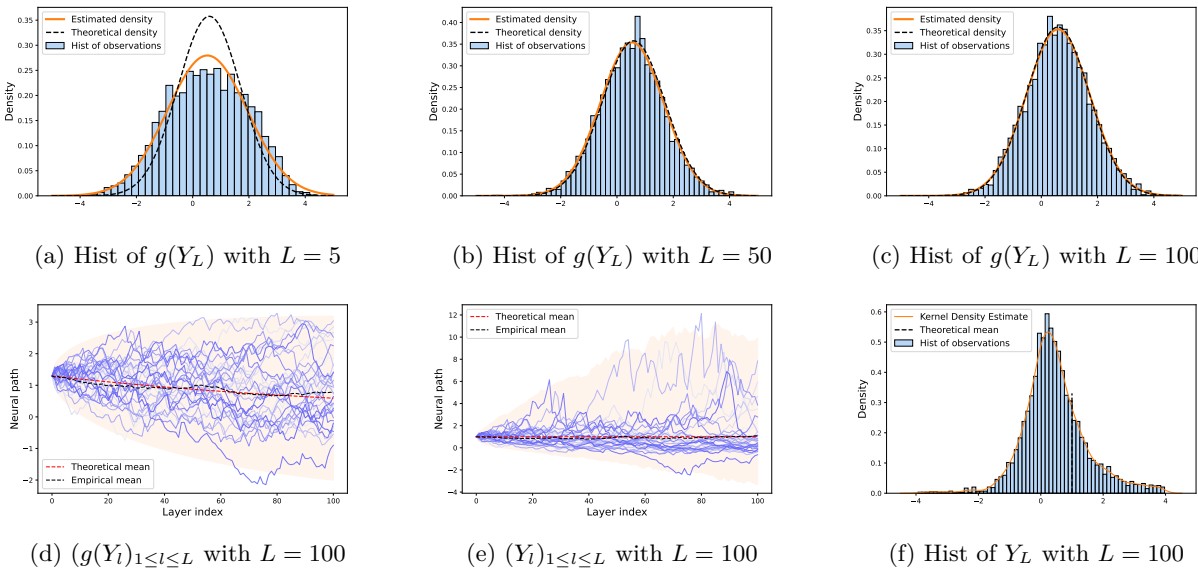

(a) Hist of $g(Y_L)$ with $L = 5$    (b) Hist of $g(Y_L)$ with $L = 50$    (c) Hist of $g(Y_L)$ with $L = 100$

(d) $(g(Y_l))_{1 \le l \le L}$ with $L = 100$    (e) $(Y_l)_{1 \le l \le L}$ with $L = 100$    (f) Hist of $Y_L$ with $L = 100$

Figure 3: Empirical verification of Proposition 4. **(a), (b), (d)** Histograms of $g(Y_L)$ based on $N = 5000$ simulations for depths $L \in \{5, 50, 100\}$ with $Y_0 = 1$. Estimated density (Gaussian kernel estimate) and theoretical density (Gaussian) are illustrated on the same graphs. **(c), (e)** 30 Simulations of the neural paths $(g(Y_l))_{l \le L}$ (c) and $(Y_l)_{l \le L}$ (e). The results are reported for depth $L = 100$, with $Y_0 = 1$, $\phi$ is given un Proposition 4. The theoretical mean of $g(Y_l)$ (conditionally on $Y_0$) is approximated by $m(l) = g(Y_0)e^{-\frac{\pi l}{3L}}$ and that of $Y_l$ is equal to $Y_0 = 1$. We also illustrate the 99% confidence intervals, based on the theoretical prediction for $g(Y_l)$ (Proposition 2), and the empirical Quantiles for $Y_l$. **(f)** Histogram of $Y_L$ based on $N = 5000$ simulations for depth $L = 100$.

---

[3]in Appendix D, we show that the activation function $\phi$ is only locally Lipschitz. Hence, the solution of this SDE exists only in the local sense and the convergence in distribution of $Y_{\lfloor tL \rfloor}$ to $X_t$ is also in the local sense (Proposition 1). However, by continuity of the Brownian path, the stopping times $\tau^L$ and $\tau$ diverge almost surely when $r$ goes to infinity. Therefore, the conclusion of Proposition 4 remains true for all $t \in [0, 1]$. Technical details are provided in Appendix D.

## 4 General width $n \geq 1$

Let $n \geq 1$ and $x \in \mathbb{R}^d$ such that $x \neq 0$. Consider the process $X$ given by the SDE

$$dX_t = \frac{1}{\sqrt{n}}\|\phi(X_t)\|dB_t, \quad X_0 = W_{in}x, \tag{5}$$

where $\phi$ is the activation function, and $B$ is an $n$-dimensional Brownian motion, independent from $W_{in}$. For $n \geq 2$, the coordinates of $X_t$ are dependent which makes the distribution of $X_t$ generally intractable. This intractability is purely due to this dependence between the coordinates and not to the non-linearity itself. To understand this, we have added in Appendix K a comprehensive analysis of the case of the identity activation and other piece-wise activation functions.

Intuitively, if for some $s$, if $\|\phi(X_s)\| = 0$ in Eq. (5), then for all $t \geq s$, $X_t = X_s$ since the increments '$dX_t$' are all zero for $t \geq s$. This holds for any choice of the activation function $\phi$, provided that the process $X$ exists, i.e. the SDE has a unique solution. We summarize this in the next lemma.

**Lemma 1** (Collapse). *Let $x \in \mathbb{R}^d$ such that $x \neq 0$, and $\phi : \mathbb{R} \to \mathbb{R}$ be a Lipschitz function. Let $X$ be the solution of the SDE given by Eq. (5). Assume that for some $s \geq 0$, $\phi(X_s) = 0$. Then, for all $t \geq s$, $X_t = X_s$, almost surely.*

Lemma 1 is a particular case of Lemma 7 in the Appendix. The proof consists of using the uniqueness of the solution of Eq. (5) when the volatility term is Lipschitz. This result is trivial in the finite depth case (Eq. (1)). When there exists $s$ such that $\phi(X_s) = 0$, the process $X$ becomes constant (equal to $X_s$) for all $t \geq s$ (almost surely). We call this phenomenon *process collapse*. In the case of finite-depth networks (Eq. (1)), we call the same phenomenon *network collapse*. Understanding when, and whether, such event occurs is useful since it has significant implications on the the large depth behaviour of neural networks. Indeed, if such event occurs, it would mean that increasing depth has no effect on the network output after some time $s$ (or approximately, after layer index $\lfloor sL \rfloor$). In the next result, we show that under mild conditions on the activation function, process collapse is a zero-probability event.

### 4.1 Network collapse

The next result gives (mild) sufficient conditions on the activation function so that the process $X$ almost surely does not collapse. In the proof, we use Itô 's lemma in the multi-dimensional case, which states that for any function $g : \mathbb{R}^n \to \mathbb{R}$ that is $\mathcal{C}^2(\mathbb{R}^n)$, we have that

$$dg(X_t) = \nabla g(X_t)^\top dX_t + \frac{1}{2n}\|\phi(X_t)\|^2 \mathrm{Tr}\left[\nabla^2 g(X_t)\right].$$

**Lemma 2.** *Let $x \in \mathbb{R}^d$ such that $x \neq 0$, and consider the stochastic process $X$ given by the following SDE*

$$dX_t = \frac{1}{\sqrt{n}}\|\phi(X_t)\|dB_t, \quad t \in [0, \infty), \quad X_0 = W_{in}x,$$

*where $\phi(z) : \mathbb{R} \to \mathbb{R}$ is Lipschitz, injective, $\mathcal{C}^2(\mathbb{R})$ and satisfies $\phi(0) = 0$, and $\phi'$ and $\phi''\phi$ are bounded on $\mathbb{R}$, and $(B_t)_{t\geq 0}$ is an $n$-dimensional Brownian motion independent from $W_{in} \sim \mathcal{N}(0, d^{-1}I)$. Let $\tau$ be the stopping time given by $\tau = \min\{t \geq 0 : \phi(X_t) = 0\}$. Then, we have that*

$$\mathbb{P}\left(\tau = \infty\right) = 1.$$

The proof of Lemma 2 is provided in Appendix G. Many standard activation functions satisfy the conditions of Lemma 2. Examples include Hyperbolic Tangent $\mathrm{Tanh}(z) = \frac{e^{2z}-1}{e^{2z}+1}$, and smooth versions of ReLU activation such as GeLU given by $\phi_{GeLU}(z) = z\Psi(z)$ where $\Psi$ is the cumulative distribution function of the standard Gaussian variable, and Swish (or SiLU) given by $\phi_{Swish}(z) = zh(z)$ where $h(z) = (1+e^{-z})^{-1}$ is the Sigmoid function. The result of Lemma 2 can be extended to the case when $\phi$ is the ReLU function with miner changes.

**Lemma 3.** *Consider the stochastic process* (8) *given by the SDE*

$$dX_t = \frac{1}{\sqrt{n}}\|\phi(X_t)\|dB_t\,, \quad t \in [0, \infty), \quad X_0 = W_{in}x,$$

*where $\phi$ is the ReLU activation function, and $(B_t)_{t \geq 0}$ is an $n$-dimensional Brownian motion independent from $W_{in} \sim \mathcal{N}(0, d^{-1}I)$. Let $\tau$ be the stopping time given by*

$$\tau = \min\{t \geq 0 : \|\phi(X_t)\| = 0\} = \min\{t \geq 0 : \forall i \in [n], X_t^i \leq 0\}.$$

*Then, we have that*

$$\mathbb{P}\left(\tau = \infty \mid \|\phi(X_0)\| > 0\right) = 1.$$

*As a result, we have that*

$$\mathbb{P}(\tau = \infty) = 1 - 2^{-n}.$$

The proof of Lemma 3 relies on a particular choice of a sequence of functions $(\phi_m)_{m \geq 1}$ that approximate the ReLU activation $\phi$. Details are provided in Appendix G.

The result of Lemma 3 shows that for all $T > 0$, with probability 1, if there exists $j \in [n]$ such that $X_0^j > 0$, then for all $t \in [0, T]$, there exists a coordinate $i$ such that $X_t^i > 0$, which implies that the volatility of the process $X$ given by $\frac{1}{\sqrt{n}}\|\phi(X_t)\|$ does not vanish in finite time $t$. Notably, this implies that for any $t \in [0, 1]$, the norm of post-activations given by $\|\phi(X_t)\|$ does not vanish (with probability 1). This is important as it ensures that the vector $\phi(X_t)$, which represents the post-activations in the infinite-depth network, does not vanish, and therefore the process $X_t$ does not get stuck in an absorbent point. The dependence between the co-ordinates of the process $X_t$ is crucial in this result. In the opposite case where $X_t$ are independent, the event $\{\|\phi(X_t)\| = 0\}$ has probability $2^{-n}$. Notice also that this result holds only in the infinite-depth limit. With finite-depth ResNet (Eq. (1)) with ReLU activation, it is not hard to show that the network collapse event $\{\exists l \in [L], \text{ s.t. } \|\phi(Y_{\lfloor tL \rfloor})\| = 0\}$ has non-zero probability. However, as the depth increases, the probability of network collapse goes to zero. Fig. 4 shows the probability of network collapse for a finite-width and depth ResNet (Eq. (1)). As the depth $L$ increases, it becomes unlikely that the network collapses. This is in agreement with our theoretical prediction that the infinite-depth network represented by the process $X_t$ has zero-probability collapse event, conditionally on the fact that $\|\phi(X_0)\| > 0$. The probability of neural collapse also decreases with width, which is expected, since it becomes less likely to have all pre-activations non-positive as the width increases.

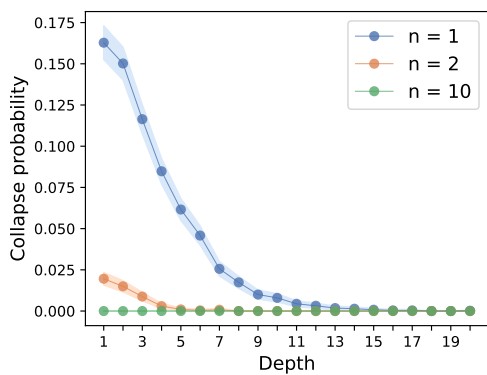

Figure 4: Probability of the event $\{\exists l \in [L] \text{ such that } \phi(Y_l) = 0\}$ (collapse) for varying widths and depths. The probability and the 95% confidence intervals are estimated using $N = 5000$ samples.

### 4.2 Post-activation norm

As a result of Lemma 3, conditionally on $\|\phi(X_0)\| > 0$, we can safely consider manipulating functions that require positiveness such as the logarithm of the norm of the post-activations. In the next result, we show that the norm of the post-activations has a distribution that resembles the log-normal distribution. We call this Quasi Geometric Brownian Motion distribution (Quasi-GBM).

**Theorem 1** (Quasi-GBM behaviour of the post-activations norm)**.** *We have that for all $t \in [0, 1]$,*

$$\|\phi(X_t)\| = \|\phi(X_0)\|\exp\left(\frac{1}{\sqrt{n}}\hat{B}_t + \frac{1}{n}\int_0^t \mu_s ds\right), \quad \text{almost surely},$$

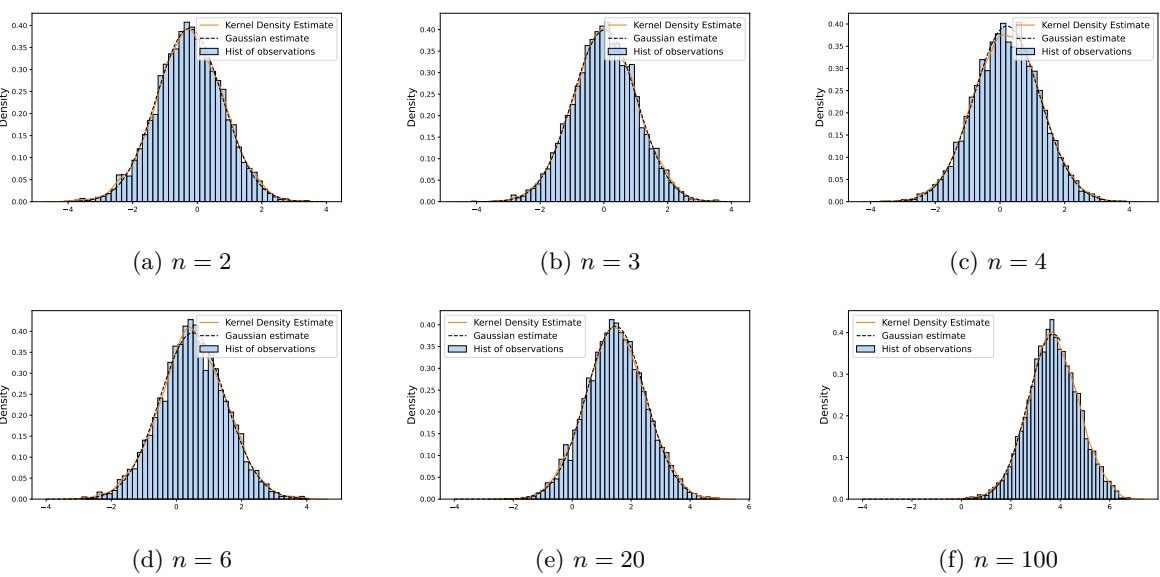

Figure 5: Histogram of $\sqrt{n}\log(\|\phi(Y_L)\|/\|\phi(Y_0)\|)$ for depth $L = 100$ and different widths $n \in \{2, 3, 4, 6, 20, 100\}$ based on $N = 5000$ simulations. Gaussian density estimate and (Gaussian) kernel density estimates are shown. We observe a great match between the best Gaussian estimate and the empirical distribution, which confirms the quasi-log-normal theoretical predictions from Theorem 1.

*where $\mu_s = \frac{1}{2}\|\phi'(X_s)\|^2 - 1$, and $(\hat{B})_{t \geq 0}$ is a one-dimensional Brownian motion. As a result, for all $0 \leq s \leq t \leq 1$*

$$\mathbb{E}\left[\log\left(\frac{\|\phi(X_t)\|}{\|\phi(X_s)\|}\right) \mid \|\phi(X_0)\| > 0\right] = \left(\frac{(1-2^{-n})^{-1}}{4} - \frac{1}{n}\right)(t-s).$$

*Moreover, for $n \geq 2$, we have*

$$\mathrm{Var}\left[\log\left(\frac{\|\phi(X_t)\|}{\|\phi(X_s)\|}\right) \mid \|\phi(X_0)\| > 0\right] \leq \left(n^{-1/2} + \Gamma_{s,t}^{1/2}\right)^2 (t-s),$$

*where $\Gamma_{s,t} = \frac{1}{4}\int_s^t \left(\left(\mathbb{E}\phi'(X_u^1)\phi'(X_u^2) - \frac{(1-2^{-n})^2}{4}\right) + n^{-1}\left(\frac{1-2^{-n}}{2} - \mathbb{E}\phi'(X_u^1)\phi'(X_u^2)\right)\right) du.*

Different tools from stochastic calculus and probability theory are used in the proof of Theorem 1. Technical details are provided in Appendix H. The first result in the theorem suggests that the norm of the post-activations has a quasi-log-normal distribution (conditionally on $X_0$). The first term in the exponential is Gaussian ($n^{-1/2}\hat{B}_t$) and the second term depends on $n^{-1}\mu_s$, which involves an average over $(\phi'(X_s^i))_{1 \leq i \leq n}$. In the large width limit, this average concentrates around its mean as we will see in Theorem 2. In Fig. 5, we show the histogram of $\sqrt{n}\log(\|\phi(Y_L)\|/\|\phi(X_0)\|)$ for depth $L = 100$ and varying widths $n$. Surprisingly, the log-normal approximation fits the empirical distribution very well even for small widths $n \in \{2, 3, 4, 6\}$ for which the term $n^{-1}\mu_s$ is not necessarily close to its mean[4]. More interestingly, the result of Theorem 1 sheds light on an intriguing change of regime that occurs between widths $n = 3$ and $n = 4$. Indeed, for $n \leq 3$, the logarithmic growth factor of the norm of the post-activations $\|\phi(X_t)\|$ tends to decrease with depth on average, while it increases for $n \geq 4$. When $n = 4$, the average growth is positive although very small. This change of regime phenomenon suggests that for $n \leq 3$, the random variable $\|\phi(X_t)\|/\|\phi(X_s)\|$ has significant probability mass in the region $(0, 1)$. This probability mass tends to $0$ as $n$ increases since $\|\phi(X_t)\|/\|\phi(X_s)\|$ converges to a deterministic constant (we will see this in the next theorem), and the variance upperbound in Theorem 1 converges to $0$ when $n$ goes to infinity, which can be explained by the fact that $\mathbb{E}\phi'(X_u^1)\phi'(X_u^2) \overset{n \to \infty}{\longrightarrow} 1/4$ (the coordinates become independent in the large width limit, see next

---

[4]We currently do not have a rigorous explanation for this effect. A possible explanation for this empirical result is that the integral over $\mu_s$ has some 'averaging' effect.

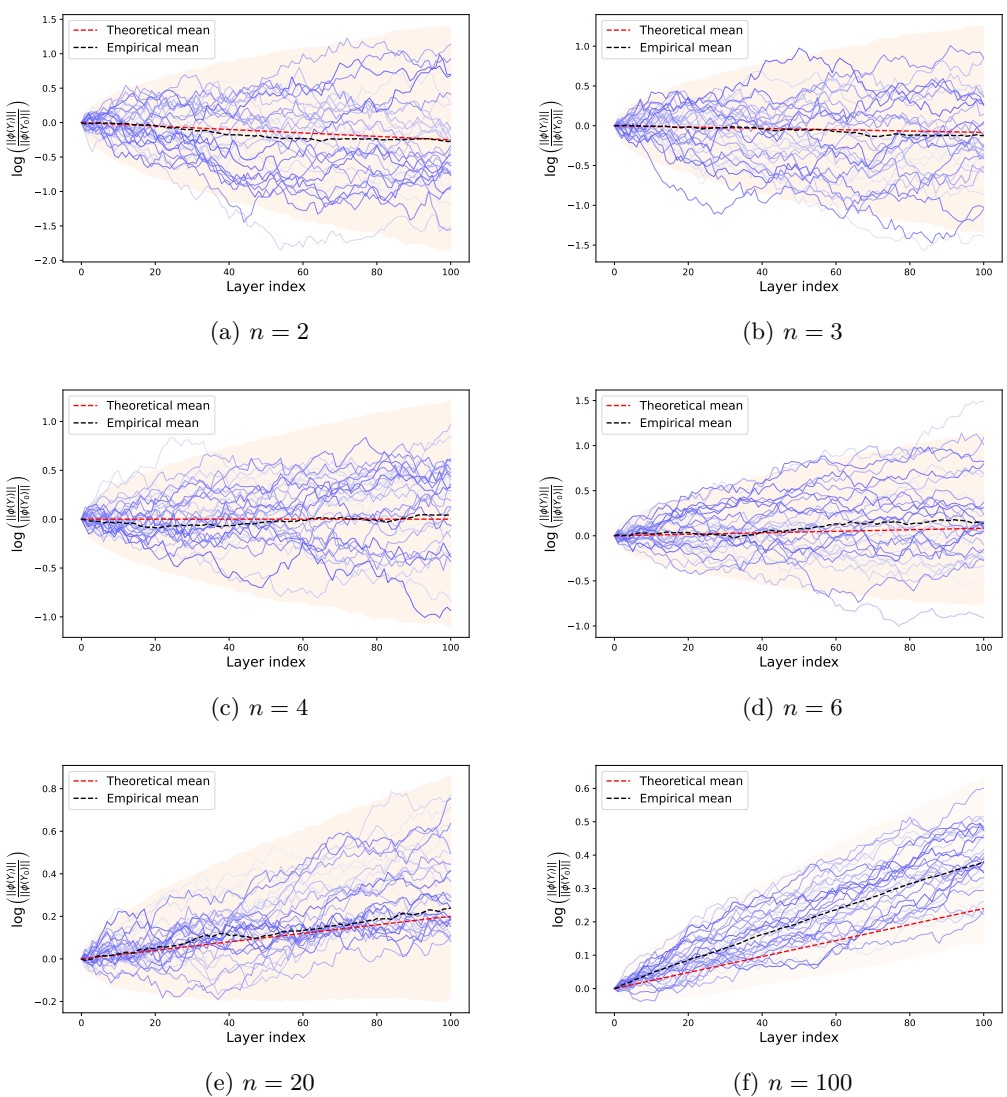

Figure 6: 30 simulations of the sequence $(\log(\|\phi(Y_l)\|/\|\phi(Y_0)\|))_{1 \le l \le L}$ for depth $L = 100$ and different widths $n \in \{2, 3, 4, 6, 20, 100\}$. Theoretical means from Theorem 1 are shown in red dashed lines and compared to their empirical counterparts. We observe that when the ratio $L/n$ increases (especially for $n = 100$), the empirical mean also increases and becomes significantly different from the theoretical prediction.

theorem). Experiments showing this concentration are provided in Appendix L.5. In Fig. 6, we simulate 30 neural paths (i.e. $(Y_l)_{1 \le l \le L}$) for depth $L = 100$ and compute the logarithmic factor $\log(\|\phi(Y_l)\|/\|\phi(Y_0)\|)$. An excellent match with the theoretical results is observed for widths $n \in \{2, 3, 4, 6, 20\}$. A mismatch between theory and empirical results appears when $n = 50$, which is expected, since the theoretical results of Theorem 1 yield good approximations only when $n \ll L$.

Notice that the case of $n = 1$ matches the result of Proposition 3. Indeed, the latter implies that conditionally on $\phi(X_0) > 0$, we have $\log(\phi(X_t)/\phi(X_0)) = \log(X_t/X_0) \sim -t/2 + B_t$ where $B$ is a one-dimensional Brownian motion, and where we have used the fact that $X_t > 0$ for all $t$. This result can be readily obtained from Theorem 1 by setting $n = 1$.

An interesting question is that of the infinite-width limit of the process $X_t$, which corresponds to the sequential limit infinite-depth-then-infinite-width of the ResNet $Y_{\lfloor tL \rfloor}$ (Eq. (1)). We discuss this in the next section.

### 4.3 Infinite-width limit of infinite-depth networks

In the next result, we show that when the width goes to infinity, the ratio $\|\phi(X_t)\|/\|\phi(X_0)\|$ concentrates around a layer dependent ($t$-dependent) constant. In this limit, the coordinates of $X_t$ converge in $L_2$ to a Mckean-Vlasov process, which allows us to recover the Gaussian behaviour of the pre-activations of the ResNet. We later compare this with the converse sequential limit infinite-width-then-infinite-depth where the pre-activations are also normally distributed, and show a key difference in the variance of the Gaussian distribution.

**Theorem 2** (Infinite-depth-then-infinite-width limit)**.** *For $0 \le s \le t \le 1$, we have*

$$\log\left(\frac{\|\phi(X_t)\|}{\|\phi(X_s)\|}\right)\mathbb{1}_{\{\|\phi(X_0)\|>0\}} \xrightarrow[n\to\infty]{} \frac{t-s}{4}, \quad and, \quad \frac{\|\phi(X_t)\|}{\|\phi(X_s)\|}\mathbb{1}_{\{\|\phi(X_0)\|>0\}} \xrightarrow[n\to\infty]{} \exp\left(\frac{t-s}{4}\right).$$

*where the convergence holds in $L_1$.*

*Moreover, we have that*

$$\sup_{i\in[n]}\mathbb{E}\left(\sup_{t\in[0,1]}|X_t^i - \tilde{X}_t^i|^2\right) = \mathcal{O}(n^{-1}),$$

*where $X_t^i$ is the solution of the following (Mckean-Vlasov) SDE*

$$d\tilde{X}_t^i = \left(\mathbb{E}\phi(\tilde{X}_t^i)^2\right)^{1/2} dB_t^i, \quad \tilde{X}_0^i = X_0^i.$$

*As a result, the pre-activations $Y^i_{\lfloor tL\rfloor}$ (Eq. (1)) converge in distribution to a Gaussian distribution in the limit infinite-depth-then-infinite-width*

$$\forall i \in [n], \quad Y^i_{\lfloor tL\rfloor} \xrightarrow{L\to\infty \ then \ n\to\infty} \mathcal{N}(0, d^{-1}\|x\|^2 \exp(t/2)).$$

The proof of Theorem 2 requires the use of a special variant of the Law of large numbers for non *iid* random variables, and a convergence result of particle systems from the theory of Mckean-Vlasov processes. Details are provided in Appendix I. In neural network terms, Theorem 2 shows that the logarithmic growth factor of the norm of the post-activations, given by $\log\left(\|\phi(Y_{\lfloor tL\rfloor})\|/\|\phi(Y_{\lfloor sL\rfloor})\|\right)$, converges to $(t-s)/4$ in the sequential limit $L \to \infty$, then $n \to \infty$. More importantly, the pre-activations $Y^i_{\lfloor tL\rfloor}$ converge in distribution to a zero-mean Gaussian distribution in this limit, with a layer-dependent variance. In the converse sequential limit, i.e. $n \to \infty$, then $L \to \infty$, the limiting distribution of the pre-activations $Y^i_{\lfloor tL\rfloor}$ is also Gaussian with the same variance. We show this in the following result, which uses Lemma 5 in (Hayou et al., 2021).

**Theorem 3** (Infinite-width-then-infinite-depth limit)**.** *Let $t \in [0,1]$. Then, in the limit $\lim_{L\to\infty}\lim_{n\to\infty}$ (infinite width, then infinite depth), we have that*

$$\frac{\|\phi(Y_{\lfloor tL\rfloor})\|}{\|\phi(Y_0)\|}\mathbb{1}_{\{\|\phi(Y_0)\|>0\}} \longrightarrow \exp\left(\frac{t}{4}\right),$$

*where the convergence holds in probability.*

*Moreover, the pre-activations $Y^i_{\lfloor tL\rfloor}$ (Eq. (1)) converge in distribution to a Gaussian distribution in the limit infinite-width-then-infinite-depth*

$$\forall i \in [n], \quad Y^i_{\lfloor tL\rfloor} \xrightarrow{n\to\infty \ then \ L\to\infty} \mathcal{N}(0, d^{-1}\|x\|^2 \exp(t/2)).$$

The proof of Theorem 3 is provided in Appendix J. We use existing results from Hayou et al. (2021) on the infinite-depth asymptotics of the neural network Gaussian process (NNGP). It turns out that the order to the sequential limit (taking the width to infinity first, then taking the depth to infinity, or the converse) does not affect the limiting distribution, which is a Gaussian with variance $\propto \exp(t/2)$. Intuitively, by taking the width to infinity first, we make the coordinates independent from each other, and the processes $(Y_l^i)_{1\le l\le L}$ become *iid* Markov chains. Taking the infinite-depth limit after the infinite-width limit consists of taking the

infinite-depth limit of one-dimensional Markov chains. On the other hand, when we take depth to infinity first, the coordinates $(X_t^i)_{1 \le i \le n}$ remain dependent (through the volatility term $n^{-1/2}\|\phi(X_t)\|$), which results in the Quasi-log-normal behaviour of the norm of the post-activations (Theorem 1). Taking the width to infinity then yields an asymptotic norm of the post-activations equal to $\|\phi(X_0)\| \exp(t/2)$ (Theorem 2) which is the same norm in the converse limit (Theorem 3). It remains to take the width to infinity to decouple the coordinates and obtain the Gaussian distribution (through the Mckean-Vlasov dynamics). Knowing that the variance of the pre-activations is mainly determined by the norm of the post-activations (Eq. (5)), we can see why the variance is similar in both sequential limits.

## 5 Discussion on the case of multiple inputs

The result of Proposition 1 can be easily generalized to the multiple input case, and the resulting dynamics is still an SDE. The generalization to the multiple inputs case is given by Proposition 5 in the Appendix.

An important question in the literature on infinite-width neural networks is the behaviour of the correlation of the pre-activations (or the post-activations) for different inputs $a$ and $b$, which is given by $\frac{\langle Y_{\lfloor tL \rfloor}(a), Y_{\lfloor tL \rfloor}(b) \rangle}{\|Y_{\lfloor tL \rfloor}(a)\|\|Y_{\lfloor tL \rfloor}(b)\|}$. This correlation can be as a geometric measure of the information as it propagates through the network. In the infinite-width-then-depth limit, this correlation (generally) converges to a degenerate limit (a constant value) which results in either a constant or a sharp landscape of the network output and causes gradient exploding/vanishing issues (Schoenholz et al., 2017; Yang & Schoenholz, 2017; Hayou et al., 2019a). Techniques such block scaling (Hayou et al., 2021), or kernel shaping (Zhang et al., 2022; Martens et al., 2021) solve this problem and ensure that the correlation is well-behaved in the large depth limit.

In our case, when the width $n$ is finite and the depth $L$ is taken to infinity, we can define the correlation for two inputs $a \ne b$ and time $t \in [0, 1]$ by

$$c_t(a, b) \stackrel{def}{=} \frac{\langle X_t(a), X_t(b) \rangle}{\|X_t(a)\|\|X_t(b)\|}.$$

Using Itô's lemma, $c_t$ has dynamics of the form

$$dc_t(a, b) = \Psi(X_t(a), X_t(b))dB_t, \tag{6}$$

for some non-trivial mapping $\Psi$. Unfortunately, this kind of dynamics (which is not an SDE) is generally intractable, and we are currently investigating these dynamics for future work. However, since we scale the ResNet blocks with the factor $1/\sqrt{L}$ (Eq. (1)), which is the same scaling that solves the degeneracy issue in the infinite-width-then-depth limit (Hayou et al., 2021), it should be expected that the correlation kernel $c_t$ does not converge to a degenerate limit. In Fig. 7, we simulate the correlation path in a ResNet of depth $L = 200$ and width $n = 20$. The paths exhibits some level of stochasticity but no degeneracy can be observed. Understanding

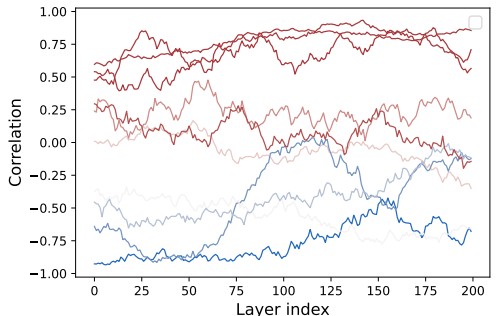

Figure 7: 10 Simulations of the correlation path $\left( \frac{\langle Y_{\lfloor tL \rfloor}(a), Y_{\lfloor tL \rfloor}(b) \rangle}{\|Y_{\lfloor tL \rfloor}(a)\|\|Y_{\lfloor tL \rfloor}(b)\|} \right)_{1 \le l \le L}$ for depth $L = 200$, width $n = 20$, and different $(a, b)$ (different initial correlations $c_0$). The color code depends only on the initial correlation value $c_0$ (red for the largest correlations values)

the correlation dynamics (Eq. (6)) in the infinite-depth limit of finite-width networks is an interesting open question. The infinite-width limit[5] of these dynamics is also an interesting open question. We leave this for future work.

## 6 Practical implications

Our theoretical analysis has many interesting implications from a practical standpoint. Here we summarize some key insights form our results.

---

[5]The infinite-width limit of infinite-depth correlations

**Initialization and stability in the large depth limit.** An important factor pertaining to the trainability of neural networks is the behaviour of the neurons (pre/post-activations). Ensuring that the neurons are well-behaved at initialization is crucial for training since the first step of any gradient-based training algorithm depends on the values of the neurons at initialization. This has led to interesting developments in initialization schemes for MLPs such as the Edge of Chaos (Poole et al., 2016; Schoenholz et al., 2017) which ensures that the variance of the pre-activation does not (exponentially) vanish or explode in the large depth limit. In the case of ResNet, we know from the existing theory on the infinite-width limit of neural networks that scaling the residual blocks with $1/\sqrt{L}$ stabilizes the pre/post-activations in the large depth limit (Hayou et al., 2021). Hence, we do not need a special initialization scheme with this scaling. However, one could argue that this (approximately) ensures stability *only* when the width is much larger than the depth. What about the other cases when $n \approx L$ or $n \ll L$? the last case can be studied by fixing the width and taking the depth to infinity. In our paper, we not *only* show that the neurons remain stable in fixed-width large-depth networks, but we fully characterize their behaviour when the depth is infinite and show that it follows an SDE in this limit. To summarize, we show that initializing ResNet Eq. (1) with standard Gaussian random variables and scaling the blocks with $1/\sqrt{L}$ ensures stability inside the network in large-depth (fixed-width) networks (notice that this is actually equivalent to scaling the variance of the initialization weights with $1/L$, which can be seen as an initialization scheme). Intuitively, by stabilizing the pre-activations, we also stabilize the gradients. To confirm this intuition, we show in Fig. 8 the evolution of gradient norms as they back-propagate through the network. This experiment was conducted by fixing the last layer's gradient to a constant value and back-propagating the gradient from there. The result shows that the $1/\sqrt{L}$ scaling, along with standard Gaussian initialization, ensure well-behaved gradients which is a desirable property for gradient-based training. Another interesting property of the Edge of Chaos initialization scheme for MLPs is that it ensures that correlation kernel (correlation between the pre-activations for different inputs) does not exponentially converge to a degenerate value (constant value)[6]. We discussed some aspects of the correlation kernel in Section 5 and showed empirically that with the $1/\sqrt{L}$ scaling, the correlation is well-behaved and does not converge to degenerate values (Fig. 7).

**Network collapse.** Another issue that could occur in finite-width networks is that of network collapse, i.e. when the pre-activations in a hidden layer are all negative, which causes the post-activations to be all zero. In ResNet (Eq. (1)), this implies that increasing depth

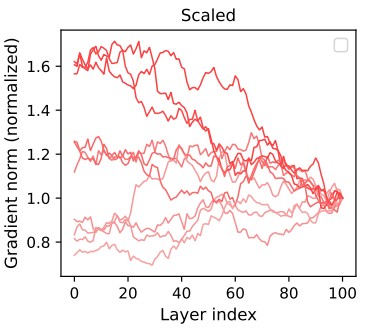

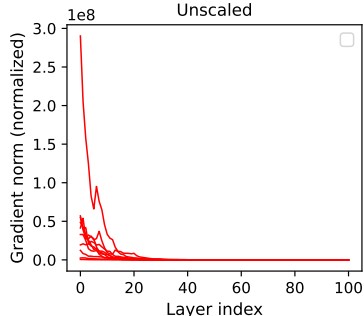

Figure 8: 10 Simulations of the gradient norm for scaled ResNet (Eq. (1)) and non-scaled ResNet ($Y_l = Y_{l-1} + W_l \phi(Y_{l-1})$) for depth $L = 100$ and width $n = 10$. We normalize the gradients norms by the gradient norm of the last layer. The color code depends only on the ratio of the graident norm at the first layer to that of the last layer (dark red for the largest values). Without scaling, the gradient norm explodes (highly likely). The $1/\sqrt{L}$ stabilizes the gradients as they back-propagate through the network.

beyond some level has no effect on the network output. This is problematic since the weights in those 'inactive' layers have zero gradient and thus will not be updated when such event occurs. A simple way to understand network collapse is to see what happens at initialization. When the width $n$ is sufficiently large, one can expect that such event is unlikely to occur. What about small-width neural networks? we offer a simple answer to this question: for finite-width neural networks, increasing the depth $L$ ensures that such event is unlikely to happen. This is true even for extremely small widths, e.g. $n = 2, 3$, which is counter-intuitive. Empirical results in Fig. 4 support this theoretical prediction.

---

[6] The correlation still converges to 1 with an EOC initialization. The benefit of the EOC lies in the fact that the convergence rate is much slower (polynomial Vs exponential) (Schoenholz et al., 2017; Hayou et al., 2019a)

**No universal kernel regime.** An interesting application of fixed-depth infinite-width neural network is the so-called Neural Network Gaussian Process (NNGP). This is the Gaussian process limit of neural networks, that can be used to perform posterior inference and obtain uncertainty estimates (Lee et al., 2018). The converse case, i.e. fixed-width infinite-depth, has been however poorly understood, and the question of whether the infinite-depth limit of finite-width networks has some universal behaviour has been an open question since. We addressed this question in this work and showed that the limit (in the case of the ResNet architecture Eq. (1)) does not admit a universal distribution (e.g. Gaussian process in the infinite-width limit). More precisely, this limit is highly sensitive to the choice of the activation function.

**What about infinite-depth-then-width?** the infinite-depth limit of infinite-width neural networks has been studied in the literature (Hayou et al., 2019a; 2020). It is known that in this limit, the network behaves as a Gaussian process with a well-defined kernel. What about the converse limit, i.e. infinite-width limit of infinite-depth networks? this has been so far an open question, and our work addresses one part of it. We show that the marginal distributions are zero-mean Gaussians with the same variance as in the infinite-width-then-depth limit. Characterizing the full covariance kernel is still however an open question (see Section 5 for a discussion on this topic).

## 7   Conclusion and limitations

Understanding the limiting laws of randomly initialized neural networks is important on many levels. Primarily, understanding these limiting laws allows us to derive new designs that are immune to exploding/vanishing pre-activations/gradients phenomena. Next, they also enable a deeper understanding of overparameterized neural networks, and (often) yield many interesting (and simple) justifications to the apparent advantage of overparameterization. So far, the focus has been mainly on the infinite-width limit (and infinite-width-then-infinite-depth limit) with few developments on the joint limit. Our work adds to this stream of papers by studying the infinite-depth limit of finite-width neural networks. We showed that unlike the infinite-width limit, where we always obtain (under some mild conditions on the activation function) a Gaussian distribution, the infinite-depth limit is highly sensitive to the choice of the activation function; using the Itô 's lemma, we showed how we can obtain certain known distributions by carefully tuning the activation function. In the general width limit, we showed an important characteristic of infinite-depth neural networks with general activation functions (including ReLU, conditionally on $\|\phi(X_0)\| > 0$): the probability of process collapse is zero, meaning that with probability one, the process $X_t$ does not get stuck at any absorbent point. This is not true for finite-depth ResNets as we can see in Fig. 4, which highlights the fact that as we increase depth, the collapse probability tends to decrease, and eventually converges to zero in the infinite-depth limit, which is in agreement with our results.

This work, although novel in many aspects, is still far from depicting a complete picture of the infinite-depth limit of finite-width networks. There are still numerous interesting open questions in this research direction. Indeed, one of these is the dynamics of the gradient, and more specifically the behaviour of the NTK in the infinite-depth limit of finite-width neural networks. For instance, we already know that in the joint infinite-width-depth limit of MLPs, the NTK is random (Hanin, 2019); but what happens when the width is fixed and the depth goes to infinity? In the MLP case, a degenerate NTK should be expected. Henceforth, questions remain as to whether a suitable scaling leads to interesting (non-degenerate) infinite-depth limit of the NTK as is the case of the infinite-depth limit of infinite-width NTK (Hayou et al., 2021).

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

# A   Discussion on the infinite-width limit

The infinite-width limit of neural network architectures has been extensively studied in the literature, and has led to many interesting theoretical and algorithmic innovations. We summarize these results below.

- *Initialization schemes*: the infinite-width limit of neural networks has been extensively studied in the literature. In particular, for multi-layer perceptrons (MLP), a new initialization scheme that stabilizes forward and backward propagation (in the infinite-width limit) was derived in (Poole et al., 2016; Schoenholz et al., 2017). This initialization scheme is known as the Edge of Chaos, and empirical results show that it significantly improves performance. In Yang & Schoenholz (2017); Hayou et al. (2021), the authors derived similar results for the ResNet architecture, and showed that this architecture is *placed* by-default on the Edge of Chaos for any choice of the variances of the initialization weights.

- *Gaussian process behaviour*: Multiple papers (e.g. Neal (1995); Lee et al. (2018); Yang (2020); Matthews et al. (2018); Hron et al. (2020)) studied the weak limit of neural networks when the width goes to infinity. The results show that a randomly initialized neural network (with Gaussian weights) has a similar behaviour to that of a Gaussian process, for a wide range of neural architectures, and under mild conditions on the activation function. In Lee et al. (2018), the authors leveraged this result and introduced the neural network Gaussian process (NNGP), which is a Gaussian process model with a neural kernel that depends on the architecture and the activation function. Bayesian regression with the NNGP showed that NNGP surprisingly achieves performance close to the one achieved by an SGD-trained finite-width neural network. The large depth limit of this Gaussian process was studied in Hayou et al. (2021), where the authors showed that with proper scaling, the infinite-depth (weak) limit is a Gaussian process with a universal kernel[7].

- *Neural Tangent Kernel (NTK)*: the infinite-width limit of the NTK is the so-called NTK regime or Lazy-training regime. This topic has been extensively studied in the literature. The optimization and generalization properties (and some other aspects) of the NTK have been studied in Liu et al. (2022); Arora et al. (2019); Seleznova & Kutyniok (2022); Hayou et al. (2019b). The large depth asymptotics of the NTK have been studied in (Hayou et al., 2020; 2022; Jacot et al., 2022; Xiao et al., 2020). We refer the reader to Jacot (2022) for a comprehensive discussion on the NTK.

# B   Review of Stochastic Calculus

In this section, we introduce the required mathematical framework and tools to handle stochastic differential equations (SDEs). We suppose that we have a probability space $(\Omega, \mathcal{F}, \mathbb{P})$, where $\Omega$ is the event space, $\mathbb{P}$ is the probability measure, and $\mathcal{F}$ is the sigma-algebra associated with $\Omega$. For $n \geq 1$, we denote by $B$ the standard $n$-dimensional Brownian motion, and $\mathcal{F}_t$ its natural filtration. Equipped with $(\mathcal{F}_t)_{t \geq 0}$, we say that the probability space $(\Omega, \mathcal{F}, (\mathcal{F}_t)_{t \geq 0}, \mathbb{P})$ is a filtered probability space. $\mathcal{F}_t$ is the collection of events that are measurable up to time $t$, i.e. can be verified if we have knowledge of the Brownian motion $B$ (and potentially some other independent source such as the initial condition of a process $X$ defined by a $B$-driven stochastic differential equation) up to time $t$. We are now ready to define a special type of stochastic processes known as Itô processes.

## B.1   Existence and uniqueness

**Definition 1** (Itô diffusion process). *A stochastic process $(X_t)_{t \in [0,T]}$ valued in $\mathbb{R}^n$ is called an Itô diffusion process if it can be expressed as*

$$X_t = X_0 + \int_0^t \mu_s ds + \int_0^t \sigma_s dB_s,$$

*where $B$ is a $n$-dimensional Brownian motion and $\sigma_t \in \mathbb{R}^{n \times n}, \mu \in \mathbb{R}^n$ are predictable processes satisfying $\int_0^T (\|\mu_s\|_2 + \|\sigma_s \sigma_s^\top\|_2) ds < \infty$ almost surely.*

---

[7]A kernel is called universal when any continuous function on some compact set can be approximated arbitrarily well with kernel features.

The following result gives conditions under which a strong solution of a given SDE exists, and is unique.

**Theorem 4** (Thm 8.3 in Tankov & Touzi (2018)). *Let $n \geq 1$, and consider the following SDE*

$$dX_t = \mu(t, X_t)dt + \sigma(t, X_t)dB_t, \quad X_0 \in L_2,$$

*where $B$ is a $m$-dimensional Brownian process for some $m \geq 1$, and $\mu : \mathbb{R}^+ \times \mathbb{R}^n \to \mathbb{R}^n$ and $\sigma : \mathbb{R}^+ \times \mathbb{R}^n \to \mathbb{R}^{n \times m}$ are measurable functions satisfying*

1. *there exists a constant $K > 0$ such that for all $t \geq 0$, $x, x' \in \mathbb{R}^n$*

$$\|\mu(t, x) - \mu(t, x')\| + \|\sigma(t, x) - \sigma(t, x')\| \leq k\|x - x'\|.$$

2. *the functions $\|\mu(., 0)\|$ and $\|\sigma(., 0)\|$ are $L_2(\mathbb{R}^+)$ with respect to the Lebesgue measure on $\mathbb{R}^+$.*

*Then, for all $T \geq 0$, there exists a unique strong solution of the SDE above.*

### B.2    Itô 's lemma

The following result, known as Itô 's lemma, is a classic result in stochastic calculus. We state a version of this result from Tankov & Touzi (2018). Other versions and extensions exist in the literature (e.g. Ingersoll (1987); Øksendal (2003); Kloeden & Platen (1995)).

**Lemma 4** (Itô 's lemma, Thm 6.7 in Tankov & Touzi (2018)). *Let $X_t$ be an Itô diffusion process (Definition 1) of the form*

$$dX_t = \mu_t dt + \sigma_t dB_t, t \in [0, T], X_0 \sim \nu$$

*where $\nu$ is some given distribution. Let $g : \mathbb{R}^+ \times \mathbb{R}^n \to \mathbb{R}$ be $\mathcal{C}^{1,2}([0, T], \mathbb{R}^n)$ (i.e. $\mathcal{C}^1$ in the first variable $t$ and $\mathcal{C}^2$ in the second variable $x$). Then, with probability $1$, we have that*

$$f(t, X_t) = f(0, X_0) + \int_0^t \nabla_x f(s, X_s) \cdot dX_s + \int_0^t \left( \partial_t f(s, X_s) + \frac{1}{2} \mathrm{Tr} \left[ \sigma_s^\top \nabla_x^2 f(s, X_s) \sigma_s \right] \right) ds,$$

*where $\nabla_x f$ and $\nabla_x^2 f$ refer to the gradient and the Hessian, respectively. This can also be expressed as an SDE*

$$df(t, X_t) = \nabla_x f(t, X_t) \cdot dX_t + \left( \partial_t f(t, X_t) + \frac{1}{2} \mathrm{Tr} \left[ \sigma_t^\top \nabla_x^2 f(t, X_t) \sigma_t \right] \right) dt.$$

### B.3    Convergence of Euler's scheme to the SDE solution

The following result gives a convergence rate of the Euler discretization scheme to the solution of the SDE.

**Theorem 5** ( Corollary of Thm 10.2.2 in Kloeden & Platen (1995)). *Let $d \geq 1$ and consider the $\mathbb{R}^d$-valued ito process $X$ (Definition 1) given by*

$$X_t = X_0 + \int_0^t \mu(s, X_s)ds + \int_0^t \sigma(s, X_s)dB_s,$$

*where $B$ is a $m$-dimensional Brownian motion for some $m \geq 1$, $X_0$ satisfies $\mathbb{E}\|X_0\|^2 < \infty$, and $\mu : \mathbb{R}^+ \times \mathbb{R}^d \to \mathbb{R}^d$ are $\sigma : \mathbb{R}^+ \times \mathbb{R}^d \to \mathbb{R}^{d \times m}$ are measurable functions satisfying the following conditions:*

1. *There exists a constant $K > 0$ such that for all $t \in \mathbb{R}, x, x' \in \mathbb{R}^d$,*

$$\|\mu(t, x) - \mu(t, x')\| + \|\sigma(t, x) - \sigma(t, x')\| \leq K\|x - x'\|.$$

2. *There exists a constant $K' > 0$ such that for all $t \in \mathbb{R}, x \in \mathbb{R}^d$*

$$\|\mu(t, x)\| + \|\sigma(t, x)\| \leq K'(1 + \|x\|).$$

3. *There exists a constant $K'' > 0$ such that for all $t, s \in \mathbb{R}, x \in \mathbb{R}^d$,*

$$\|\mu(t,x) - \mu(s,x)\| + \|\sigma(t,x) - \sigma(s,x)\| \leq K''(1 + \|x\|)|t - s|^{1/2}.$$

*Let $\delta \in (0,1)$ such that $\delta^{-1} \in \mathbb{N}$ (integer), and consider the times $t_k = k\delta$ for $k \in \{1, \ldots, \delta^{-1}\}$. Consider the Euler scheme given by*

$$Y^i_{k+1} = Y^i_k + \mu^i(t_k, Y^k_n)\delta + \sum_{j=1}^m \sigma^{i,j}(t_k, Y^k_n)\Delta B^j_k, \quad Y^i_0 = X^i_0,$$

*where $Y^i, \mu^i, \sigma^{i,j}$ denote the coordinates of these vectors for $i \in [d], j \in [m]$, and $\Delta B^j_k \sim \mathcal{N}(0, \delta)$. Then, we have that*

$$\mathbb{E} \sup_{t \in [0,1]} \|X_t - Y_{\lfloor t\delta^{-1}\rfloor}\|^2 = \mathcal{O}(\delta).$$

We can extend the result of Theorem 5 to the case of locally Lipschitz drift and volatility functions $\mu$ and $\sigma$. For this purpose, let us first define local convergence.

**Definition 2.** *Let $(X^L)_{L \geq 1}$ be a sequence of processes and $X$ be a stochastic process. For $r > 0$, define the following stopping times*

$$\tau^L = \{t \geq 0 : |X^L_t| \geq r\}, \tau = \{t \geq 0 : |X_t| \geq r\}.$$

*We say that $X^L$ converges locally to $X$ if for any $r > 0$, $X^L_{t \wedge \tau^L}$ converge to $X_{t \wedge \tau}$.*
*This definition is general for any type of convergence, we will specify clearly the type of convergence when we use this notion of local convergence.*

**Lemma 5** (Locally-Lipschitz coefficients)**.** *Consider the same setting of Theorem 5 with the following conditions instead*

1. *For any $r > 0$, there exists a constant $K > 0$ such that for all $t \in \mathbb{R}, x, x' \in \mathbb{R}^d$ with $\|x\|, \|x'\| \leq r$,*

$$\|\mu(t,x) - \mu(t,x')\| + \|\sigma(t,x) - \sigma(t,x')\| \leq K\|x - x'\|.$$

2. *For any $r > 0$, there exists a constant $K' > 0$ such that for all $t \in \mathbb{R}, x \in \mathbb{R}^d$ satisfying $\|x\| \leq r$*

$$\|\mu(t,x)\| + \|\sigma(t,x)\| \leq K'(1 + \|x\|).$$

3. *For any $r > 0$, there exists a constant $K'' > 0$ such that for all $t, s \in \mathbb{R}, x \in \mathbb{R}^d$ satisfying $\|x\| \leq r$,*

$$\|\mu(t,x) - \mu(s,x)\| + \|\sigma(t,x) - \sigma(s,x)\| \leq K''(1 + \|x\|)|t - s|^{1/2}.$$

*Then, for any $r > 0$, we have that*

$$\mathbb{E} \sup_{t \in [0,1]} \|X_{t \wedge \tau} - Y_{\lfloor (t \wedge \tau_\delta)\delta^{-1}\rfloor}\|^2 = \mathcal{O}(\delta),$$

*where $\tau_\delta = \inf\{t \geq 0 : \|Y_{\lfloor t\delta^{-1}\rfloor}\| > r\}$, and $\tau = \inf\{t \geq 0 : \|X_t\| > r\}$.*

We omit the proof here as it consists of the same techniques used in Kloeden & Platen (1995), with the only difference consisting of considering the stopped process $X^\tau$. By stopping the process, we force the process to stay in a region where the coefficients are Lipschitz.

## B.4 Convergence of Particles to the solution of Mckean-Vlasov process

The next result gives sufficient conditions for the system of particles to converge to its mean-field limit, known as the Mckean-Vlasov process.

**Theorem 6** ( Mckean-Vlasov process, Corollary of Thm 3 in Jourdain et al. (2007))**.** *et $d \geq 1$ and consider the $\mathbb{R}^d$-valued ito process $X$ (Definition 1) given by*

$$dX_t = \sigma(X_t, \nu_t^n)dB_t, \quad X_0 \text{ has iid components,}$$

*where $B$ is a d-dimensional Brownian motion, $\nu_t^n \stackrel{def}{=} \frac{1}{d}\sum_{i=1}^{d}\delta_{\{X_t^i\}}$ the empirical distribution of the coordinates of $X_t$, and $\sigma$ is real-valued given by $\sigma(x,\nu) = \int \zeta(x,y)d\nu(y)$ for all $x \in \mathbb{R}^d$ and distribution $\nu$. Assume that the function $\zeta$ is Lipschitz continuous. Then, we have that for all $T \in \mathbb{R}^+$,*

$$\sup_{i \in [n]} \mathbb{E}\left(\sup_{t \leq T}|X_t^i - \tilde{X}_t^i|^2\right) = \mathcal{O}(n^{-1}),$$

*where $\tilde{X}^i$ is the solution of the following Mckean-Vlasov equation*

$$d\tilde{X}_t^i = \sigma(\tilde{X}_t^i, \nu_t^i)dB_t^i, \quad \tilde{X}_0^i = X_0^i,$$

*where $\nu_t^i$ is the distribution of $\tilde{X}^i$.*

### B.5 Other results from probability and stochastic calculus

The next trivial lemma has been opportunely used in Li et al. (2021) to derive the limiting distribution of the network output (multi-layer perceptron) in the joint infinite width-depth limit. This simple result will also prove useful in our case of the finite-width-infinite-depth limit.

**Lemma 6.** *Let $W \in \mathbb{R}^{n \times n}$ be a matrix of standard Gaussian random variables $W_{ij} \sim \mathcal{N}(0,1)$. Let $v \in \mathbb{R}^n$ be a random vector independent from $W$ and satisfies $\|v\|_2 = 1$ . Then, $Wv \sim \mathcal{N}(0,I)$.*

*Proof.* The proof follows a simple characteristic function argument. Indeed, by conditioning on $v$, we observe that $Wv \sim \mathcal{N}(0,I)$. Let $u \in \mathbb{R}^n$, we have that

$$\mathbb{E}_{W,v}[e^{i\langle u, Wv\rangle}] = \mathbb{E}_v[\mathbb{E}_W[e^{i\langle u, Wv\rangle}|v]]$$
$$= \mathbb{E}_v[e^{-\frac{\|u\|^2}{2}}]$$
$$= e^{-\frac{\|u\|^2}{2}}.$$

This concludes the proof as the latter is the characteristic function of a random Gaussian vector with Identity covariance matrix. □

The next theorem shows when a stochastic process (ito)

**Theorem 7** (Variation of Thm 8.4.3 in Øksendal (2003))**.** *Let $(X_t)_{t \in [0,T]}$ and $(Y_t)_{t \in [0,T]}$ be two stochastic processes given by*

$$\begin{cases} dX_t = b(X_t)dt + \sigma(X_t)dB_t, \quad X_0 = x \in \mathbb{R}. \\ dY_t = b_t dt + v_t d\hat{B}_t, \quad Y_0 = X_0, \end{cases}$$

*where $\sigma : \mathbb{R} \to \mathbb{R}^{1 \times k}$, $(b_t)_{t \geq 0}$ and $(v_t)_{t \geq 0}$ are real valued adapted stochastic processes, and $v$ is adapted to the filtration of the Brownian motion $(\hat{B}_t)_{t \geq 0}$, $(B_t)_{t \geq 0}$ is an k-dimensional Brownian motion and $(\hat{B}_t)_{t \geq 0}$ is a 1-dimensional Brownian motion. Assume that $\mathbb{E}[b_t | \mathcal{N}_t] = b(Y_t)$ where $\mathcal{N}_t = \sigma((Y_s)_{s \leq t})$ is the $\sigma$-Algebra generated by $\{Y_s : s \leq t\}$, and $v_t^2 = \sigma(Y_t)\sigma(Y_t)^\top$ almost surely (in terms of $dt \times dP$ measure where $dt$ is the natural Borel measure on $[0,T]$ and $dP$ is the probability measure associated with the probability space). Then, $X_t$ and $Y_t$ have the same distribution for all $t \in [0,T]$.*

*Proof.* The proof of this theorem is the same as that of Thm 8.4.3 in Øksendal (2003) with small differences. Indeed, our result is slightly different from that of Øksendal (2003) in the sense that here we consider Brownian motions with different dimensions, while in their theorem, the author considers the case where the

Brownian motions involved in $(X_t)$ and $(Y_t)$ are of the same dimension. However, both results make use of the so-called Martingale problem, which characterizes the weak uniqueness and hence the distribution of Ito processes[8]. The generator of $X_t$ is given for $f \in \mathcal{C}^2(\mathbb{R})$ by

$$\mathcal{G}(f)(x) = b(x)\frac{\partial f}{\partial x} + \frac{1}{2}\sigma(x)\sigma(x)^\top \frac{\partial^2 f}{\partial x^2}.$$

Now define the process $\mathcal{H}(f)$ for $f \in \mathcal{C}^2(\mathbb{R})$ by

$$\mathcal{H}(f)(t) = b_t \frac{\partial f}{\partial x}(Y_t) + \frac{1}{2}v_t^2 \frac{\partial^2 f}{\partial x^2}(Y_t).$$

Let $\mathcal{N}_t = \sigma((Y_s)_{s \le t})$ be the $\sigma$-Algebra generated by $\{Y_s : s \le t\}$. Using Itô lemma, we have that for $s < t$,

$$\mathbb{E}[f(Y_s)|\mathcal{N}_t] = f(Y_t) + \mathbb{E}[\int_t^s \mathcal{H}(f)(r)dr|\mathcal{N}_t]$$
$$= f(Y_t) + \mathbb{E}\left[\int_t^s \mathbb{E}[\mathcal{H}(f)(r)|\mathcal{N}_r]dr|\mathcal{N}_t\right]$$
$$= f(Y_t) + \mathbb{E}\left[\int_t^s \mathcal{G}(f)(Y_r)dr|\mathcal{N}_t\right],$$

where we have used the fact that $\mathbb{E}[b_r|\mathcal{N}_r] = b(Y_r)$. Now define the process $M$ by

$$M_t = f(Y_t) - \int_0^t \mathcal{G}(f)(Y_r)dr.$$

For $s > t$, we have that

$$\mathbb{E}[M_s|\mathcal{N}_t] = f(Y_t) + \mathbb{E}\left[\int_t^s \mathcal{G}(f)(Y_r)dr|\mathcal{N}_t\right] - \mathbb{E}\left[\int_0^s \mathcal{G}(f)(Y_r)dr|\mathcal{N}_t\right] \quad \text{(by Itô lemma)},$$
$$= f(Y_t) - \mathbb{E}\left[\int_0^t \mathcal{G}(f)(Y_r)dr|\mathcal{N}_t\right] = M_t.$$

Hence, $M_t$ is a martingale (w.r.t to $\mathcal{N}_t$). We conclude that $Y_t$ has the same law as $X_t$ by the uniqueness of the solution of the martingale problem (see 8.3.6 in Øksendal (2003)). $\square$

The next result is a simple corollary of the existence and uniqueness of the strong solution of an SDE under the Lipschitz conditions on the drift and the volatility. It basically shows that a zero-drift process collapses (becomes constant) once the volatility is zero.

**Lemma 7.** *Let $g : \mathbb{R}^n \to \mathbb{R}$ be a Lipschitz function. Let $Z$ be the solution of the stochastic differential equation*

$$dZ_t = g(Z_t)dB_t, \quad Z_0 \in \mathbb{R}^n.$$

*If $g(Z_0) = 0$, then $Z_t = Z_0$ almost surely.*

*Proof.* This follows for the uniqueness of the strong solution of an SDE(Theorem 4). $\square$

---

[8]We omit the details on the Martingale problem here. We invite the curious reader to check Chapter 8 in Øksendal (2003) for further details.

### B.6 Proof of Proposition 1

We are now ready to prove the following result.

**Proposition 1.** *Assume that the activation function $\phi$ is Lipschitz on $\mathbb{R}^n$. Then, in the limit $L \to \infty$, the process $X_t^L = Y_{\lfloor tL \rfloor}$, $t \in [0,1]$, converges in distribution to the solution of the following SDE*

$$dX_t = \frac{1}{\sqrt{n}} \|\phi(X_t)\| dB_t, \quad X_0 = W_{in}x, \tag{7}$$

*where $(B_t)_{t \geq 0}$ is a Brownian motion (Wiener process). Moreover, we have that for any $t \in [0,1]$ Lipschitz function $\Psi : \mathbb{R}^n \to \mathbb{R}$,*

$$\mathbb{E}\Psi(Y_{\lfloor tL \rfloor}) = \mathbb{E}\Psi(X_t) + \mathcal{O}(L^{-1/2}),$$

*where the constant in $\mathcal{O}$ does not depend on $t$.*
*Moreover, if the activation function $\phi$ is only locally Lipschitz, then $X_t^L$ converges locally to $X_t$. More precisely, for any fixed $r > 0$, we consider the stopping times*

$$\tau^L = \inf\{t \geq 0 : |X_t^L| \geq r\}, \quad \tau = \inf\{t \geq 0 : |X_t| \geq r\},$$

*then the stopped process $X_{t \wedge \tau^L}^L$ converges in distribution to the stopped solution $X_{t \wedge \tau}$ of the above SDE.*

*Proof.* The proof is based on Theorem 5 in the appendix. It remains to express Eq. (1) in the required form and make sure all the conditions are satisfied for the result to hold. Using Lemma 6, we can write Eq. (1) as

$$Y_l = Y_{l-1} + \frac{1}{\sqrt{L}} \sigma(Y_{l-1}) \zeta_{l-1}^L,$$

where $\sigma(y) \overset{def}{=} \frac{1}{\sqrt{n}} \|\phi(y)\|$ for all $y \in \mathbb{R}^n$ and $\zeta_l^L$ are *iid* random Gaussian vectors with distribution $\mathcal{N}(0, I)$. This is equal in distribution to the Euler scheme of SDE Eq. (8). Since $\sigma$ trivially inherits the Lipschitz or local Lipschitz properties of $\phi$, we conclude for the convergence using Theorem 5 and Lemma 5.

Now let $\Psi$ be $K$-Lipschitz for some constant $K > 0$. We have that

$$|\mathbb{E}\Psi(Y_{\lfloor tL \rfloor}) - \mathbb{E}\Psi(X_t)| \leq K \mathbb{E} \sup_{t \in [0,1]} \|\bar{Y}_{\lfloor tL \rfloor} - X_t\| = \mathcal{O}(L^{-1/2}),$$

where $\bar{Y}$ is the Euler scheme as in Theorem 5, and where we have used the fact that $Y_{\lfloor tL \rfloor}$ and $\bar{Y}_{\lfloor tL \rfloor}$ have the same distribution. $\qquad\square$

The result of Proposition 1 can be generalized to the case with multiple inputs with minimal changes in the proof. We summarize this result in the next proposition.

**Proposition 5.** *Let $x_1, x_2, \ldots, x_k \in \mathbb{R}^d$ be non-zero inputs, and denote by $Y_l(x_i)$ the pre-activation vector in layer $l$ for the input $x_i$. Consider the vector $\boldsymbol{Y}_l^k = (Y_l(x_1)^\top, Y_l(x_2)^\top, \ldots, Y_l(x_k)^\top)^\top \in \mathbb{R}^{k \cdot n}$ consisting of the concatenation of the pre-activations vectors for all inputs $x_i$. Assume that the activation function $\phi$ is Lipschitz on $\mathbb{R}^n$. Then, in the limit $L \to \infty$, the process $\boldsymbol{X}_t^{L,k} = \boldsymbol{Y}_{\lfloor tL \rfloor}^k$, $t \in [0,1]$, converges in distribution to the solution of the following SDE*

$$d\boldsymbol{X}_t^k = \frac{1}{\sqrt{n}} \Sigma(\boldsymbol{X}_t^k)^{1/2} d\boldsymbol{B}_t, \quad \boldsymbol{X}_0^k = ((W_{in}x_1)^\top, \ldots, (W_{in}x_k)^\top)^\top, \tag{8}$$

*where $(\boldsymbol{B}_t)_{t \geq 0}$ is an $kn$-dimensional Brownian motion (Wiener process), independent from $W_{in}$, and $\Sigma(\boldsymbol{X}_t^k)$ is the covariance matrix given by*

$$\Sigma(\boldsymbol{X}_t^k) = \begin{bmatrix} \alpha_{1,1}I_n & \alpha_{1,2}I_n & \ldots & \alpha_{1,k}I_n \\ \alpha_{2,1}I_n & \alpha_{2,2}I_n & \ldots & \alpha_{2,k}I_n \\ \vdots & \vdots & \vdots & \vdots \\ \alpha_{k,1}I_n & \ldots & \ldots & \alpha_{k,k}I_n \end{bmatrix},$$

where $\alpha_{i,j} = \langle \phi(\boldsymbol{X}_t^{k,i}), \phi(\boldsymbol{X}_t^{k,j}) \rangle$, with $(X_t^{k,1\top}, \dots, X_t^{k,k\top})^\top \overset{def}{=} \boldsymbol{X}_t^k$. Moreover, if the activation function $\phi$ is only locally Lipschitz, then $\boldsymbol{X}_t^{L,k}$ converges locally to $\boldsymbol{X}_t^k$. More precisely, for any fixed $r > 0$, we consider the stopping times $\tau^L = \inf\{t \geq 0 : |\boldsymbol{X}_t^{L,k}| \geq r\}$, and $\quad \tau = \inf\{t \geq 0 : |\boldsymbol{X}_t^k| \geq r\}$, then the stopped process $\boldsymbol{X}_{t \wedge \tau^L}^{L,k}$ converges in distribution to the stopped solution $\boldsymbol{X}_{t \wedge \tau}^k$ of the above SDE.

*Proof.* The proof is similar to that of Proposition 1. The only difference lies the definition of the Gaussian vector $\zeta_l^L$. In this case, we have for all $x_i$

$$Y_l(x_i) = Y_{l-1}(x_i) + \frac{1}{\sqrt{L}} \frac{1}{\sqrt{n}} \zeta_{l-1}^L(Y_{l-1}(x_i)),$$

where $\zeta_{l-1}^L(Y_{l-1}(x_i)) \overset{def}{=} \sqrt{n} W_l \phi(Y_{l-1}(x_i))$. Concatenating these identities yield

$$\boldsymbol{Y}_l^k = \boldsymbol{Y}_{l-1}^k + \frac{1}{\sqrt{L}} \frac{1}{\sqrt{n}} \boldsymbol{\zeta}_{l-1}^L,$$

where $\boldsymbol{\zeta}_{l-1}^L$ is the concatenation of the vector $\zeta_{l-1}^L(Y_{l-1}(x_i))$ for $i = 1, \dots, k$. It is straightforward that the covariance matrix of the Gaussian vector $\boldsymbol{\zeta}_{l-1}^L$ is given by the matrix $\Sigma$ above (with $X$ replaced by $Y$). We conclude using Theorem 5.

$\square$

## C  Some technical results for the proofs

### C.1  Approximation of $X$

In the next lemma, we provide an approximate stochastic process $X^m$ to $X$, that differs from $X$ by the volatility term. The upper-bound on the $L_2$ norm of the difference between $X^m$ and $X$ will prove useful in the proofs of other results. The proof of this lemma requires the use of Gronwall's lemma, a tool that is often used in stochastic calculus.

**Lemma 8.** *Let $x \in \mathbb{R}^d$ such that $x \neq 0$, $m \geq 1$ be an integer, and consider the two stochastic processes $X^m$ and $X$ given by*

$$\begin{cases} dX_t^m = \frac{1}{\sqrt{n}} \|\phi_m(X_t^m)\| dB_t, & t \in [0, \infty), \quad X_0^m = W_{in}x, \\ dX_t = \frac{1}{\sqrt{n}} \|\phi(X_t)\| dB_t, & t \in [0, \infty), \quad X_0 = W_{in}x, \end{cases}$$

*where $\phi_m(z) = \int_0^z h(mu) du$ where $h$ is the Sigmoid function given by $h(u) = (1 + e^{-u})^{-1}$, $\phi$ is the ReLU activation function, and $(B_t)_{t \geq 0}$ is an $n$-dimensional Brownian motion. We have the following*

$$\forall t \geq 0, \ \mathbb{E} \|X_t^m - X_t\|^2 \leq \frac{2nt}{m^2} e^{2t}.$$

*Proof.* Let $t \geq 0$. We have that

$$\mathbb{E} \|X_t^m - X_t\|^2 = \frac{1}{n} \mathbb{E} \left\| \int_0^t (\|\phi_m(X_s^m)\| - \|\phi(X_s)\|) dB_s. \right\|^2$$

Using Itô isometry and the fact that $(\|\phi_m(X_s^m)\| - \|\phi(X_s)\|)^2 \leq \|\phi_m(X_s^m) - \phi(X_s)\|^2$, we obtain

$$\mathbb{E} \|X_t^m - X_t\|^2 \leq \int_0^t \mathbb{E} \|\phi_m(X_s^m) - \phi(X_s)\|^2 ds$$

$$\leq 2 \int_0^t \mathbb{E} \|\phi_m(X_s^m) - \phi(X_s^m)\|^2 ds + 2 \int_0^t \mathbb{E} \|\phi(X_s^m) - \phi(X_s)\|^2 ds$$

$$\leq \frac{2nt}{m^2} + 2 \int_0^t \mathbb{E} \|X_s^m - X_s\|^2 ds,$$

where we have used Lemma 9 and the fact that ReLU is 1-Lipschitz. We concldue using Gronwall's lemma.

$\square$

## C.2 Approximation of $\phi$

The next lemma provides a simple upper-bound on the distance between the ReLU activation $\phi$ and an approximate function $\phi_m$ that converges to $\phi$ in the limit of large $m$.

**Lemma 9.** *Consider the function* $\phi_m(z) = \int_0^z h(mu)du$ *where* $z \in \mathbb{R}$ *where* $m \geq 1$. *We have that*

$$\sup_{z \in \mathbb{R}} |\phi_m(z) - \phi(z)| \leq \frac{1}{m}.$$

*Proof.* Let $m \geq 1$ and $z \in \mathbb{R}$. Assume that $z > 0$. We have that

$$
\begin{aligned}
|\phi_m(z) - \phi(z)| &= \int_0^z \frac{e^{-mu}}{1 + e^{-mu}} du \\
&\leq \int_0^z e^{-mu} du \\
&= \frac{1}{m}(1 - e^{-mz}) \leq \frac{1}{m}.
\end{aligned}
$$

For the case where $z \leq 0$, the proof is the same. We have that

$$
\begin{aligned}
|\phi_m(z) - \phi(z)| &= \int_0^z \frac{e^{mu}}{1 + e^{mu}} du \\
&\leq \int_0^z e^{mu} du \\
&= \frac{1}{m}(1 - e^{mz}) \leq \frac{1}{m},
\end{aligned}
$$

which concludes the proof. $\square$

## C.3 Other lemmas

The next lemma shows that the logarithmic growth factor $\log\left(\frac{\|\phi_m(X_t^m)\|}{\|\phi_m(X_0^m)\|}\right)$ converges to $\log\left(\frac{\|\phi(X_t)\|}{\|\phi(X_0)\|}\right)$ when $m$ goes to infinity, where the convergence holds in $L_1$. The key ingredient in the use of uniform integrability coupled with convergence in probability, which is sufficient to conclude on the $L_1$ convergence. This result will help us conclude in the proof of Theorem 1.

**Lemma 10.** *Let* $x \in \mathbb{R}^d$ *such that* $x \neq 0$, $m \geq 1$ *be an integer, and consider the two stochastic processes* $X^m$ *and* $X$ *given by*

$$
\begin{cases}
dX_t^m = \frac{1}{\sqrt{n}}\|\phi_m(X_t^m)\|dB_t, & t \in [0, \infty), \quad X_0^m = W_{in}x, \\
dX_t = \frac{1}{\sqrt{n}}\|\phi(X_t)\|dB_t, & t \in [0, \infty), \quad X_0 = W_{in}x,
\end{cases}
$$

*where* $\phi_m(z) = \int_0^z h(mu)du$ *where* $h$ *is the Sigmoid function given by* $h(u) = (1 + e^{-u})^{-1}$, $\phi$ *is the ReLU activation function, and* $(B_t)_{t \geq 0}$ *is an* $n$-*dimensional Brownian motion. Then, conditionally on the fact that* $\|\phi(X_0)\| > 0$, *we have that*

$$\forall t \geq 0, \ \log\left(\frac{\|\phi_m(X_t^m)\|}{\|\phi_m(X_0^m)\|}\right) \xrightarrow{L^1} \log\left(\frac{\|\phi(X_t)\|}{\|\phi(X_0)\|}\right).$$

*Proof.* Let $t > 0$. From Lemma 8, we know that $X^m$ converges in $L^2$ to $X$. Using Lemma 9 and the fact that ReLU is 1-Lipschitz, we obtain

$$\mathbb{E}\|\phi_m(X_t^m) - \phi(X_t)\|^2 \leq \frac{2n}{m^2} + 2\mathbb{E}\|X_t^m - X_t\|^2,$$

which implies that $\phi_m(X_t^m)$ converges in $L^2$ to $\phi(X_t)$. In particular, the convergence holds in probability. Using this fact with the Continuous mapping theorem, we obtain that

$$\forall t \geq 0, \ \log\left(\|\phi_m(X_t^m)\|\right) \xrightarrow{\mathbb{P}} \log\left(\|\phi(X_t)\|\right). \tag{9}$$

Let us show the following,

$$\forall t \geq 0, \ \log\left(\frac{\|\phi_m(X_t^m)\|}{\|\phi_m(X_0^m)\|}\right) \xrightarrow{\mathbb{P}} \log\left(\frac{\|\phi(X_t)\|}{\|\phi(X_0)\|}\right).$$

Let $\epsilon > 0$ and $t > 0$. We have

$$\mathbb{P}\left(\left|\log\left(\frac{\|\phi_m(X_t^m)\|}{\|\phi_m(X_0^m)\|}\right) - \log\left(\frac{\|\phi(X_t)\|}{\|\phi(X_0)\|}\right)\right| \geq \epsilon\right) \leq \mathbb{P}\left(|\log\|\phi_m(X_t^m)\| - \log\left(\|\phi(X_t)\|\right)| \geq \epsilon/2\right)$$
$$+ \mathbb{P}\left(|\log\|\phi_m(X_0^m)\| - \log\left(\|\phi(X_0)\|\right)| \geq \epsilon/2\right),$$

where the first term converges to zero by Eq. (9), and the second term converges to zero by Lemma 9. Hence, the convergence in probability holds.

To conclude, it suffices to show that the sequence of random variables $\left(Y_t^m = \log\left(\frac{\|\phi_m(X_t^m)\|}{\|\phi_m(X_0^m)\|}\right)\right)_{m \geq 1}$ is uniformly integrable.

Let $K > 0$. From the proof of Lemma 2, with $\zeta = \phi_m$, we have that

$$Y_t^m = \frac{1}{\sqrt{n}}\int_0^t \mu(X_s^m)ds + \frac{1}{2n}\int_0^t \sum_{i=1}^n \sigma_i(X_s^m)dB_s^i,$$

where $\sigma_i(X_s^m) = \frac{|\phi_m'(X_s^{m,i})\phi_m(X_s^{m,i})|}{\|\phi_m(X_s^m)\|}$, and $\mu(X_s^m) = \frac{1}{2}\sum_{i=1}^n \left(\phi_m''(X_s^{m,i})\phi_m(X_s^{m,i}) + \phi_m'(X_s^{m,i})^2\right) - \frac{\|\phi_m'(X_s^m)\circ\phi_m(X_s^m)\|^2}{\|\phi_m(X_s^m)\|^2}$. Therefore,

$$\mathbb{E}|Y_t^m|^2 = \frac{1}{n}\mathbb{E}\left(\int_0^t \mu(X_s^m)ds\right)^2 + \frac{1}{4n^2}\mathbb{E}\int_0^t \sum_{i=1}^m \sigma_i(X_s^m)^2 ds$$
$$\leq \frac{t}{n}\int_0^t \mathbb{E}\mu(X_s^m)^2 ds + \frac{1}{4n^2}\mathbb{E}\int_0^t \sum_{i=1}^m \sigma_i(X_s^m)^2 ds, \tag{10}$$

where we have used the Itô isometry and Cauchy-Schwartz inequality. Using the conditions on $\phi_m$, it is straightforward that term $\frac{1}{4n^2}\mathbb{E}\int_0^t \sum_{i=1}^m \sigma_i(X_s^m)^2 ds$ is uniformly bounded. It remains to bound the first term. Similarly to the proof of Theorem 1, we condition on the regions of $|X_s^{m,i}|$ and obtain that the terms $\mathbb{E}\mu(X_s^m)^2$ are uniformly bounded over $m$ (we omit the proof here as it is just a repetition of the techniques used in the proof of Theorem 1). Therefore, we have that $\sup_{m \geq 1}\mathbb{E}|Y_t^m|^2 < \infty$, which implies uniform integrability. This concludes the proof. $\qquad\square$

**Lemma 11.** *Let $x \in \mathbb{R}^d$ such that $x \neq 0$, $m \geq 1$ be an integer, and consider the stochastic processes $X$ given by*

$$dX_t = \frac{1}{\sqrt{n}}\|\phi(X_t)\|dB_t, \quad t \in [0, \infty), \quad X_0 = W_{in}x,$$

where $\phi$ is the ReLU activation function, and $(B_t)_{t\geq 0}$ is an $n$-dimensional Brownian motion independent from $X_0$. Then, conditionally on the fact that $\|\phi(X_0)\| > 0$, we have that for all $s \in [0,1], i \in [n]$

$$\mathbb{P}(|X_s^i| \leq \delta) = \mathcal{O}_{\delta \to 0}(\delta),$$

where the bound holds uniformly over $s \in [0,1]$.

*Proof.* We have that

$$X_s^i = X_0^i + \frac{1}{\sqrt{n}} \int_0^s \|\phi(X_u)\| dB_u^i.$$

Since $\phi(X_u) > 0$ for all $u \geq 0$ almost surely, and by the independence of $B$ and $X_0$, we can easily see that $X_s^i$ has no Dirac mass and is the sum of two continuous random variables (not independent) $X_0^i$ and $\frac{1}{\sqrt{n}} \int_0^s \|\phi(X_u)\| dB_u^i$ that have bounded density functions, and thus $X_s^i$ has a bounded density function $h_s$. Hence, writing $\mathbb{P}(|X_s^i| \leq \delta) = \int_{-\delta}^{\delta} h_s(t) dt = \mathcal{O}(\delta)$ concludes the proof. The bound can be taken uniformly over $s \in [0,1]$ by taking $\sup_{s \in [0,1]} |h_s(0)|$. □

## D  The Ornstein-Uhlenbeck (OU) process

The OU process is the (unique) strong solution to the following diffusion

$$dX_t = a(b - X_t)dt + \sigma dB_t, \tag{11}$$

where $a, b, \sigma \in \mathbb{R}$ are constants, and $B$ is a one dimensional Brownian motion. In financial mathematics, the OU process is used as a model of short-term interest rate under the name of the Vasicek model. The OU process has a closed-form expression and its marginal distribution is Gaussian. The next lemma gives a full characterization of the marginal distributions of an OU process.

**Lemma 12.** *Eq.* (11) *admits the following solution*

$$X_t = X_0 e^{-at} + b(1 - e^{-at}) + \sigma \int_0^t e^{-a(t-s)} dB_s.$$

*As a result, we have the following*

- $X_t$ *is Gaussian.*

- $\mathbb{E}[X_t] = X_0 e^{-at} + b(1 - e^{-at})$.

- $\mathrm{Cov}(X_t, X_s) = \frac{\sigma^2}{2a} \left( e^{-a|t-s|} - e^{-a(t+s)} \right)$.

*Proof.* Consider the process $Z_t = e^{at} X_t$, using Itô lemma, we have that

$$dZ_t = aZ_t dt + e^{at} dX_t$$
$$= abe^{at} dt + \sigma e^{at} dW_t.$$

Integrating between $0$ and $t$ yields

$$Z_t = Z_0 + b(e^{at} - 1) + \sigma \int_0^t e^{as} dW_s.$$

We conclude by multiplying both sides with $e^{-at}$.

The result for $\mathbb{E}[X_t]$ is straightforward since $\mathbb{E}\left[\int_0^t e^{-a(t-s)}dW_s\right] = 0$ by the properties of Itô integral and the Brownian motion. For the covariance, without loss of generality assume that $t > s \geq 0$. We have that

$$
\begin{aligned}
\mathrm{Cov}(X_t, X_s) &= \sigma^2 \mathbb{E}\left[\int_0^t e^{-a(t-u)}dW_u \int_0^s e^{-a(s-u)}dW_u\right] \\
&= \sigma^2 \mathbb{E}\left[\int_0^s e^{-a(t-u)}dW_u \int_0^s e^{-a(s-u)}dW_u\right] \\
&= \sigma^2 \int_0^s e^{-2a(\frac{s+t}{2}-u)}du \\
&= \frac{\sigma^2}{2a}\left(e^{-a(t-s)} - e^{-a(t+s)}\right),
\end{aligned}
$$

which completes the proof.

$\square$

We would like to find sufficient conditions on the activation function $\phi$ and a function $g$ such that the process $g(X_t)$ (Eq. (8)) follows an the OU dynamics. For this purpose, we proceed by reverse-engineering the problem; Using Itô 's lemma (Eq. (4)), this is satisfied when there exist constants $a, b, \sigma$ such that

$$
\begin{cases}
\mu(y) = \frac{1}{2n}\phi(y)^2 g''(y) = a(b - g(y)) \\
\sigma(y) = \frac{1}{\sqrt{n}}\phi(y)g'(y) = \sigma.
\end{cases}
$$

This implies that $\frac{g''(y)}{g'^2(y)} = 2a\sigma^{-2}(b - g(y))$. Letting $G = \int g$ be the primitive function of $g$, we obtain that $G$ satisfies a differential equation of the form

$$
\frac{1}{G''(y)} = \alpha y + \beta G(y) + \zeta,
$$

where $\alpha, \beta, \zeta \in \mathbb{R}$ are constants.

Let us consider the case where $\alpha = \zeta = 0$ and $\beta \neq 0$, i.e. $\frac{1}{G''(y)} = \beta G(y)$. Equivalently, we solve the differential equation $G''(y) = \frac{\beta}{G}$ where $\beta \in \mathbb{R}$. Multiplying both sides by $G'$ and integrating we obtain $\frac{1}{2}G'(y)^2 = \beta \log(|G|) + \gamma$. A sufficient condition for this to hold is to have $G > 0$ and $G$ satisfies

$$
\frac{G'(y)}{\sqrt{\log(G) + \gamma}} = \zeta
$$

for some constants $\zeta, \gamma$. Integrating the left-hand side yields

$$
\int^y \frac{G'(u)}{\sqrt{\log(G(u)) + \gamma}}du = \int^{G(y)} \frac{1}{\sqrt{\log(u) + \gamma}}du = \alpha \, \mathrm{Erfi}(\sqrt{\log(G(y)) + \gamma}) + \beta.
$$

where Erfi is the imaginary error function[9] given by

$$
\mathrm{Erfi}(z) = \frac{2}{\sqrt{\pi}}\int_0^z e^{t^2}dt.
$$

To alleviate the notation, we denote $h := \mathrm{Erfi}$ in the rest of this section. From the above, $G$ should have the form

$$
G(y) = \exp\left(\zeta + \left(h^{-1}(\alpha y + \beta)\right)^2\right),
$$

where $\alpha, \beta, \zeta$ are all constants, and $h^{-1}$ is the inverse function of the imaginary error function. We conclude that the activation function $\phi$ should have the form

$$
\phi(y) = \frac{2\sigma}{\alpha^2 \pi}\exp(-\zeta + h^{-1}(\alpha y + \beta)^2).
$$

---

[9]Although the name might be misleading, the imaginary error function is real when the input is real.

In this case, the coefficients $a$ and $b$ are given by

$$b = 0, a = \frac{\sigma^2}{\alpha^2 \pi} \exp(-2\zeta).$$

Letting $g = G'$, the process $g(X_t)$ has the following dynamics

$$dg(X_t) = -ag(X_t)dt + \sigma dB_t,$$

Hence $g(X_t)$ is an OU process, and we can conclude that the network output in the infinite-depth limit $X_1$ satisfies

$$g(X_1) \sim \mathcal{N}\left(g(X_0)e^{-a}, \frac{\sigma^2}{2a}\left(1 - e^{-2a}\right)\right).$$

We can then infer the distribution of $X_1$ by a simple change of variable. Note that this distribution is non-trivial, and unlike the infinite-width limit of the same ResNet (Hayou et al. (2021)) where the distribution is Gaussian, here the distribution of the pre-activations is directly impacted by the choice of the activation function $\phi$.

However, with this particular choice of the activation function $\phi$, the existence of the process $X$ can only be proven in the local sense, because $\phi$ is only locally Lipschitz. Let us first show this in the next lemma. We will see how we can mitigate this issue later.

**Lemma 13.** *Let $\phi : \mathbb{R} \to \mathbb{R}$ defined by*

$$\phi(y) = \exp(h^{-1}(\alpha y + \beta)^2),$$

*where $\alpha, \beta \in \mathbb{R}$ are two constants.*

*We have that $\phi$ is locally Lipschitz, meaning that for any compact set $K \subset \mathbb{R}$, there exists $C_K$ such that*

$$\forall x, x' \in K, |\phi(x') - \phi(x)| \leq C_K |x' - x|.$$

*Proof.* It suffices to show that the derivative of $\phi$ is locally bounded to conclude. We have that

$$\phi'(y) = 2\alpha(h^{-1})'(\alpha y + \beta)h^{-1}(\alpha y + \beta)\exp(h^{-1}(\alpha y + \beta)^2)$$
$$= \alpha\sqrt{\pi}\, h^{-1}(\alpha y + \beta).$$

Since $h^{-1}$ is continuous on $\mathbb{R}$, then $\phi'$ is bounded on any compact set of $\mathbb{R}$, which concludes the proof. $\square$

Now we can rigorously prove the following result.

**Proposition 6.** *Let $x \in \mathbb{R}$ such that $x \neq 0$. Consider the following activation function $\phi$*

$$\phi(y) = \frac{2\sigma}{\alpha^2 \pi} \exp(-\zeta + h^{-1}(\alpha y + \beta)^2),$$

*where $\alpha, \beta \in \mathbb{R}$ and $\sigma, \zeta > 0$ are constants. Let $g$ be the function defined by*

$$g(y) = \alpha\sqrt{\pi}\exp(\zeta)h^{-1}(\alpha y + \beta).$$

*Consider the stochastic process $X_t$ defined by*

$$dX_t = |\phi(X_t)|dB_t, \quad X_0 = W_{in}x.$$

*Then, we have that for all $t \in [0, 1]$,*

$$g(X_t) \sim \mathcal{N}\left(g(X_0)e^{-at}, \frac{\sigma^2}{2a}(1 - e^{-2at})\right),$$

*where $a = \frac{\sigma^2}{\alpha^2 \pi} \exp(-2\zeta)$.*

*Proof.* For $N > 0$, consider the stopping time $\tau_N$ defined by

$$\tau_N = \inf\{t \geq 0 : |X_t| \geq N\}.$$

Using the continuity of paths of $X$, it is straightforward that $\lim_{N \to \infty} \tau_N = \infty$ almost surely. Let $N > 0$ be large enough. The SDE satisfied by the process $X$ has a unique strong solution for $t \in [0, \tau_N)$ since the activation function $\phi$ is Lipschitz on the interval $(-N, N)$. By applying Itô lemma for $t \in (0, \tau_N)$, we have that

$$dg(X_t) = -ag(X_t)dt + \sigma dB_t,$$

(from previous results). Using the fact that $\lim_{N \to \infty} \tau_N = \infty$ almost surely, and taking $N$ large enough, we obtain that for all $t \in (0, 1]$, we have that

$$dg(X_t) = -ag(X_t)dt + \sigma dB_t,$$

we conclude using Lemma 12.

$\square$

## E   The Geometric Brownian Motion (GBM)

The GBM dynamics refers to stochastic differential equations of the form

$$dX_t = aX_t dt + \sigma X_t dB_t, \tag{12}$$

where $a, \sigma$ are constants and $B$ is a one dimensional Brownian motion. This SDE played a crucial role in financial mathematics and is often used as a model of stock prices. It admits a closed-form solution given in the next lemma.

**Lemma 14.** *Eq.* (12) *admits the following solution*

$$X_t = X_0 \exp\left(\left(a - \frac{1}{2}\sigma^2\right)t + \sigma B_t\right).$$

*The distribution of $X_t$ is known as a log-Gaussian distribution. Moreover, the solution is unique.*

*Proof.* The existence and uniqueness of the solution follows from Theorem 4. Indeed, it suffices to have the drift and the volatility both Lipschitz to obtain the result. This is satisfied in the case of GBM. Now consider the process $Z_t = \log(X_t)$. Using Itô lemma[10], it is easy to verify that

$$dZ_t = \left(a - \frac{1}{2}\sigma^2\right)dt + \sigma dB_t,$$

we conclude by integrating both sides.

$\square$

Now let us find sufficient conditions under which the infinite-depth network represented by the process $X$ has a GBM behaviour. In order for this to hold, it suffices to have

$$\begin{cases} \mu(y) = \frac{1}{2n}\phi(y)^2 g''(y) = ag(y) \\ \sigma(y) = \frac{1}{\sqrt{n}}\phi(y)g'(y) = \sigma g(y). \end{cases}$$

This implies $\frac{g''}{g'^2} \propto \frac{1}{g}$, or equivalently $\frac{g''}{g'} \propto \frac{g'}{g}$, which in turn yields $\log(|g'|) = \alpha \log(|g|) + \beta$, and therefore $|g'| \propto |g|^\zeta$. Assuming that $g', g > 0$, we can easily verify that functions of the form $g(y) = \alpha(y + \beta)^\gamma$ where $\alpha, \beta, \gamma > 0$ satisfy the requirements. Hence, the activation function should satisfy $\phi(y) = \sigma\gamma^{-1}(y + \beta)$, i.e.

---

[10]Notice that here, $X_t$ should be positive in order to consider $\log(X_t)$. This is easy to show and the proof is similar to that of Lemma 15.

the activation should be linear. In this case, we have $a = \frac{1}{2}\sigma^2\gamma^{-1}(\gamma - 1)$ and the process $g(X_t)$ has the following GBM dynamics

$$dg(X_t) = ag(X_t)dt + \sigma g(X_t)dB_t.$$

From Lemma 14, we conclude that

$$g(X_1) \sim g(X_0)\exp\left(\left(a - \frac{1}{2}\sigma^2\right)t + \sigma B_1\right).$$

Observe that in the special case of $\gamma = 1, \beta = 0, \alpha = 1$, we have $g(y) = y$ and $a = 0$. In this case, we obtain $Y_1 \sim Y_0\exp\left(-\frac{1}{2}\sigma^2 t + \sigma B_1\right)$.

We summarize the previous results in following proposition.

**Proposition 7.** *Let $x \in \mathbb{R}$ such that $x \neq 0$. Consider the following activation function $\phi$*

$$\phi(y) = \alpha y + \beta,$$

*where $\alpha > 0, \beta \in \mathbb{R}$ are constants. Let $\sigma > 0$ and define the function $g$ by*

$$g(y) = (\alpha y + \beta)^\gamma.$$

*where $\gamma = \sigma\alpha^{-1}$. Consider the stochastic process $X_t$ defined by*

$$dX_t = |\phi(X_t)|dB_t, \quad X_0 = W_{in}x.$$

*Then, the process $g(X_t)$ satisfies the following GBM dynamics*

$$dg(X_t) = ag(X_t)dt + \sigma g(X_t)dB_t,$$

*where $a = \frac{1}{2}\sigma^2\gamma^{-1}(\gamma - 1)$. As a result, we have that for all $t \in [0, 1]$,*

$$g(X_t) \sim g(X_0)\exp\left(\left(a - \frac{1}{2}\sigma^2\right)t + \sigma B_t\right).$$

## F  ReLU in the case $n = d = 1$

Consider the process $X$ given by the SDE

$$dX_t = \phi(Y_t)dB_t, \quad t \in [0, 1], X_0 > 0.$$

where $\phi(z) = \max(z, 0)$ for $z \in \mathbb{R}$ is the ReLU activation function. Note that we assume $X_0 > 0$ in this case. We will deal with the general case later in this section.

It is straightforward that if $X_s \leq 0$ for some $s \in [0, 1]$, then for all $t \geq s, X_t = X_s$. This is because $dX_t = 0 \times dB_t$ whenever $X_t \leq 0$. A rigorous justification is provided in Lemma 1. Hence, the event $\{X_s \leq 0\}$ constitutes a stopping event where the process becomes constant. We also say that 0 is an absorbent point of the process $X$. A classic tool in stochastic calculus to deal with such situations is the notion of *stopping time* which is a random variable that depend on the trajectory of $X$ (or equivalently on the natural filtration $\mathcal{F}_t$ associated with the Brownian motion $B$). Consider the following stopping time

$$\tau = \inf\{t \in [0, 1], \text{ s.t. } X_s \leq 0\}. \tag{13}$$

Observe that we have for all $t \in [0, \tau]$

$$dX_t = X_t dB_t,$$

which implies that $Y_t$ is a Geometric Brownian motion in the interval $[0, \tau]$. Hence, if $\tau > 1$ (a.s.), the network output has also a log-normal distribution in the infinite-depth limit. In the next lemma, we show that $\tau = \infty$ with probability 1 which confirms the above.

**Lemma 15.** *Let $\tau$ be the stopping time defined by Eq. (13). We have that*

$$\mathbb{P}(\tau = \infty) = 1.$$

*Proof.* By continuity of the Brownian path and the ReLU function $\phi$, the paths of the process $X$ are also continuous[11]. we have that $\tau > 0$ almost surely. From the observation above, taking the limit $t \to \tau^-$ and using the continuity, we obtain

$$X_\tau = X_0 \exp\left(-\frac{1}{2}\tau + B_\tau\right).$$

For some $\omega \in \{\tau < \infty\}$, we have that $X_\tau(\omega) = 0$ (by continuity). Hence $-\frac{1}{2}\tau(w) + X_\tau(\omega) = -\infty$. This happens with probability zero, which means that the event $\{\tau < \infty\}$ has probability zero. This concludes the proof. $\qquad\square$

Hence, with the ReLU activation function, given $X_0 > 0$, the network output is distributed as

$$X_1 \sim X_0 \exp\left(-\frac{1}{2} + B_1\right).$$

Now let us go back to the original setup for $X_0$. Recall that $X_0 = W_{in}x$ for some $x \neq 0$ and $W_{in} \sim \mathcal{N}(0,1)$. By conditioning on $X_0$ and observing that 0 is an absorbent point of the process $X$, we obtain that

$$X_1 \sim \mathbb{1}_{\{X_0 > 0\}} X_0 \exp\left(-\frac{1}{2} + B_1\right) + \mathbb{1}_{\{X_0 \leq 0\}} X_0.$$

We summarize these results in the next proposition.

**Proposition 8.** *Let $x \in \mathbb{R}$ such that $x \neq 0$, and let $\phi$ be the ReLU activation function given by $\phi(z) = \max(z, 0)$ for all $z \in \mathbb{R}$. Consider the stochastic process $X_t$ defined by*

$$dX_t = \phi(X_t)dB_t, \quad X_0 = W_{in}x.$$

*Then, the process $X$ is a mixture of a Geometric Brownian motion and a constant process. More precisely, we have for all $t \in [0, 1]$*

$$X_t \sim \mathbb{1}_{\{X_0 > 0\}} X_0 \exp\left(-\frac{1}{2}t + B_t\right) + \mathbb{1}_{\{X_0 \leq 0\}} X_0.$$

*Hence, conditionally on $X_0 > 0$, the process $X$ is a Geometric Bronwian motion.*

## G   Proof of Lemma 2 and Lemma 3

**Lemma 2.** *Let $x \in \mathbb{R}^d$ such that $x \neq 0$, and consider the stochastic process $X$ given by the following SDE*

$$dX_t = \frac{1}{\sqrt{n}}\|\phi(X_t)\|dB_t, \quad t \in [0, \infty), \quad X_0 = W_{in}x,$$

*where $\phi(z) : \mathbb{R} \to \mathbb{R}$ is Lipschitz, injective, $\mathcal{C}^2(\mathbb{R})$ and satisfies $\phi(0) = 0$, and $\phi'$ and $\phi''\phi$ are bounded on $\mathbb{R}$, and $(B_t)_{t \geq 0}$ is an $n$-dimensional Brownian motion independent from $W_{in} \sim \mathcal{N}(0, d^{-1}I)$. Let $\tau$ be the stopping time given by*

$$\tau = \min\{t \geq 0 : \phi(X_t) = 0\}.$$

*Then, we have that*

$$\mathbb{P}(\tau = \infty) = 1.$$

---

[11] This is a classic result in stochastic calculus. More rigorously, $X$ can be chosen to have continuous paths with probability 1.

*Proof.* It is straightforward that with probability 1 we have $\|\phi(X_0)\| > 0$, which implies that with probability 1, $\tau > 0$. Let $t < \tau$. Using Itô 's lemma with the function $g(z) = \frac{1}{2}\log(\|\zeta(x)\|^2)$, we obtain

$$dg(X_t) = \nabla g(X_t)^\top dX_t + \frac{1}{2n}\|\zeta(X_t)\|^2 \mathrm{Tr}(\nabla^2 g(X_t))dt.$$

Therefore,

$$g(X_t) - g(X_0) = \frac{1}{\sqrt{n}}\int_0^t \mu(X_s)ds + \frac{1}{2n}\int_0^t \sum_{i=1}^n \sigma_i(X_s)dB_s^i,$$

where $\sigma_i(X_s) = \frac{|\phi'(X_s^i)\phi(X_s^i)|}{\|\phi(X_s)\|}$, and $\mu(X_s) = \frac{1}{2}\sum_{i=1}^n \left(\phi''(X_s^i)\phi(X_s^i) + \phi'(X_s^i)^2\right) - \frac{\|\phi'(X_s)\circ\phi(X_s)\|^2}{\|\phi(X_s)\|^2}$, and $\circ$ refers to the Hadamard product of vectors, i.e. coordinate-wise product.

For some $\omega \in \{\tau < \infty\}$, using the path continuity of the process $X$ and the continuity of $g$, we have that $\lim_{t\to\tau(\omega)^-} g(X_{\tau(\omega)}(\omega)) = -\infty$. Therefore, we should also have

$$\frac{1}{\sqrt{n}}\int_0^{\tau(\omega)} \mu(X_s(\omega))ds + \frac{1}{2n}\int_0^{\tau(\omega)} \sum_{i=1}^n \sigma_i(X_s(\omega))dB_s^i(\omega) = -\infty.$$

Hence, we have that

$$\mathbb{P}\left(\tau < \infty\right) \leq \mathbb{P}\left(\frac{1}{\sqrt{n}}\int_0^t \mu(X_s)ds + \frac{1}{2n}\int_0^t \sum_{i=1}^n \sigma_i(X_s)dB_s^i = -\infty\right)$$

$$= \lim_{A\to\infty} \mathbb{P}\left(\frac{1}{\sqrt{n}}\int_0^t \mu(X_s)ds + \frac{1}{2n}\int_0^t \sum_{i=1}^n \sigma_i(X_s)dB_s^i \leq -A\right)$$

$$= \lim_{A\to\infty} \mathbb{P}\left(\frac{1}{\sqrt{n}}\int_0^t \mu(X_s)ds + \frac{1}{2n}\int_0^t \sigma(X_s)d\hat{B}_s \leq -A\right),$$

where $\hat{B}$ is a one-dimensional Brownian motion, and where we use Theorem 7, and $\sigma(X_s) = (\sum_{i=1}^n \sigma_i(X_s)^2)^{1/2} = \frac{\|\phi'(X_s)\circ\phi(X_s)\|}{\|\phi(X_s)\|}$. Using the conditions on $\phi$, there exists a constant $K > 0$ such $|\phi'| \leq K, |\phi''\phi| \leq K$. With this we obtain for all $Z \in \mathbb{R}^n$

$$|\sigma(Z)| = \frac{\|\phi'(Z)\circ\phi(Z)\|}{\|\phi(Z)\|} \leq K,$$

and

$$|\mu(Z)| = \left|\frac{1}{2}\sum_{i=1}^n \left(\phi''(Z^i)\phi(Z^i) + \phi'(Z^i)^2\right) - \frac{\|\phi'(Z)\circ\phi(Z)\|^2}{\|\phi(Z)\|^2}\right| \leq \frac{1}{2}nK + \left(\frac{1}{2}n + 1\right)K^2.$$

Hence, the random variable $\frac{1}{\sqrt{n}}\int_0^t \mu(X_s)ds + \frac{1}{2n}\int_0^t \sigma(X_s)d\hat{B}_s$ is finite with probability 1. We conclude that

$$\mathbb{P}\left(\tau = \infty\right) = 1.$$

$\square$

**Lemma 3.** *Consider the stochastic process* (8) *given by the SDE*

$$dX_t = \frac{1}{\sqrt{n}}\|\phi(X_t)\|dB_t, \quad t \in [0, \infty), \quad X_0 = W_{in}x,$$

*where $\phi$ is the ReLU activation function, and $(B_t)_{t\geq 0}$ is an $n$-dimensional Brownian motion. Let $\tau$ be the stopping time given by*

$$\tau = \min\{t \geq 0 : \|\phi(X_t)\| = 0\} = \min\{t \geq 0 : \forall i \in [n], X_t^i \leq 0\}.$$

*Then, we have that*

$$\mathbb{P}\left(\tau = \infty \mid \|\phi(X_0)\| > 0\right) = 1.$$

*As a result, we have that*

$$\mathbb{P}(\tau = \infty) = 1 - 2^{-n}.$$

*Proof.* Let $t_0 > 0$. Using Lemma 7, we know that if for some $t_1$, $\|\phi(X_{t_1})\| = 0$, then for all $t \geq t_1$, we have that $X_t = X_{t_1}$ and $\|\phi(X_t)\| = 0$. Hence, we have that

$$\mathbb{P}\left(\tau \leq t_0 \mid \|\phi(X_0)\| > 0\right) = \mathbb{P}\left(\|\phi(X_{t_0})\| = 0 \mid \|\phi(X_0)\| > 0\right).$$

Let $m \geq 1$ and consider the function $\phi_m(z) = \int_0^z h(m\,u)du$ and $h(t) = (1 + e^{-t})^{-1}$ is the Sigmoid function[12]. It is straightforward that $\phi_m$ satisfies the conditions of Lemma 2. Let $X^m$ be the solution of the following SDE (the solution exists and is unique since $\phi_m$ is trivially Lipschitz)

$$dX_t^m = \frac{1}{\sqrt{n}}\|\phi_m(X_t^m)\|dB_t, \quad t \in [0, \infty), \quad X_0 = W_{in}x.$$

We know from Lemma 8 that $X_t^m$ converges in $L^2$ to $X_t$ (uniformly over $t \in [0, T]$ for any $T > 0$). In particular, this implies convergence in distribution. Moreover, observe that for all $t$

$$\mathbb{E}\|\phi_m(X_t^m) - \phi(X_t)\|^2 \leq \frac{2n}{m^2} + 2\mathbb{E}\|X_t^m - X_t\|^2,$$

where we used triangular inequality and the upperbound from Lemma 9. Thus, we have that $\phi_m(X_t^m)$ converges in $L^2$ (and in distribution) to $\phi(X_t)$.

Let $\delta_k = [1/(k+1), 1/k)$ for $k \geq 1$, and define $\delta_0 = [1, \infty)$. For $m \geq 1$, using Lemma 9, we have that

$$\mathbb{P}\left(\|\phi(X_{t_0})\| = 0 \cap \|\phi(X_0)\| > 0\right) \leq \sum_{k=0}^{\infty} \mathbb{P}\left(\|\phi_m(X_{t_0})\| \leq 1/m \cap \|\phi(X_0)\| \in \delta_k\right).$$

Given $k \geq 0$, we have that for $m > n^{1/2}(k+1)$,

$$\begin{aligned}
\mathbb{P}\left(\|\phi_m(X_{t_0})\| \leq 1/m \cap \|\phi(X_0)\| \in \delta_k\right) &\leq \mathbb{P}\left(\|\phi_m(X_{t_0}^m)\| \leq 1/m + \log(m)/m \cap \|\phi(X_0)\| \in \delta_k\right) \\
&\quad + \mathbb{P}\left(\|\phi_m(X_{t_0}^m) - \phi(X_{t_0})\| > \log(m)/m \cap \|\phi(X_0)\| \in \delta_k\right)
\end{aligned} \quad (14)$$

Let us deal with the first term. Using Lemma 9, we have that

$$\begin{aligned}
&\mathbb{P}\left(\|\phi_m(X_{t_0}^m)\| \leq 1/m + \log(m)/m \cap \|\phi(X_0)\| \in \delta_k\right) \\
&\leq \mathbb{P}\left(\|\phi_m(X_{t_0}^m)\| \leq 1/m + \log(m)/m \cap \|\phi_m(X_0^m)\| \geq 1/(k+1) - n^{1/2}m^{-1} \cap \|\phi(X_0)\| \in \delta_k\right) \\
&\leq \mathbb{P}\left(\log\left(\frac{\|\phi_m(X_{t_0}^m)\|}{\|\phi_m(X_0^m)\|}\right) \leq -\log(m/(1 + \log(m)) + \log((k+1)^{-1} - m^{-1}n^{1/2}) \cap \|\phi(X_0)\| \in \delta_k\right)
\end{aligned}$$

From Lemma 10, we know that the random variable $\log\left(\frac{\|\phi_m(X_{t_0}^m)\|}{\|\phi_m(X_0^m)\|}\right)$ converges in $L^1$ and thus it is bounded in $L^1$ norm (over $m$). Therefore, a simple application of Markov's inequality yields that the probability above goes to 0 when $m$ goes to $\infty$.

The second term in Eq. (14) also converges to 0 using the $L^2$ convergence of $\phi_m(X_{t_0}^m)$ to $\phi(X_{t_0})$ coupled with a simple application of Markov's inequality. We therefore obtain that for all $k \geq 0$,

---

[12]Note that $\phi_m$ has a closed-form formula given by $\phi_m(z) = m^{-1}(\log(1 + e^{mz}) - \log(2))$, which can be seen as a shifted and scaled version of the Softplus function. However, we do not need the closed-form formula in our analysis.

$\lim_{m \to \infty} \mathbb{P} \left( \|\phi_m(X_{t_0})\| \leq 1/m \cap \|\phi(X_0)\| \in \delta_k \right) = 0$. Using the Dominated convergence theorem, we obtain that for all $t_0$,

$$\mathbb{P} \left( \|\phi(X_{t_0})\| = 0 \cap \|\phi(X_0)\| > 0 \right) = 0,$$

which implies that $\mathbb{E} \left[ \mathbb{1}_{\{\tau > t\}} | \|\phi(X_0)\| > 0 \right] = 1$ for all $t > 0$. Another application of the Dominated convergence theorem yields the result. The second part is straightforward by observing that $\mathbb{P} \left( \|\phi(X_0)\| = 0 \right) = 2^{-n}$. $\qquad \square$

## H    Proof of Theorem 1

**Theorem 1.**    *We have that for all $t \in [0, 1]$,*

$$\|\phi(X_t)\| = \|\phi(X_0)\| \exp \left( \frac{1}{\sqrt{n}} \hat{B}_t + \frac{1}{n} \int_0^t \mu_s ds \right), \quad \text{almost surely,}$$

*where $\mu_s = \frac{1}{2}\|\phi'(X_s)\|^2 - 1$, and $(\hat{B})_{t \geq 0}$ is a one-dimensional Brownian motion. As a result, we have that for all $0 \leq s \leq t \leq 1$*

$$\mathbb{E} \left[ \log \left( \frac{\|\phi(X_t)\|}{\|\phi(X_s)\|} \right) | \|\phi(X_0)\| > 0 \right] = \left( \frac{1 - 2^{-n}}{4} - \frac{1}{n} \right) (t - s),$$

*Moreover, for $n \geq 2$, we have*

$$\text{Var} \left[ \log \left( \frac{\|\phi(X_t)\|}{\|\phi(X_s)\|} \right) | \|\phi(X_0)\| > 0 \right] \leq \left( n^{-1/2} + \Gamma_{s,t}^{1/2} \right)^2 (t - s),$$

*where $\Gamma_{s,t} = \frac{1}{4} \int_s^t \left( \left( \mathbb{E}\phi'(X_u^1)\phi'(X_u^2) - \frac{(1-2^{-n})^2}{4} \right) + n^{-1} \left( \frac{1-2^{-n}}{2} - \mathbb{E}\phi'(X_u^1)\phi'(X_u^2) \right) \right) du$.*

*Proof.* Let $t \in [0, 1]$. Let us firs consider the case where $\|\phi(X_0)\| = 0$. For all $t$ we have $\|\phi(X_t)\| = 0$ and the result is trivial.

We consider the case where $\|\phi(X_0)\| > 0$ (happens with probability $1 - 2^{-n}$), and all the expectations in this proof are conditionally on this event. Consider the function $g : \mathbb{R}^n \to \mathbb{R}$ given by

$$g(x) = \log(\|\phi(x)\|) = \frac{1}{2} \log(\|\phi(x)\|^2).$$

Ideally, we would like to use Itô 's lemma and Lemma 3, which ensures that $\|\phi(X_t)\|$ remains positive on $[0, 1]$, and obtain for all $t \in [0, 1]$

$$dg(X_t) \stackrel{dist}{=} \frac{1}{\sqrt{n}} d\hat{B}_t + \frac{1}{n} \mu_t dt,$$

where $\mu_t$ is some well defined quantity. This would let us conclude. However, Itô 's lemma requires that the function be $\mathcal{C}^2(\mathbb{R}^n)$, which is violated by our choice of $g$. To mitigate this issue, we consider a sequence of function $(g_m)_{m \geq m}$ that approximates the function $g$ when $m$ goes to infinity. For $m \geq 1$, let $g_m$ be defined by

$$g_m(x) = \frac{1}{2} \log(\|\phi_m(x)\|^2),$$

where $\phi_m(t) = \int_0^t h(m\,u)du$ and $h(t) = (1 + e^{-t})^{-1}$ is the Sigmoid function. We have that

$$\begin{cases} \frac{\partial g_m}{\partial x_i}(x) = \frac{h(mx_i)\phi_m(x_i)}{\|\phi_m(x)\|^2} \\ \frac{\partial^2 g_m}{\partial x_i^2}(x) = \frac{mh(mx_i)(1-h(mx_i))\phi_m(x_i)+h(mx_i)^2}{\|\phi_m(x)\|^2} - 2\frac{h(mx_i)^2\phi_m(x_i)^2}{\|\phi_m(x)\|^4} \end{cases}$$

Let $X^m$ be the solution of the following SDE

$$dX_t^m = \frac{1}{\sqrt{n}} \|\phi_m(X_t^m)\| dB_t, \quad t \in [0, \infty), \quad X_0 = W_{in}x,$$

Using Itô 's lemma, we have that

$$dg_m(X_t^m) = \frac{1}{n}\mu_s^m ds + \frac{1}{\sqrt{n}}\sum_{i=1}^n \sigma_s^{m,i} dB_s^i,$$

where

$$\mu_s^m = \frac{1}{2}\sum_{i=1}^n \left(mh(mX_s^{m,i})(1-h(mX_s^{m,i}))\phi_m(X_s^{m,i}) + h(mX_s^{m,i})^2\right) - \frac{\|h(mX_s^m)\circ\phi_m(X_s^m)\|^2}{\|\phi_m(X_s^m)\|^2},$$

and $\sigma_s^{m,i} = \frac{|h(mX_s^{m,i})\circ\phi_m(X_s^{m,i})|}{\|\phi_m(X_s^m)\|}$. By Lemma 10, we know that $g_m(X_s^m) - g_m(X_0^m)$ converges in $L_1$ to $g(X_s) - g(X_0)$. Let us now compute the limit of $g_m(X_s^m) - g_m(X_0^m)$ from the equation above to conclude. More precisely, let us show that for all $s \in (0,1)$

$$\lim_{m\to\infty} \mathbb{E}\,|g_m(X_s^m) - g_m(X_0^m) - (Z_s - Z_0)| = 0,$$

where $Z$ is the process given by

$$Z_t = g(X_0) + \frac{1}{\sqrt{n}}\left(\sum_{i=1}^n \int_0^t \sigma_s^i dB_t^i\right) + \frac{1}{n}\int_0^t \mu_s ds,$$

where $\mu_s = \frac{1}{2}\|\phi'(X_s^i)\|^2 - 1$, and $\sigma_s^i = \frac{\phi(X_s^i)}{\|\phi(X_s)\|}$.

Let $t \in (0,1]$. Using triangular inequality, Itô isometry, and Cauchy-Schwartz inequality, we have that

$$\begin{aligned}
\mathbb{E}\,|g_m(X_t^m) - g_m(X_0^m) - (Z_t - Z_0)| &\le \frac{1}{n}\int_0^t \mathbb{E}\,|\mu_s^m - \mu_s|\,ds + \frac{1}{\sqrt{n}}\mathbb{E}\left|\int_0^t \sum_{i=1}^n (\sigma_s^i - \sigma_s^{m,i})dB_s^i\right| \\
&\le \frac{1}{n}\int_0^t \mathbb{E}\,|\mu_s^m - \mu_s|\,ds + \frac{1}{\sqrt{n}}\left(\mathbb{E}\int_0^t \sum_{i=1}^n (\sigma_s^i - \sigma_s^{m,i})^2 ds\right)^{1/2}
\end{aligned} \tag{15}$$

We first deal with the the term $\int_0^t \mathbb{E}\,|\mu_s^m - \mu_s|\,ds$. Let us show that this term converges to 0. Let us show the following,

$$\forall s > 0, \lim_{m\to\infty} \mathbb{E}\,|\mu_s^m - \mu_s| = 0.$$

Let $s \in [0,1]$. We have that

$$\mu_s^m = \frac{1}{2}\sum_{i=1}^n J_i^m - G^m,$$

where $J_i^m = mh(mX_s^{m,i})(1-h(mX_s^{m,i}))\phi_m(X_s^{m,i}) + h(mX_s^{m,i})^2$, and $G^m = \frac{\|h(mX_s^m)\circ\phi_m(X_s^m)\|^2}{\|\phi_m(X_s^m)\|^2}$. Let us start with the term $G^m$. Observe that $G^m \le 1$ almost surely. We have that

$$\begin{aligned}
\mathbb{E}|1 - G^m| &= \mathbb{E}\left[\frac{\|(1-h(mX_s^m)^2)^{1/2}\circ\phi_m(X_s^m)\|^2}{\|\phi_m(X_s^m)\|^2}\right] \\
&= \mathbb{E}\left[\frac{\|(1-h(mX_s^m)^2)^{1/2}\circ\phi_m(X_s^m)\|^2}{\|\phi_m(X_s^m)\|^2}\mathbb{1}_{\{\min_i |X_s^{m,i}|\ge\log(m)/m\}}\right] \\
&\quad + \mathbb{E}\left[\frac{\|(1-h(mX_s^m)^2)^{1/2}\circ\phi_m(X_s^m)\|^2}{\|\phi_m(X_s^m)\|^2}\mathbb{1}_{\{\min_i |X_s^{m,i}|<\log(m)/m\}}\right]
\end{aligned}$$

When $\min_i |x^i| \geq \log(m)/m$, we have that for all $i \in [n]$

$$(1 - h(mx^i)^2) \leq 2(1 - h(mx^i)) \leq 2\exp(-m \times \log(m)/m) = 2m^{-1}.$$

Therefore,

$$\mathbb{E}\left[\frac{\|(1 - h(mX_s^m)^2)^{1/2} \circ \phi_m(X_s^m)\|^2}{\|\phi_m(X_s^m)\|^2}\mathbb{1}_{\{\min_i |X_s^{m,i}| \geq \log(m)/m\}}\right] \leq 2m^{-1}.$$

For the remaining term, using the fact that $1 - h^2 \leq 1$, we have that

$$\mathbb{E}\left[\frac{\|(1 - h(mX_s^m)^2)^{1/2} \circ \phi_m(X_s^m)\|^2}{\|\phi_m(X_s^m)\|^2}\mathbb{1}_{\{\min_i |X_s^{m,i}| < \log(m)/m\}}\right] \leq \mathbb{P}\left(\min_i |X_s^{m,i}| < \log(m)/m\right)$$
$$\leq \sum_{i=1}^{n} \mathbb{P}\left(|X_s^{m,i}| < \log(m)/m\right).$$

Now using Lemma 8, we have that

$$\mathbb{P}(\|X_s^m - X_s\| \geq \log(m)/m) \leq \frac{C_n}{\log(m)^2},$$

for some constant $C_n$ that depends on $n$. Therefore, for all $i$, we have

$$\mathbb{P}(|X_s^{m,i}| < \log(m)/m) \leq \mathbb{P}(|X_s^i| < 2\log(m)/m) + \mathbb{P}(|X_s^{m,i} - X_s^i| \geq \log(m)/m)$$
$$\leq \mathbb{P}(|X_s^i| < 2\log(m)/m) + \frac{C_n}{\log(m)^2}.$$

Recall that $X_s^i = X_0^i + \frac{1}{\sqrt{n}}\int_0^s \|\phi(X_u)\|dB_u$. By Lemma 11, we know that

$$\mathbb{P}(|X_s^i| < 2\log(m)/m) = \mathcal{O}(\log(m)/m),$$

We conclude that $\lim_{m\to\infty} \mathbb{E}|1 - G^m| = 1$.

Now let us show that for all $i$, $\lim_{m\to\infty} \mathbb{E}|J_i^m - \phi'(X_s^i)| = 0$. Let $i \in [n]$ and $A_s^{m,i} = mh(mX_s^{m,i})(1 - h(mX_s^{m,i}))\phi_m(X_s^{m,i})$. We have that

$$\mathbb{E}|A_s^{m,i}| = \mathbb{E}|A_s^{m,i}|\mathbb{1}_{\{|X_s^{m,i}| \leq 2\log(m)/m\}} + \mathbb{E}|A_s^{m,i}|\mathbb{1}_{\{|X_s^{m,i}| \leq 2\log(m)/m\}}$$
$$\leq m \times 2\log(m)/m \times \mathbb{P}(|X_s^{m,i}| \leq 2\log(m)/m) + m^{-1}\mathbb{E}|\phi_m(X_s^{m,i})|$$
$$\leq 2\log(m) \times \left(\mathbb{P}(|X_s^i| \leq 3\log(m)/m) + \mathbb{P}(|X_s^{m,i} - X_s^i| \geq \log(m)/m)\right) + m^{-1}\mathbb{E}|\phi_m(X_s^{m,i})|$$
$$\leq 2\log(m) \times \mathbb{P}(|X_s^i| \leq 3\log(m)/m) + \frac{2C_n}{\log(m)} + m^{-1}\mathbb{E}|\phi_m(X_s^{m,i})|,$$

where we have used Lemma 8 and Markov's inequality. Using Lemma 9 and the fact that the absolute value function is Lipschitz, we know that $\lim_{m\to\infty} \mathbb{E}|\phi_m(X_s^{m,i})| = \mathbb{E}|\phi(X_s^i)| < \infty$. Therefore, the third term vanishes in the limit $m \to \infty$. The second term $2C_n/\log(m)$ also vanishes. The first term also vanished using Lemma 11. Therefore, $\lim_{m\to\infty} \mathbb{E}|A_s^{m,i}| = 0$.

Let us now deal with the last term in $J_i^m$. We have that

$$
\begin{aligned}
\mathbb{E}\left|h(mX_s^{m,i})^2 - \phi'(X_s^i)\right| &= \mathbb{E}\left|h(mX_s^{m,i})^2 - \phi'(X_s^i)\right| \mathbb{1}_{\{|X_s^{m,i}| \geq \log(m)/m\}} \\
&+ \mathbb{E}\left|h(mX_s^{m,i})^2 - \phi'(X_s^i)\right| \mathbb{1}_{\{|X_s^{m,i}| < \log(m)/m\}} \\
&\leq 3m^{-1} + 2\mathbb{P}(|X_s^{m,i}| < \log(m)/m) \\
&\leq 3m^{-1} + 2\mathbb{P}(|X_s^i| < 2\log(m)/m) + 2\frac{C_n}{\log(m)^2}
\end{aligned}
$$

Using Lemma 11, we obtain that $\lim_{m\to\infty} \mathbb{E}|h^2(mX_s^{m,i}) - \phi'(X_s^i)| = 0$. Hence, we obtain that $\lim_{m\to\infty} \mathbb{E}|J_i^m - \phi'(X_s)| = 0$. We conclude that $\lim_{m\to\infty} \mathbb{E}|\mu_s^m - \mu_s| = 0$. Moreover, from the analysis above, it is easy to see that $\sup_{m\geq 1,\, s\in(0,1]} \mathbb{E}|\mu_s^m - \mu_s| < \infty$.

We now deal with the second term $\left(\mathbb{E}\int_0^t \sum_{i=1}^n (\sigma_s^i - \sigma_s^{m,i})^2 ds\right)^{1/2}$ from Eq. (15). For this part only, we define the stopping time $\tau_\epsilon$ for $\epsilon \in (0, \|\phi(X_0)\| \wedge \|\phi(X_0)\|^{-1})$ (Recall that the analysis is conducted conditionally on the fact that $\|\phi(X_0)\| > 0$) by

$$
\tau_\epsilon = \inf\{t \geq 0, \text{ s.t. } \|\phi(X_t)\| \in [0, \epsilon] \cup [\epsilon^{-1}, \infty)\}.
$$

Notice that $\tau_\epsilon > 0$ almost surely since $\|\phi(X_0)\| \in (\epsilon, \epsilon^{-1})$.

Let $s \in (0, 1]$. We have that

$$
\begin{aligned}
\mathbb{E}\int_0^{t\wedge\tau_\epsilon} \sum_{i=1}^n (\sigma_s^i - \sigma_s^{m,i})^2 &\leq 2\mathbb{E}\int_0^{t\wedge\tau_\epsilon} \sum_{i=1}^n \left(\frac{\phi(X_s^i)}{\|\phi(X_s)\|} - \frac{\phi_m(X^{m,i})}{\|\phi_m(X_s^m)\|}\right)^2 ds \\
&+ 2\mathbb{E}\int_0^{t\wedge\tau_\epsilon} \frac{\|(1 - h(mX_s^m)^2)^{1/2} \circ \phi_m(X_s^m))\|}{\|\phi(X_s^m)\|^2} ds.
\end{aligned}
$$

The second term can be upperbounded in the following fashion

$$
\begin{aligned}
\mathbb{E}\int_0^{t\wedge\tau_\epsilon} \frac{\|(1 - h(mX_s^m)^2)^{1/2} \circ \phi_m(X_s^m))\|}{\|\phi(X_s^m)\|^2} ds &\leq \int_0^t \mathbb{E}\frac{\|(1 - h(mX_s^m)^2)^{1/2} \circ \phi_m(X_s^m))\|}{\|\phi(X_s^m)\|^2} ds \\
&= \int_0^t \mathbb{E}|1 - G_m|,
\end{aligned}
$$

where $G_m$ is defined above. We know that $\int_0^t \mathbb{E}|1 - G_m|$ converges to 0 in the limit $m \to \infty$ by the Dominated convergence theorem (the integrand is bounded). Let us show that the first term also vanishes. We have that

$$
\begin{aligned}
\mathbb{E}\int_0^{t\wedge\tau_\epsilon} \sum_{i=1}^n \left(\frac{\phi(X_s^i)}{\|\phi(X_s)\|} - \frac{\phi_m(X^{m,i})}{\|\phi_m(X_s^m)\|}\right)^2 ds &\leq 2\mathbb{E}\int_0^{t\wedge\tau_\epsilon} \sum_{i=1}^n \left(\frac{\phi(X_s^i) - \phi_m(X^{m,i})}{\|\phi(X_s)\|}\right)^2 ds \\
&+ 2\mathbb{E}\int_0^{t\wedge\tau_\epsilon} \sum_{i=1}^n \phi_m(X_s^{m,i})^2 \left(\frac{1}{\|\phi(X_s)\|} - \frac{1}{\|\phi(X_s^m)\|}\right)^2 ds \\
&\leq 2\epsilon^{-2}\mathbb{E}\int_0^{t\wedge\tau_\epsilon} \|\phi(X_s^i) - \phi_m(X_s^{m,i})\|^2 ds \\
&+ 2\mathbb{E}\int_0^{t\wedge\tau_\epsilon} \sum_{i=1}^n \phi_m(X_s^{m,i})^2 \left(\frac{1}{\|\phi(X_s)\|} - \frac{1}{\|\phi(X_s^m)\|}\right)^2 ds.
\end{aligned}
$$

The first term $2\epsilon^{-2}\mathbb{E}\int_0^{t\wedge\tau_\epsilon}\|\phi(X_s^i) - \phi_m(X_s^{m,i})\|^2 ds$ converges to 0 in the limit $m \to \infty$ by Lemma 9 and Lemma 8. Let us deal with the second term. Define the event $E = \{\sup_{s\in(0,1]}\|X_s^m - X_s\| \leq \log(m)/m\}$ for $m$ large enough such that $\log(m)/m < \epsilon$. Observe that on the event $E$, we have that for all $s \in (0,1]$, $\|\phi(X_s^m) - \phi(X_s)\| \leq (\sqrt{n} + \log(m))m^{-1}$. Hence,

$$\mathbb{E}\,\mathbb{1}_E \int_0^{t\wedge\tau_\epsilon} \sum_{i=1}^n \phi_m(X_s^{m,i})^2 \left(\frac{1}{\|\phi(X_s)\|} - \frac{1}{\|\phi(X_s^m)\|}\right)^2 ds \leq \epsilon^{-2}(\sqrt{n}+\log(m))^2 m^{-2} \to_{m\to\infty} 0.$$

Moreover, letting $E^c$ be the complementary event of $E$, we have

$$\mathbb{E}\,\mathbb{1}_{E^c} \int_0^{t\wedge\tau_\epsilon} \sum_{i=1}^n \phi_m(X_s^{m,i})^2 \left(\frac{1}{\|\phi(X_s)\|} - \frac{1}{\|\phi(X_s^m)\|}\right)^2 ds$$

$$\leq \mathbb{E}\,\mathbb{1}_{E^c} \int_0^{t\wedge\tau_\epsilon} \sum_{i=1}^n \phi_m(X_s^{m,i})^2 \left(\frac{2}{\|\phi(X_s)\|^2} + \frac{2}{\|\phi(X_s^m)\|^2}\right) ds$$

$$\leq 2\epsilon^{-2}\mathbb{E}\,\mathbb{1}_{E^c} \int_0^{t\wedge\tau_\epsilon} \|\phi_m(X_s^m)\|^2 ds + 2\mathbb{P}(E^c).$$

Using the fact that $\|\phi_m(X_s^m)\| \leq \frac{\sqrt{n}}{m} + \|\phi(X_s)\| + \|X_s^m - X_s\|$ (by Lemma 9 and the fact that ReLU is Lipschitz), we obtain

$$\mathbb{E}\,\mathbb{1}_{E^c} \int_0^{t\wedge\tau_\epsilon} \|\phi_m(X_s^m)\|^2 ds \leq 3nm^{-2} + 3\sup_{s\leq 1}\mathbb{E}\|X_s^m - X_s\|^2 + 3\epsilon^{-2}\mathbb{P}(E^c).$$

The term $\sup_{s\leq 1}\mathbb{E}\|X_s^m - X_s\|^2$ converges to 0 by Lemma 8. Using Doob's martingale inequality on the submartingale $\|X_s^m - X_s\|$ (with respect to the natural filtration generated by the Brownian motion $B$)[13] we obtain that

$$\mathbb{P}(E^c) \leq \frac{\mathbb{E}\|X_1^m - X_1\|^2}{m^{-2}\log(m)^2} = \frac{C_n}{\log(m)^2},$$

where we have used Lemma 8. We conclude that $\lim_{m\to\infty}\mathbb{E}\int_0^{t\wedge\tau_\epsilon}\sum_{i=1}^n(\sigma_s^i - \sigma_s^{m,i})^2 = 0$.

By observing that $\mathbb{E}\int_0^{t\wedge\tau_\epsilon}|\mu_s^m - \mu_s|ds \leq \mathbb{E}\int_0^t \mathbb{E}|\mu_s^m - \mu_s|ds$, a simple application of the Dominated convergence theorem yields $\lim_{m\to\infty}\mathbb{E}\int_0^{t\wedge\tau_\epsilon}|\mu_s^m - \mu_s|ds = 0$. Hence, we proved that

$$\lim_{m\to\infty}\mathbb{E}\left|g_m(X_{t\wedge\tau_\epsilon}^m) - g_m(X_0^m) - (Z_{t\wedge\tau_\epsilon} - Z_0)\right| = 0.$$

From Lemma 10, we know that $g_m(X_{t\wedge\tau_\epsilon}^m) - g_m(X_0^m)$ converges in $L_1$ to $g(X_{t\wedge\tau_\epsilon}) - g(X_0)$[14], therefore $\mathbb{E}|g(X_{t\wedge\tau_\epsilon}) - g(X_0) - (Z_{t\wedge\tau_\epsilon} - Z_0)| = 0$ which implies that almost surely,

$$\log\left(\frac{\|\phi(X_{t\wedge\tau_\epsilon})\|}{\|\phi(X_0)\|}\right) = Z_{t\wedge\tau_\epsilon} - Z_0.$$

Recall that this holds for any $\epsilon$ small enough. Observe that $\tau_\epsilon$ is almost surely non-decreasing as we decrease $\epsilon$. Hence $\tau_\epsilon$ has a limit almost surely. Using Lemma 3 and the continuity of the paths of $X_s$ we have that $\lim_{\epsilon\to 0^+}\tau_\epsilon = \infty$. Taking the limit $\epsilon \to^+$, we conclude that almost surely we have

$$\log\left(\frac{\|\phi(X_t)\|}{\|\phi(X_0)\|}\right) = Z_t - Z_0.$$

---

[13]The submartingale behaviour is a result of the convexity of the norm function and the fact that $X_s^m X_s$ is a martingale since $d(X_s^m X_s) = \frac{1}{\sqrt{n}}(\phi_m(X_s^m) - \phi(X_s))dB_s$. A simple application of Jensen's inequality yields the result.

[14]the result of Lemma 10 holds when $t$ is replaced by $t \wedge \tau_\epsilon$ and the proof is exactly the same. We omit the proof to avoid redundancies

Now observe that the coordinates of $X_t$ are identically distributed (not independent since we condition on $\|\phi(X_0)\| > 0$). Thus, for all $i \in [n]$, $\mathbb{E}\phi'(X_s^i) = \mathbb{E}\phi'(X_s^1)$ where $X_s^1$ is the first coordinate of the vector $X_s$. Another key observation is that the event $\{X_s^1 > 0\}$ is included in the event $\{\|\phi(X_0)\| > 0\}$ (Lemma 1). Hence, $\mathbb{P}(X_s^1 > 0 \cap \|\phi(X_0)\| > 0) = \mathbb{P}(X_s^1 > 0)$, where the last term $\mathbb{P}(X_s^1 > 0)$ is *free* from any conditioning on $\|\phi(X_0)\| > 0$. By observing that the random variable $X_s^1 = X_0^1 + \frac{1}{\sqrt{n}}\int_0^s \|\phi(X_u)\| dB_u^1$ has a symmetric distribution around zero (by properties of the $X_0^1$ and the Brownian motion $B$), we have that $\mathbb{P}(X_s^1 > 0 | \|\phi(X_0)\| > 0) = \mathbb{P}(X_s^1 > 0)\mathbb{P}(\|\phi(X_0)\| > 0)^{-1} = \frac{1}{2}(1 - 2^{-n})^{-1}$.

We conclude that

$$\mathbb{E}\,\mu_s = \frac{n}{4}(1 - 2^{-n})^{-1} - 1,$$

which yields the desired result for the conditional mean by substraction.

Now let us deal with the variance. To alleviate the notation, we omit the conditioning on the event $\{\|\phi(X_0)\| > 0\}$. All the expectations below are taken conditionally on this event. Let $0 \le s \le t \le 1$. Let $\lambda > 0$. We have that

$$\mathrm{Var}\left[\log\left(\frac{\|\phi(X_t)\|}{\|\phi(X_s)\|}\right)^2\right] = \mathrm{Var}(Z_t - Z_s)^2$$

$$\le \frac{(1 + \lambda^{-1})(t - s)}{n} + \frac{1 + \lambda}{n^2}\mathbb{E}\left(\int_s^t (\mu_u - \mathbb{E}\,\mu_u)du\right)^2$$

$$\le (t - s)\left(\frac{1 + \lambda^{-1}}{n} + \frac{1 + \lambda}{n^2}\int_s^t \mathrm{Var}\mu_u\,du\right),$$

where we have used the inequality $(a + b)^2 \le (1 + \lambda^{-1})a^2 + (1 + \lambda)b^2$ and the Cauchy-Schwartz inequality. It remains to simplify $\mathrm{Var}\mu_u^2$. Let $p_1^u = \mathbb{E}\phi'(X_u^1) = 2^{-1}(1 - 2^{-n})$ and $p_2^u = \mathbb{E}\phi'(X_u^1)\phi'(X_u^2)$. We have that

$$\mathbb{E}\mu_u^2 = \frac{1}{4}\mathbb{E}\|\phi'(X_u)\|^4 - \mathbb{E}\|\phi'(X_u)\|^2 + 1$$

$$= \frac{n(n - 1)}{4}p_2^u - \frac{3n}{4}p_1^u + 1,$$

where we have used the exchangeability property of the family $\{\phi'(X_u^i), i = 1, \ldots n\}$. Thus, for the variance $\mathrm{Var}\mu_u^2$, we obtain

$$\mathrm{Var}\mu_u^2 = \frac{n^2}{4}((p_2^u - (p_1^u)^2) + n^{-1}(p_1^u - p_2^u)).$$

Therefore,

$$\mathrm{Var}\left[\log\left(\frac{\|\phi(X_t)\|}{\|\phi(X_s)\|}\right)^2\right] \le (t - s)\left(\frac{1 + \lambda^{-1}}{n} + (1 + \lambda)\Gamma_{s,t}\right),$$

where $\Gamma_{s,t} \stackrel{def}{=} \int_s^t \frac{1}{4}((p_2^u - (p_1^u)^2) + n^{-1}(p_1^u - p_2^u))\,du$. Optimizing over $\lambda$ yields

$$\mathrm{Var}\left[\log\left(\frac{\|\phi(X_t)\|}{\|\phi(X_s)\|}\right)^2\right] \le (t - s)\left(n^{-1/2} + \Gamma_{s,t}^{1/2}\right)^2.$$

The term $\Gamma_{s,t}$ can be shown to have $\mathcal{O}(n^{-1/2})$ asymptotic behaviour using tools from Mckean-Vlasov theory. Thus, the variance term has (atmost) $\mathcal{O}(n^{-1})$ behaviour. $\qquad\square$

# I  Proof of Theorem 2

In this section, we provide the proof of Theorem 2. We use the following Law of Large numbers that does not require independence.

**Theorem 8** (Corollary 3.1 in Sung et al. (2008))**.** *Let* $(Y_n^i)_{1 \leq i \leq n, n \geq 1}$ *be a triangular array of random variables. Assume that the following holds*

- $\sup_{n \geq 1} \frac{1}{n} \sum_{i=1}^n \mathbb{E}|Y_n^i| < \infty.$

- $\lim_{a \to \infty} \sup_{n \geq 1} \frac{1}{n} \sum_{i=1}^n \mathbb{E}|Y_n^i| \mathbb{1}_{\{|Y_n^i| > a\}} = 0.$

*Then, we have that*

$$\frac{1}{n} \left( \sum_{i=1}^n Y_n^i - \zeta_n^i \right) \longrightarrow 0,$$

*where the convergence is in* $L_1$ *and* $\zeta_n^i = \mathbb{E}[Y_n^i | \mathcal{F}_{n,i-1}]$, *with* $\mathcal{F}_{n,j} = \sigma\{Y_n^k, 1 \leq k \leq j\}$, *i.e. the sigma algebra generated by the variables* $\{Y_n^k, 1 \leq k \leq j\}$, *and* $\mathcal{F}_{n,0} = \{\emptyset, \Omega\}$ *by definition.*

Let us now prove our result.

**Theorem 2.**  *For* $0 \leq s \leq t \leq 1$, *we have*

$$\log\left( \frac{\|\phi(X_t)\|}{\|\phi(X_s)\|} \right) \mathbb{1}_{\{\|\phi(X_0)\| > 0\}} \xrightarrow[n \to \infty]{} \frac{t-s}{4}, \quad and, \quad \frac{\|\phi(X_t)\|}{\|\phi(X_s)\|} \mathbb{1}_{\{\|\phi(X_0)\| > 0\}} \xrightarrow[n \to \infty]{} \exp\left( \frac{t-s}{4} \right).$$

*where the convergence holds in* $L_1$.
*Moreover, we have that*

$$\sup_{i \in [n]} \mathbb{E} \left( \sup_{t \in [0,1]} |X_t^i - \tilde{X}_t^i|^2 \right) = \mathcal{O}(n^{-1}),$$

*where* $X_t^i$ *is the solution of the following (Mackean-Vlasov) SDE*

$$d\tilde{X}_t^i = \left( \mathbb{E}\phi(\tilde{X}_t^i)^2 \right)^{1/2} dB_t^i, \quad \tilde{X}_0^i = X_0^i.$$

*As a result, the pre-activations* $Y_{\lfloor tL \rfloor}^i$ *(Eq. (1)) converge in distribution to a Gaussian distribution in the limit infinite-depth-then-infinite-width*

$$\forall i \in [n], \quad Y_{\lfloor tL \rfloor}^i \xrightarrow{L \to \infty \ then \ n \to \infty} \mathcal{N}(0, d^{-1}\|x\|^2 \exp(t/2)).$$

*Proof.* Let $0 \leq s \leq t \leq 1$. From Theorem 1, we have that almost surely

$$\log\left( \frac{\|\phi(X_t)\|}{\|\phi(X_s)\|} \right) = \exp\left( \frac{1}{\sqrt{n}}(\hat{B}_t - \hat{B}_s) + \frac{1}{n} \int_s^t \mu_u du \right).$$

We know that $\frac{1}{\sqrt{n}}(\hat{B}_t - \hat{B}_s)$ converges to zero almost surely (by continuity of Brownian paths) and in $L_1$. Let us now deal with the second term $n^{-1} \int_s^t \mu_u du$. We have that $\frac{1}{n}\mu_u = \frac{1}{2}\frac{1}{n}\sum_{i=1}^n \phi'(X_u^i) - \frac{1}{n}$. Fix $u \in [s, t]$ and let $Z_n^i = \phi'(X_u^i)$ (recall that $X_u^i$ has an implicit dependence on $n$). Since $Z_n^i$ is uniformly bounded across $i$ and $n$, it is straightforward that the conditions of Theorem 8 are satisfied. Therefore, we have the following convergence in $L_1$

$$\frac{1}{n} \left( \sum_{i=1}^n Z_n^i - \zeta_n^i \right) \longrightarrow 0,$$

where $\zeta_n^i = \mathbb{E}[Z_n^i | \mathcal{F}_{n,i-1}]$. Recall from the proof of Theorem 1 that the event $\{X_u^j > 0\}$ is included in the event $\{\|\phi(X_0)\| > 0\}$. Another key observation that will allow us to conclude is that the distribution of $X_u^i$

given $\mathcal{F}_{n,i-1}$ is symmetric around 0 since the dependence is reflected only in the variance of the Brownian motion. Hence, $\zeta_n^i = \frac{1}{2}(1 - 2^{-n})^{-1}$ almost surely. Since $n^{-1} \sum_{i=1}^n \zeta_n^i = \frac{1}{2}(1 - 2^{-n})^{-1} \longrightarrow \frac{1}{2}$, (in $L_1$), then $n^{-1} \sum_{i=1}^n Z_n^i$ converges to $1/2$ in $L_1$. Using the Dominated convergence theorem, we obtain the first result.

Let us now deal with the second result on the absolute growth factor. Let $N > 0$ and define the event

$$E_N = \left\{ \frac{\|\phi(X_t)\|}{\|\phi(X_s)\|} \leq \exp(N) \right\},$$

and let $E_N^c$ be its complementary event. For $N$ large enough, we have that

$$\mathbb{E}\left| \frac{\|\phi(X_t)\|}{\|\phi(X_s)\|} - \exp(t/4) \right| = \mathbb{E}\left| \frac{\|\phi(X_t)\|}{\|\phi(X_s)\|} - \exp((t-s)/4) \right| \mathbb{1}_{E_N} + \mathbb{E}\left| \frac{\|\phi(X_t)\|}{\|\phi(X_s)\|} - \exp((t-s)/4) \right| \mathbb{1}_{E_N^c}$$

$$\leq \exp(N) \times \mathbb{E}\left| \log\left( \frac{\|\phi(X_t)\|}{\|\phi(X_s)\|} \right) - (t-s)/4 \right|$$

$$+ \mathbb{E}\left| \frac{\|\phi(X_t)\|}{\|\phi(X_s)\|} - \exp((t-s)/4) \right| \mathbb{1}_{E_N^c}$$

$$\leq \exp(N) \times \mathbb{E}\left| \log\left( \frac{\|\phi(X_t)\|}{\|\phi(X_s)\|} \right) - (t-s)/4 \right| + K\,\mathbb{P}(E_N^c),$$

where $K$ is a ($t$ dependent) constant and where we have used Theorem 1 obtain that $\mathbb{E}\left| \frac{\|\phi(X_t)\|}{\|\phi(X_s)\|} \right|$ is finite. Taking $n$ to infinity in the inequality above, we obtain that for $N$ large enough

$$\limsup_{n \to \infty} \mathbb{E}\left| \frac{\|\phi(X_t)\|}{\|\phi(X_s)\|} - \exp(t/4) \right| \leq K\,\mathbb{P}(E_N^c)$$

$$\leq N^{-1} K\,\mathbb{E}\left| \log\left( \frac{\|\phi(X_t)\|}{\|\phi(X_s)\|} \right) \right|,$$

where we have used Markov's inequality. Since this is true for all $N$ large enough, we conclude that $\lim_{n \to \infty} \mathbb{E}\left| \frac{\|\phi(X_t)\|}{\|\phi(X_s)\|} - \exp((t-s)/4) \right| = 0$.

The convergence to Mckean-Vlasov dynamics is straightforward from Theorem 6, and the Gaussian distribution is given by Lemma 16.

$\square$

## I.1 Some technical lemmas

**Lemma 16.** *Let $x \in \mathbb{R}^d$ such that $x \neq 0$, $m \geq 1$ be an integer, and consider the real-valued (Mckean-Vlasov) stochastic process $\tilde{X}$ given by*

$$d\tilde{X}_t = \left( \mathbb{E}\phi(\tilde{X}_t)^2 \right)^{1/2} dB_t, \quad t \in [0, \infty), \quad \tilde{X}_0 = \tilde{W}_{in}^\top x,$$

*where $\phi$ is the ReLU activation function, $(B_t)_{t \geq 0}$ is a one-dimensional Brownian motion, and $\tilde{W}_{in} \sim \mathcal{N}(0, d^{-1}I)$. We have the following*

$$\forall t \geq 0, X_t \sim \mathcal{N}(0, d^{-1}\|x\|^2 \exp(t/2)).$$

*Proof.* Let $t > 0$. From the SDE, it is clear that $\tilde{X}_t$ is Gaussian with zero mean and variance $\int_0^t \|\phi(\tilde{X}_s)\|_{L_2} ds$ (by Itô isometry). Therefore, since ReLU is homogeneous, it is straightforward that for all $s > 0$, $\|\phi(\tilde{X}_s)\|_{L_2}^2 =$

$\frac{1}{2}\|\tilde{X}_s\|_{L_2}^2$. Using Itô 's lemma, we obtain

$$d\tilde{X}_t^2 = 2\tilde{X}_t d\tilde{X}_t + \frac{1}{2}\|\tilde{X}_t\|_{L_2}^2 dt.$$

Taking the expectation[15] yields the following ordinary differential equation

$$d\|\tilde{X}_t\|_{L_2}^2 = \frac{1}{2}\|\tilde{X}_t\|_{L_2}^2 dt,$$

which has a closed-form solution given by

$$\|\tilde{X}_t\|_{L_2}^2 = \|\tilde{X}_0\|_{L_2}^2 \exp(t/2).$$

We conclude by observing that $\|\tilde{X}_0\|_{L_2}^2 = \mathbb{E}\tilde{X}_0^2 = d^{-1}\|x\|^2$. $\qquad\square$

**Lemma 17.** *Let $x \in \mathbb{R}^d$ such that $x \neq 0$, $m \geq 1$ be an integer, and consider the two real-valued (Mckean-Vlasov) stochastic processes $\tilde{X}^m$ and $\tilde{X}$ given by*

$$\begin{cases} d\tilde{X}_t^m = \left(\mathbb{E}\phi_m(\tilde{X}_t^m)^2\right)^{1/2} dB_t, & t \in [0,\infty), \quad \tilde{X}_0^m = \tilde{W}_{in}^\top x, \\ d\tilde{X}_t = \left(\mathbb{E}\phi(\tilde{X}_t)^2\right)^{1/2} dB_t, & t \in [0,\infty), \quad X_0 = \tilde{W}_{in}^\top x, \end{cases}$$

*where $\phi_m(z) = \int_0^z h(mu)du$ where $h$ is the Sigmoid function given by $h(u) = (1 + e^{-u})^{-1}$, $\phi$ is the ReLU activation function, $(B_t)_{t\geq 0}$ is a one-dimensional Brownian motion, and $\tilde{W}_{in} \sim \mathcal{N}(0, d^{-1}I)$. We have the following*

$$\forall t \geq 0, \ \mathbb{E}|\tilde{X}_t^m - \tilde{X}_t|^2 \leq \frac{2t}{m^2}e^{2t}.$$

*Proof.* The proof of Lemma 17 is similar to that of Lemma 8 with the only difference of replacing the euclidean norm with the $L_2$ norm in probability space. Let $t \geq 0$, we have that

$$\begin{aligned} \mathbb{E}|\tilde{X}_t^m - \tilde{X}_t|^2 &= \mathbb{E}\left(\int_0^t (\|\phi_m(\tilde{X}_s^m)\|_{L_2} - \|\phi(\tilde{X}_s)\|_{L_2})dB_s\right)^2 \\ &= \int_0^t (\|\phi_m(\tilde{X}_s^m)\|_{L_2} - \|\phi(\tilde{X}_s)\|_{L_2})^2 ds \\ &\leq \int_0^t \|\phi_m(\tilde{X}_s^m) - \phi(\tilde{X}_s)\|_{L_2}^2 ds \\ &\leq \frac{2t}{m^2} + 2\int_0^t \|\tilde{X}_s^m - \tilde{X}_s\|_{L_2}^2 ds, \end{aligned}$$

where we have used the triangular inequality and Lemma 9. We conclude using Gronwall's lemma. $\qquad\square$

## J   Proof of Theorem 3

**Theorem 3.**   *Let $t \in [0,1]$. Then, in the limit $\lim_{L\to\infty}\lim_{n\to\infty}$ (infinite width, then infinite depth), we have that*

$$\frac{\|\phi(Y_{\lfloor tL \rfloor})\|}{\|\phi(Y_0)\|}\mathbb{1}_{\{\|\phi(Y_0)\|>0\}} \longrightarrow \exp\left(\frac{t}{4}\right),$$

*where the convergence holds in probability.*

---

[15]This should be understood as integrating the SDE, then taking the expectation, then differentiating once again.

*Moreover, the pre-activations $Y^i_{\lfloor tL \rfloor}$ (Eq. (1)) converge in distribution to a Gaussian distribution in the limit infinite-width-then-infinite-depth*

$$\forall i \in [n], \quad Y^i_{\lfloor tL \rfloor} \xrightarrow{n \to \infty \ then \ L \to \infty} \mathcal{N}(0, d^{-1}\|x\|^2 \exp(t/2)).$$

*Proof.* Let $t \in [0,1]$. It is straightforward that $\lim_{n \to \infty} \mathbb{1}_{\{\|\phi(Y_0)\|>0\}} = 1$ almost surely. Moreover, we have that for all $t \in [0,1]$, $n^{-1}\|\phi(Y_{\lfloor tL \rfloor})\|^2$ converges in distribution to $\mathbb{E}\phi(Y^1_{\lfloor tL \rfloor})^2$ when $n$ goes to infinity (Yang, 2020; Hayou et al., 2021; Matthews et al., 2018). Since the limiting value is constant, then the convergence holds also in probability. Now let $\epsilon > 0$. We have that

$$\mathbb{P}\left(\left|\frac{\|\phi(Y_{\lfloor tL \rfloor})\|}{\|\phi(Y_0)\|} \mathbb{1}_{\{\|\phi(Y_0)\|>0\}} - \exp\left(\frac{t}{4}\right)\right| > \epsilon\right) \leq \mathbb{P}\left(\frac{\|\phi(Y_{\lfloor tL \rfloor})\|}{\|\phi(Y_0)\|} \mathbb{1}_{\{\|\phi(Y_0)\|>0\}} - \exp\left(\frac{t}{4}\right) > \epsilon\right)$$
$$+ \mathbb{P}\left(\frac{\|\phi(Y_{\lfloor tL \rfloor})\|}{\|\phi(Y_0)\|} \mathbb{1}_{\{\|\phi(Y_0)\|>0\}} - \exp\left(\frac{t}{4}\right) < -\epsilon\right).$$

Let us show that the first term in the right-hand side converges to 0 in the sequential limit 'infinite width then infinite depth'. The proof is similar for the second term. We have that

$$\mathbb{P}\left(\frac{\|\phi(Y_{\lfloor tL \rfloor})\|}{\|\phi(Y_0)\|} \mathbb{1}_{\{\|\phi(Y_0)\|>0\}} - e^{\frac{t}{4}} > \epsilon\right) \leq \mathbb{P}\left(\frac{\|\phi(Y_{\lfloor tL \rfloor})\|}{\|\phi(Y_0)\|} \mathbb{1}_{\{\|\phi(Y_0)\|>0\}} - \frac{\sqrt{\mathbb{E}\phi(Y^1_{\lfloor tL \rfloor})^2}}{\sqrt{\mathbb{E}\phi(Y^1_0)^2}} > \epsilon/2\right)$$
$$+ \mathbb{1}\left(\frac{\sqrt{\mathbb{E}\phi(Y^1_{\lfloor tL \rfloor})^2}}{\sqrt{\mathbb{E}\phi(Y^1_0)^2}} - \exp\left(\frac{t}{4}\right) > \epsilon/2\right),$$

where $\mathbb{1}(z > \epsilon/2) \overset{def}{=} \mathbb{1}_{\{z > \epsilon/2\}}$ (to alleviate the notation). Using the convergence in probability of $n^{-1}\|\phi(Y_{\lfloor tL \rfloor})\|^2$ to $\mathbb{E}\phi(Y^1_{\lfloor tL \rfloor})^2$, we obtain for all $L$

$$\lim_{n \to \infty} \mathbb{P}\left(\frac{\|\phi(Y_{\lfloor tL \rfloor})\|}{\|\phi(Y_0)\|} \mathbb{1}_{\{\|\phi(Y_0)\|>0\}} - \exp\left(\frac{t}{4}\right) > \epsilon\right) \leq \mathbb{1}\left(\frac{\sqrt{\mathbb{E}\phi(Y^1_{\lfloor tL \rfloor})^2}}{\sqrt{\mathbb{E}\phi(Y^1_0)^2}} - \exp\left(\frac{t}{4}\right) > \epsilon/2\right).$$

Using Lemma 5 in Hayou et al. (2021), and the homogenous property of ReLU, we have that

$$\lim_{L \to \infty} \mathbb{E}\phi(Y^1_{\lfloor tL \rfloor})^2 = \frac{1}{2}q_t,$$

where $q_t : [0,1] \to \mathbb{R}^+$ is the solution of the ordinary differential equation $q'_t = \frac{1}{2}q_t$, which has a unique solution given by $q_t = q_0 \exp(t/2)$. Dividing by $q_0$ and taking the square root, and taking $L$ to infinity, we obtain the desired result.

Regarding the convergence in distribution of the pre-activations, in the limit $n \to \infty$, the pre-activations become Gaussian with zero mean and variance $\mathbb{E}(Y^1_{\lfloor tL \rfloor})^2$. This variance converges to $q_t$ given above in the limit $L \to \infty$. The conclusion is straightforward using Slutsky's lemma. $\square$

## K   Piece-wise linear activation functions

We have seen in Section 4 that the distribution of $X_t$ is generally intractable for $n \geq 2$. This is purely due to *finite width* $n \geq 2$ and not to the non-linearity of the activation function. To understand this, let us see what happens when the activation function is the identity function. In this case the process $X_t$ is solution of the following SDE

$$dX_t = \frac{1}{\sqrt{n}}\|X_t\|dB_t. \tag{16}$$

When $n = 1$, the SDE Eq. (16) has a closed-form solution given by the (conditional) GBM distribution (Proposition 2). For general $n \geq 2$, the entries of $X_t$ are dependent and the resulting dynamics (generally) do not admit closed-form solutions. However, we can obtain closed-form solutions for the norm $\|X_t\|$. Indeed, a simple application of Itô 's lemma yields the following results.

**Theorem 9** (Norms with the identity activation)**.** *With the linear activation, we have that for all $t \in [0, 1]$,*

$$\|X_t\| = \|X_0\| \exp\left(\frac{1}{\sqrt{n}}\hat{B}_t + \left(\frac{1}{2} - \frac{1}{n}\right)t\right), \quad almost\ surely,$$

*where $(\hat{B})_{t \geq 0}$ is a one-dimensional Brownian motion. As a result, we have that for all $0 \leq s \leq t \leq 1$*

$$\mathbb{E}\left[\log\left(\frac{\|X_t\|}{\|X_s\|}\right)\right] = \left(\frac{1}{2} - \frac{1}{n}\right)(t - s).$$

The proof of Theorem 9 is straightforward using Itô 's lemma. We omit the proof here.

**Role of the non-linearity.**     By comparing the result of Theorem 1 and Theorem 9, we observe some differences between the case of ReLU and that of the identity activation function. With ReLU, the drift term in $\log(\|\phi(X_t)\|/\|\phi(X_s)\|)$ is given by $\frac{1}{n}\int_0^t \mu_s ds$ which is a stochastic term with mean given by $\left(\frac{1-2^{-n}}{4} - \frac{1}{n}\right)t$. With the identity activation, this drift term is *deterministic* and is equal to $\left(\frac{1}{2} - \frac{1}{n}\right)t$. This allows to conclude the following:

- *Non-linearity induces stochastic drift:* the non-linearity of ReLU induces stochasticity in the drift term of $\log(\|X_t\|/\|X_0\|)$, which results in the Quasi-GBM dynamics given by Theorem 1.

- *Non-linearity induces change of regime:*    with ReLU, the mean drift of $\log(\|\phi(X_t)\|/\|\phi(X_0)\|)$ is given by $\left(\frac{1-2^{-n}}{4} - \frac{1}{n}\right)t$ which is negative for $n = 1, 2, 3$. This induces the change of regime we discussed after Theorem 1 (having a negative mean drift implies that there is a significant mass of the distribution of $\|X_t\|/\|X_0\|$ in the regime $(0, 1)$). With the identity activation function, the drift term is always non-negative for $n \geq 2$, and negative for $n = 1$. Thus, the change of regime cover some values $n \geq 2$ only when there is a non-linearity. We give more details about this observation in the next result.

To capture the effect of non-linearity in the regime change phenomenon discussed above, we study the dynamics of the post-norm activation for a special class of piece-wise linear activations that include both ReLU and the identity function. The result of Theorem 1 can be easily extended to the case of general piece-wise linear activation functions using the same proof techniques. We obtain the following result which generalizes that of Theorem 1 and Theorem 9.

**Theorem 10** (Post-activations norm for piece-wise linear activations)**.** *Let $\alpha, \beta \in \mathbb{R}$, and let $\phi_{\alpha,\beta}$ be the activation function given by $\phi_{\alpha,\beta}(z) = \alpha ReLU(z) + \beta ReLU(z)$. We have that for all $t \in [0, 1]$,*

$$\|\phi_{\alpha,\beta}(X_t)\| = \|\phi_{\alpha,\beta}(X_0)\| \exp\left(\frac{1}{\sqrt{n}}\hat{B}_t + \frac{1}{n}\int_0^t \mu_s^{\alpha,\beta}ds\right), \quad almost\ surely,$$

*where $\mu_s^{\alpha,\beta} = \frac{1}{2}\sum_{i=1}^n (\alpha^2 \mathbb{1}_{x_i \geq 0} + \beta^2 \mathbb{1}_{x_i < 0}) - 1$, and $(\hat{B})_{t \geq 0}$ is a one-dimensional Brownian motion. As a result, we have that for all $0 \leq s \leq t \leq 1$*

- *if $\alpha = 0, \beta \neq 0$*

$$\mathbb{E}\left[\log\left(\frac{\|\phi_{0,\beta}(X_t)\|}{\|\phi_{0,\beta}(X_s)\|}\right) \mid \|\phi_{0,\beta}(X_0)\| > 0\right] = \left(\frac{\beta^2(1 - 2^{-n})}{4} - \frac{1}{n}\right)(t - s).$$

- *if $\alpha \neq 0, \beta = 0$*

$$\mathbb{E}\left[\log\left(\frac{\|\phi_{\alpha,0}(X_t)\|}{\|\phi_{\alpha,0}(X_s)\|}\right) \mid \|\phi_{\alpha,0}(X_0)\| > 0\right] = \left(\frac{\alpha^2(1 - 2^{-n})}{4} - \frac{1}{n}\right)(t - s).$$

- *if $\alpha \neq 0, \beta \neq 0$*

$$\mathbb{E}\left[\log\left(\frac{\|\phi_{\alpha,\beta}(X_t)\|}{\|\phi_{\alpha,\beta}(X_s)\|}\right)\right] = \left(\frac{\alpha^2 + \beta^2}{4} - \frac{1}{n}\right)(t - s).$$

Theorem 10 generalizes that of ReLU (Theorem 1, $\alpha = 1, \beta = 0$) and the identity activation (Theorem 9, $\alpha = -\beta = 1$). The discontinuity of the mean of $\log\left(\frac{\|\phi_{\alpha,\beta}(X_t)\|}{\|\phi_{\alpha,\beta}(X_s)\|}\right)$ at the poles $\alpha = 0$ (and $\beta \neq 0$) and $\beta = 0$ (and $\alpha \neq 0$) is due to the fact that the event $\{\|\phi_{\alpha,\beta}(X_0)\| > 0\}$ has non-zero probability in these cases and zero probability when $\alpha \neq 0$ and $\beta \neq 0$.

**Perturbation analysis around the identity function.** Consider the case when $\alpha = 1$ and $\beta = 1 - \varepsilon$ for some $\varepsilon \ll 1$. The mean logarithmic growth factor is given by

$$G^n_{s,t} = \left(\frac{1 + (1 - \varepsilon)^2}{4} - \frac{1}{n}\right)(t - s) \approx \left(\frac{1 - \varepsilon}{2} - \frac{1}{n}\right)(t - s).$$

Observe that for $\varepsilon = 0$, we recover the result of Theorem 9 (identity activation). Hence, a small perturbation of the identity function has the effect of decreasing the factor $G^n_{s,t}$ which results in having negative values for $G^n_{s,t}$ for certain values of $n$. Indeed, by fixing $\alpha = 1$, notice that the minimum values of $G^n_{s,t}$ is obtained when $\beta \approx 0$, for which $\phi_{1,0} = \text{ReLU}$. Notice that we can also control the change of regime by tuning the parameter $\alpha$. This allows us to control the sign of $G^n_{s,t}$ for any $n$ by tuning the parameter $\alpha$. We leave the analysis of the practical implications of tuning $\alpha$ for future work.

# L  Additional Experiments

## L.1  Geometric Brownian motion

Additional histograms of $Y_L$ and $\log(Y_l)$ (Proposition 2) are shown in Fig. 9 and Fig. 10.

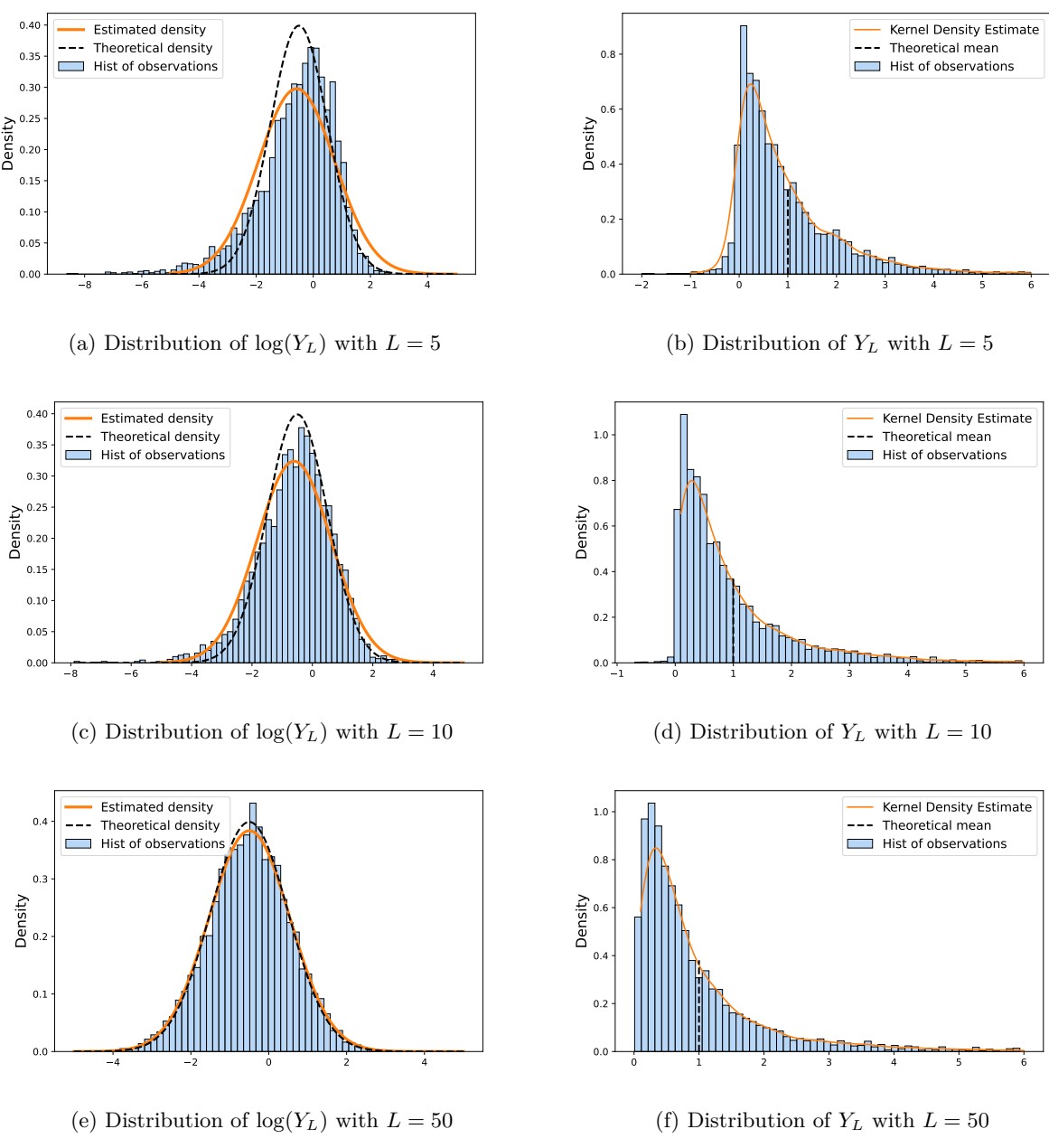

(a) Distribution of $\log(Y_L)$ with $L = 5$

(b) Distribution of $Y_L$ with $L = 5$

(c) Distribution of $\log(Y_L)$ with $L = 10$

(d) Distribution of $Y_L$ with $L = 10$

(e) Distribution of $\log(Y_L)$ with $L = 50$

(f) Distribution of $Y_L$ with $L = 50$

Figure 9: Empirical verification of Proposition 2. **(a), (c), (e)** Histograms of $\log(Y_L)$ and based on $N = 5000$ simulations for depths $L \in \{5, 10, 50\}$ with $Y_0 = 1$. Estimated density (Gaussian kernel estimate) and theoretical density (Gaussian) are illustrated on the same graphs. **(b), (d), (f)** Histograms of $Y_L$ based on $N = 5000$ simulations for depths $L \in \{5, 10, 50\}$ with $Y_0 = 1$.

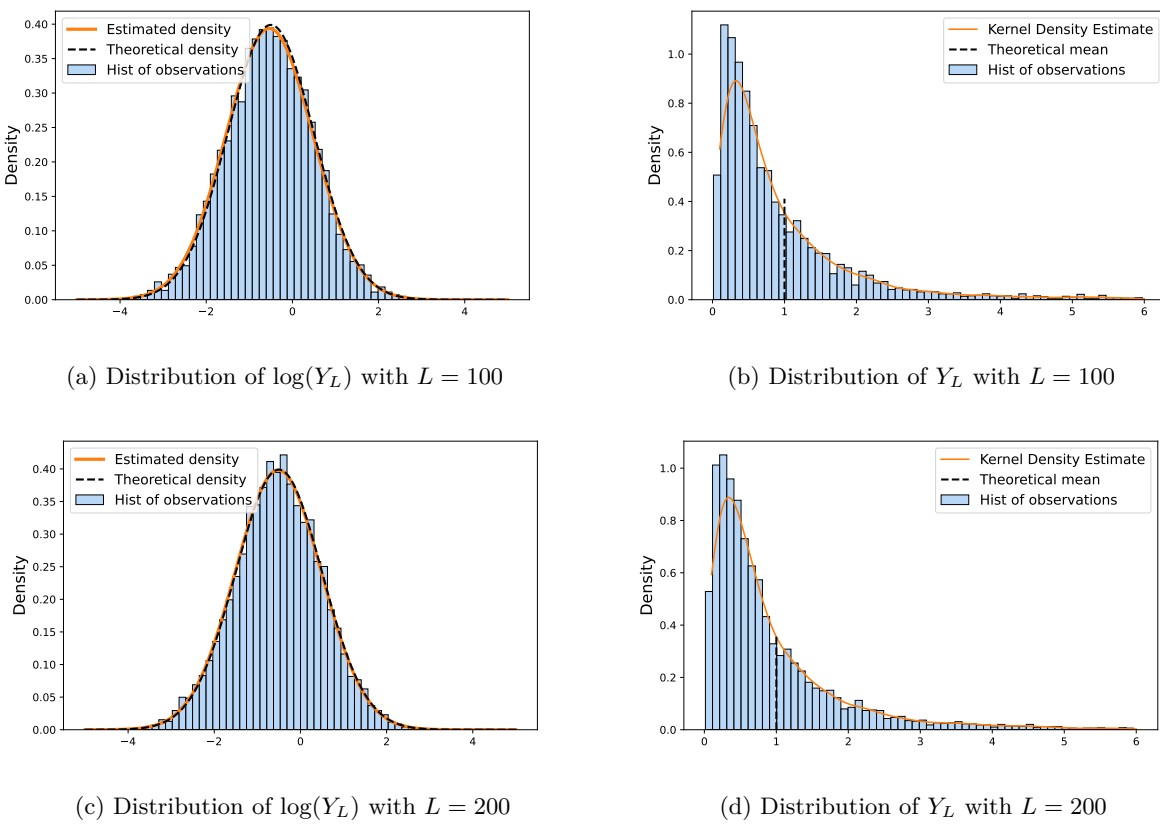

(a) Distribution of $\log(Y_L)$ with $L = 100$

(b) Distribution of $Y_L$ with $L = 100$

(c) Distribution of $\log(Y_L)$ with $L = 200$

(d) Distribution of $Y_L$ with $L = 200$

Figure 10: Empirical verification of Proposition 2. **(a), (c)** Histograms of $\log(Y_L)$ and based on $N = 5000$ simulations for depths $L \in \{100, 200\}$ with $Y_0 = 1$. Estimated density (Gaussian kernel estimate) and theoretical density (Gaussian) are illustrated on the same graphs. **(b), (d)** Histograms of $Y_L$ based on $N = 5000$ simulations for depths $L \in \{100, 200\}$ with $Y_0 = 1$.

## L.2 Ornstein-Uhlenbeck process

Additional histograms of $Y_L$ and $g(Y_l)$ (Proposition 4) are shown in Fig. 9 and Fig. 10.

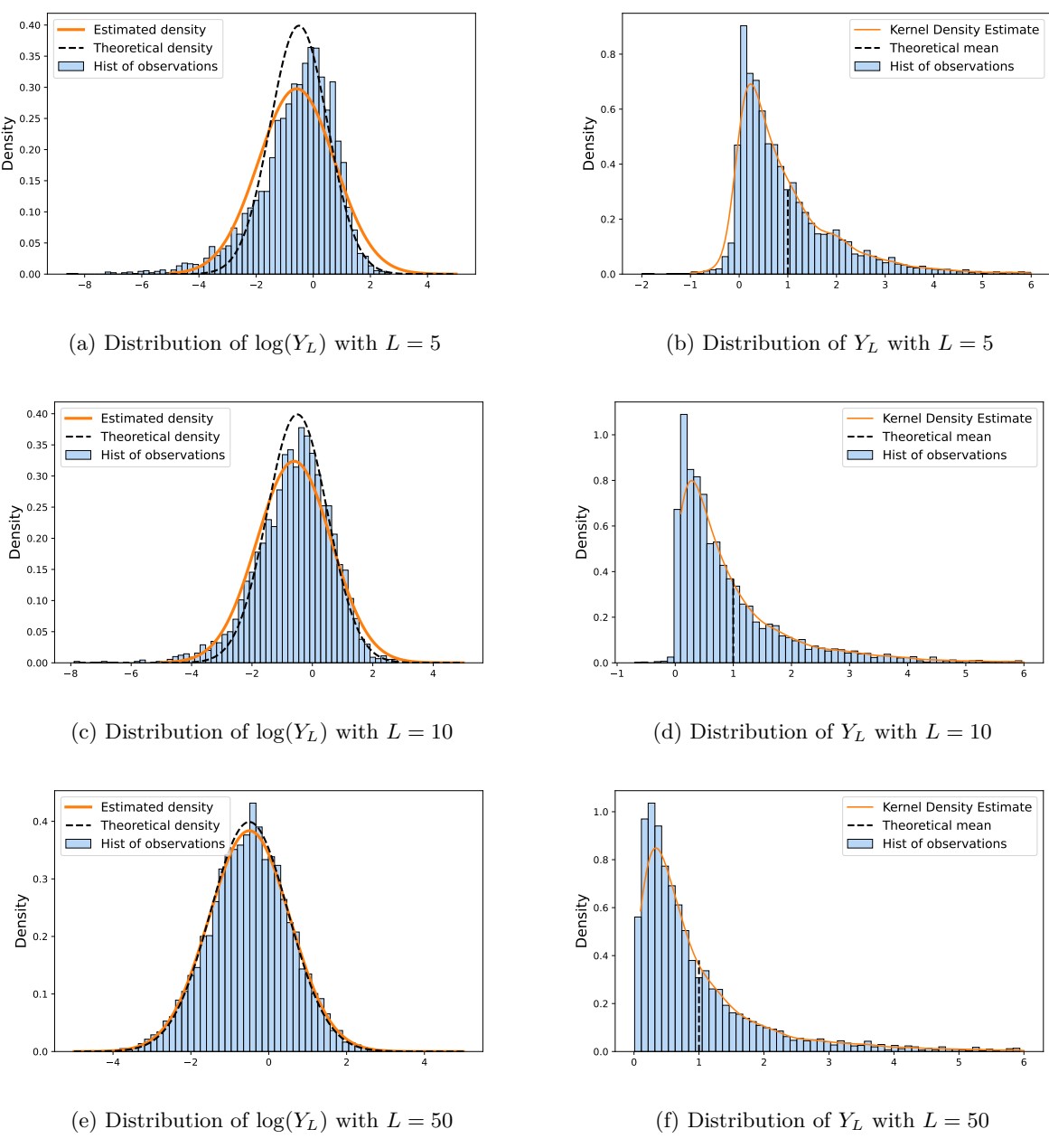

(a) Distribution of $\log(Y_L)$ with $L = 5$

(b) Distribution of $Y_L$ with $L = 5$

(c) Distribution of $\log(Y_L)$ with $L = 10$

(d) Distribution of $Y_L$ with $L = 10$

(e) Distribution of $\log(Y_L)$ with $L = 50$

(f) Distribution of $Y_L$ with $L = 50$

Figure 11: Empirical verification of Proposition 2. **(a), (c), (e)** Histograms of $\log(Y_L)$ and based on $N = 5000$ simulations for depths $L \in \{5, 10, 50\}$ with $Y_0 = 1$. Estimated density (Gaussian kernel estimate) and theoretical density (Gaussian) are illustrated on the same graphs. **(b), (d), (f)** Histograms of $Y_L$ based on $N = 5000$ simulations for depths $L \in \{5, 10, 50\}$ with $Y_0 = 1$.

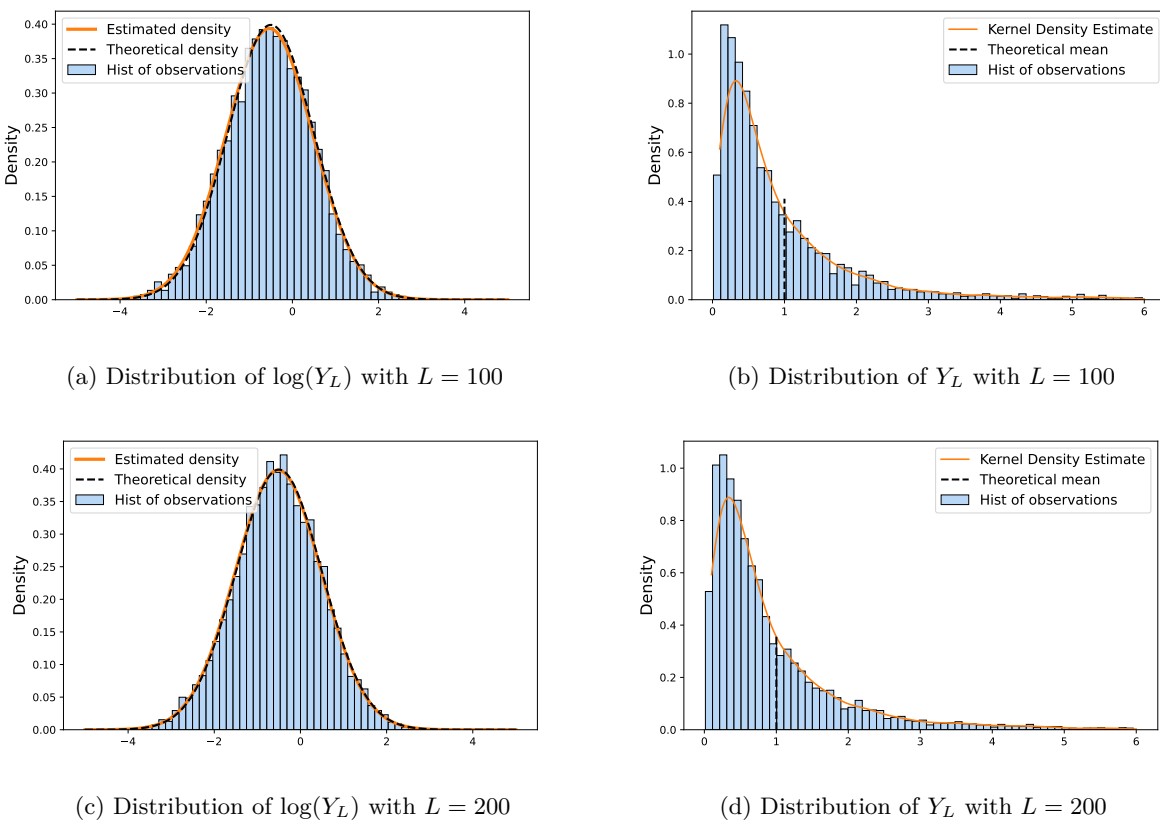

(a) Distribution of $\log(Y_L)$ with $L = 100$

(b) Distribution of $Y_L$ with $L = 100$

(c) Distribution of $\log(Y_L)$ with $L = 200$

(d) Distribution of $Y_L$ with $L = 200$

Figure 12: Empirical verification of Proposition 2. **(a), (c)** Histograms of $\log(Y_L)$ and based on $N = 5000$ simulations for depths $L \in \{100, 200\}$ with $Y_0 = 1$. Estimated density (Gaussian kernel estimate) and theoretical density (Gaussian) are illustrated on the same graphs. **(b), (d)** Histograms of $Y_L$ based on $N = 5000$ simulations for depths $L \in \{100, 200\}$ with $Y_0 = 1$.

### L.3 Histograms of non-scaled log-norm of post-activations

In Fig. 13, we show the histogram of $\log(\|\phi(Y_L)\|/\|\phi(Y_0)\|)$ based on $N = 5000$ simulations. We observe that as the width $n$ increases, the Gaussian approximate is no longer accurate, which is due to the fact that $\|\phi(Y_L)\|/\|\phi(Y_0)\|$ converges to a deterministic value (Theorem 2).

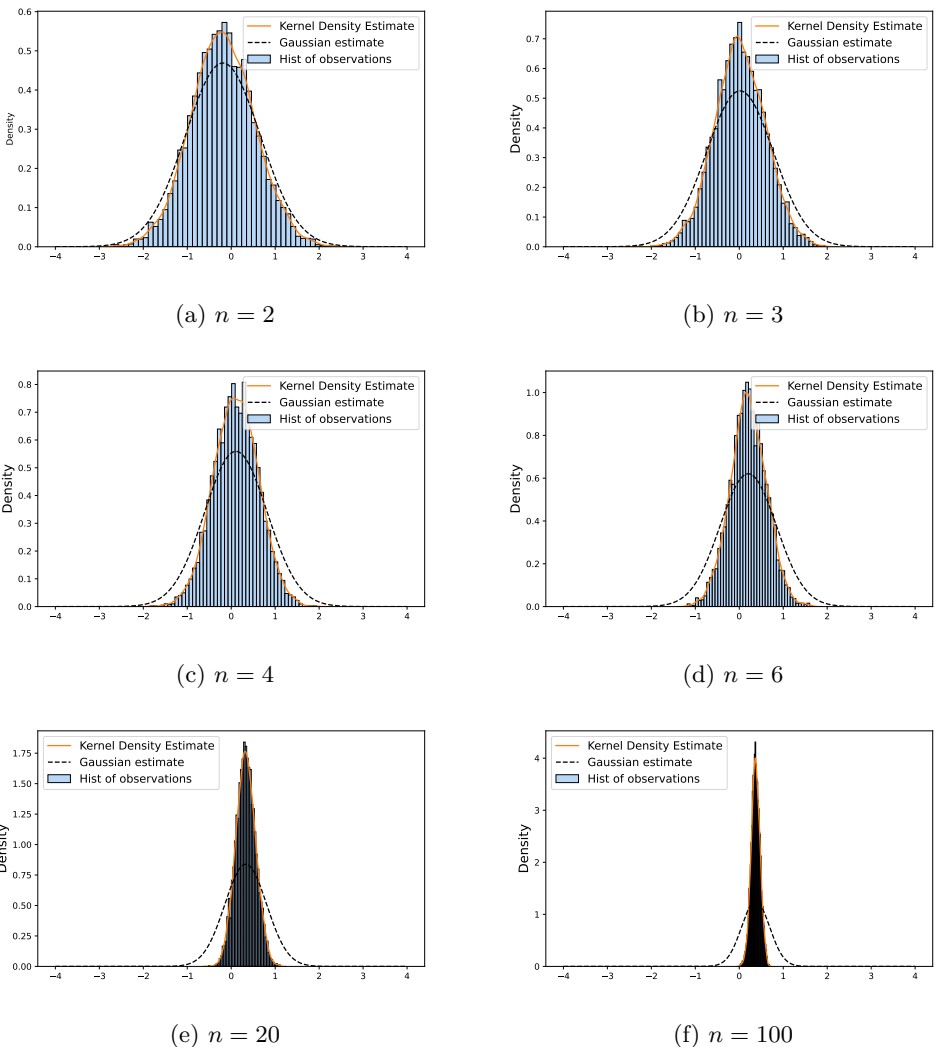

(a) $n = 2$                (b) $n = 3$

(c) $n = 4$                (d) $n = 6$

(e) $n = 20$               (f) $n = 100$

Figure 13: Histogram of $\log(\|\phi(Y_L)\|/\|\phi(Y_0)\|)$ for depth $L = 100$ and different widths $n \in \{2, 3, 4, 6, 20, 100\}$. Gaussian density estimate and (Gaussian) kernel density estimate are shown. As the width increases, we observe a deterioration of the match between the best Gaussian estimate and the empirical distribution. This is due to the fact that the norm of the post-activations concentrates around a deterministic value when $n$ goes to infinity (Theorem 2).

## L.4    Evolution of $\sqrt{n}\log(\|\phi(Y_l)\|/\|\phi(Y_0)\|)$.

In Fig. 14, Fig. 15, Fig. 16, and Fig. 17, we show the histograms of $\sqrt{n}\log(\|\phi(Y_l)\|/\|\phi(Y_0)\|)$ for depth $L = 100$, hidden layers $l \in \{10, 30, 40, 60, 70, 90\}$, and widths $n \in \{2, 3, 20, 100\}$. We observe that Gaussian distribution fits better the last layers. This was expected since the limiting distribution (Quasi-GBM) given in Theorem 1 is only valid for layer indices $\lfloor tL \rfloor$ when $L$ goes to infinity. Thus, for small $l$, it should be expected that the Gaussian distribution would not be a good approximation.

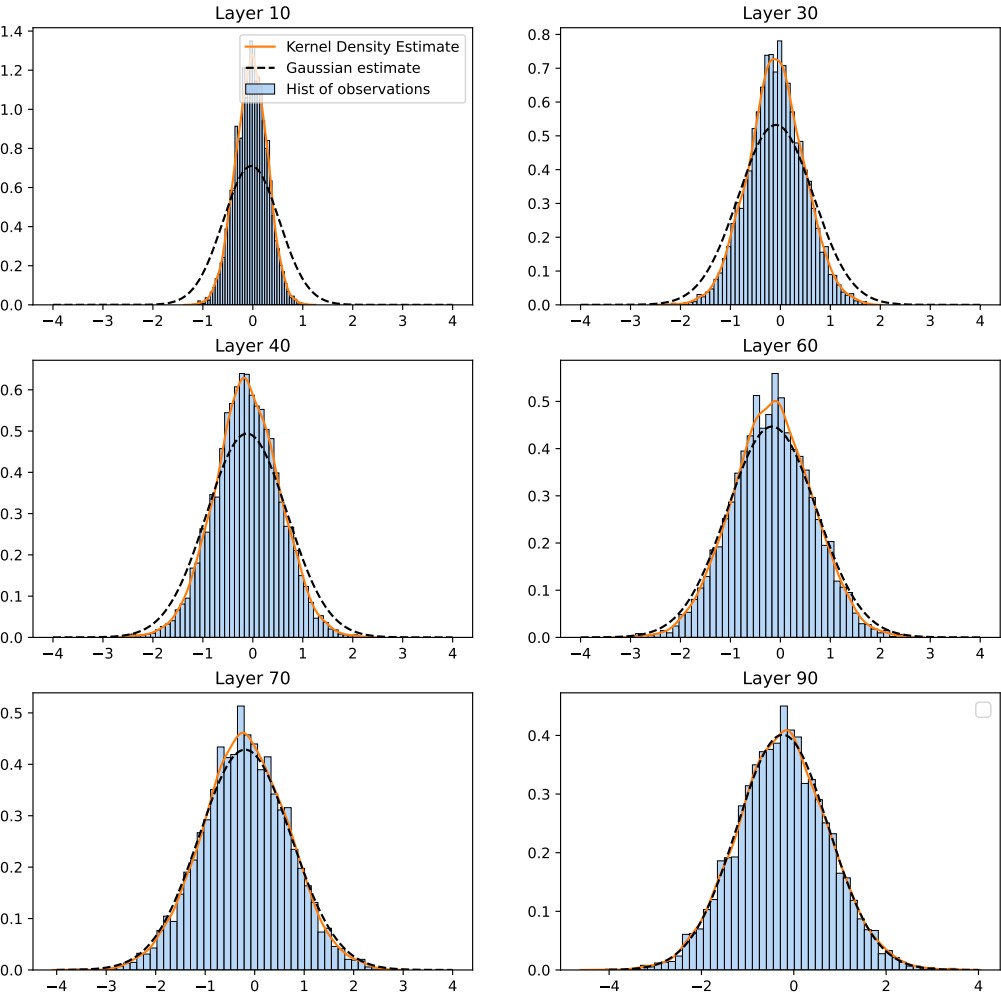

Figure 14: Distribution of $\sqrt{n}\log(\|\phi(Y_l)\|/\|\phi(Y_0)\|)$ (Eq. (1)) for different layer indices, with depth $L = 100$ and width $n = 2$

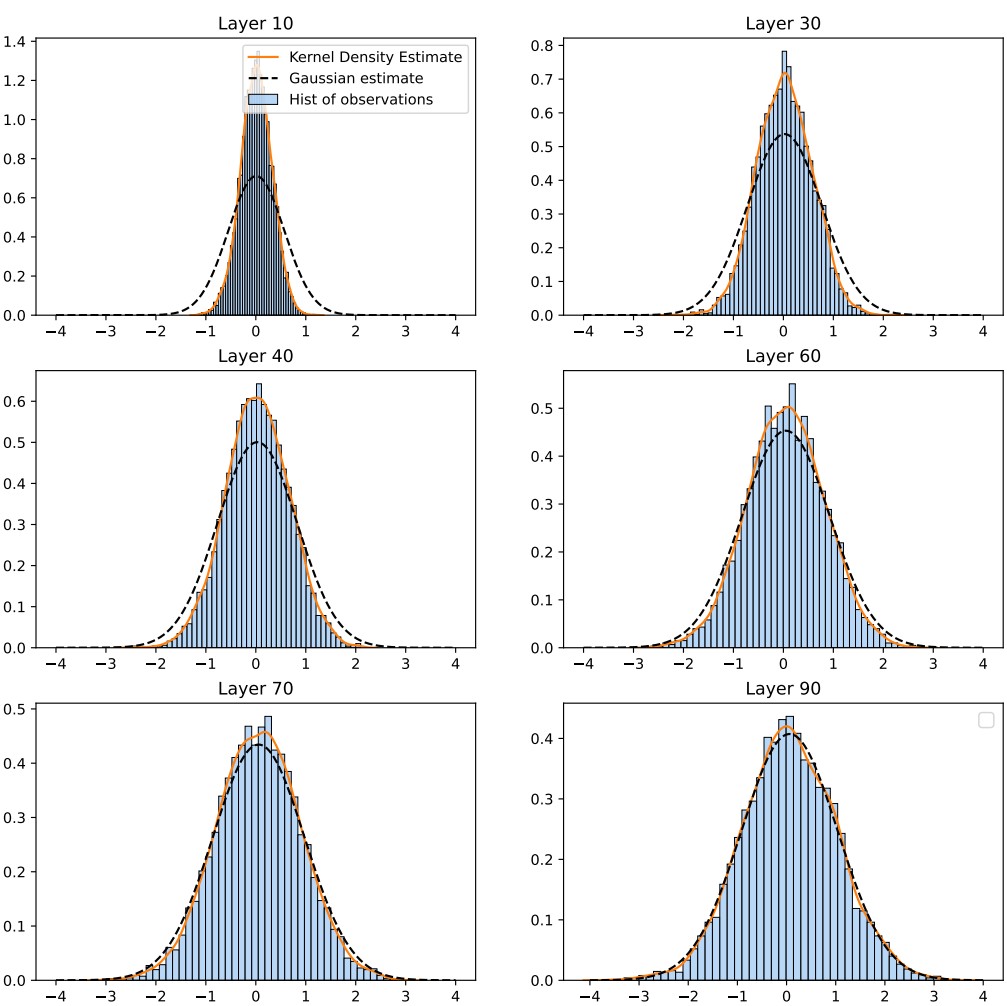

Figure 15: Distribution of $\sqrt{n}\log(\|\phi(Y_l)\|/\|\phi(Y_0)\|)$ (Eq. (1)) for different layer indices, with depth $L = 100$ and width $n = 3$

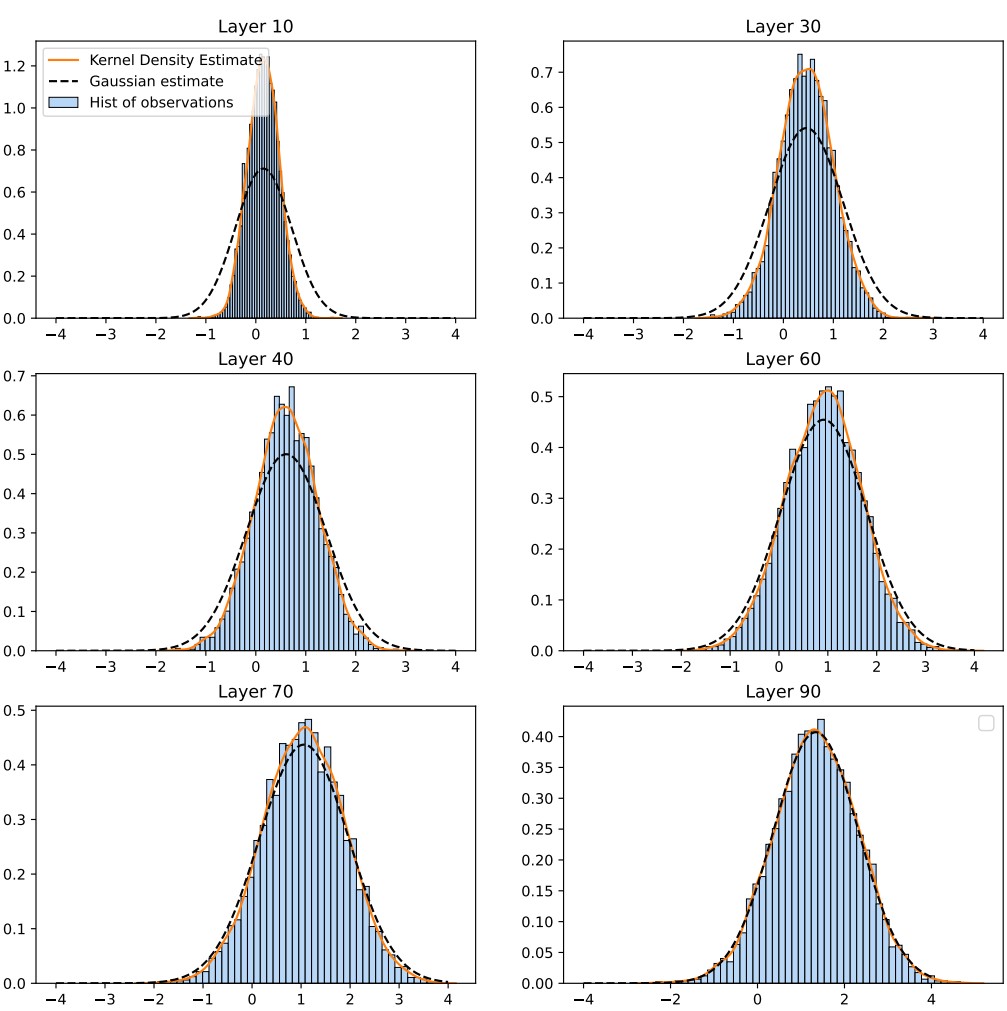

Figure 16: Distribution of $\sqrt{n}\log(\|\phi(Y_l)\|/\|\phi(Y_0)\|)$ (Eq. (1)) for different layer indices, with depth $L = 100$ and width $n = 20$

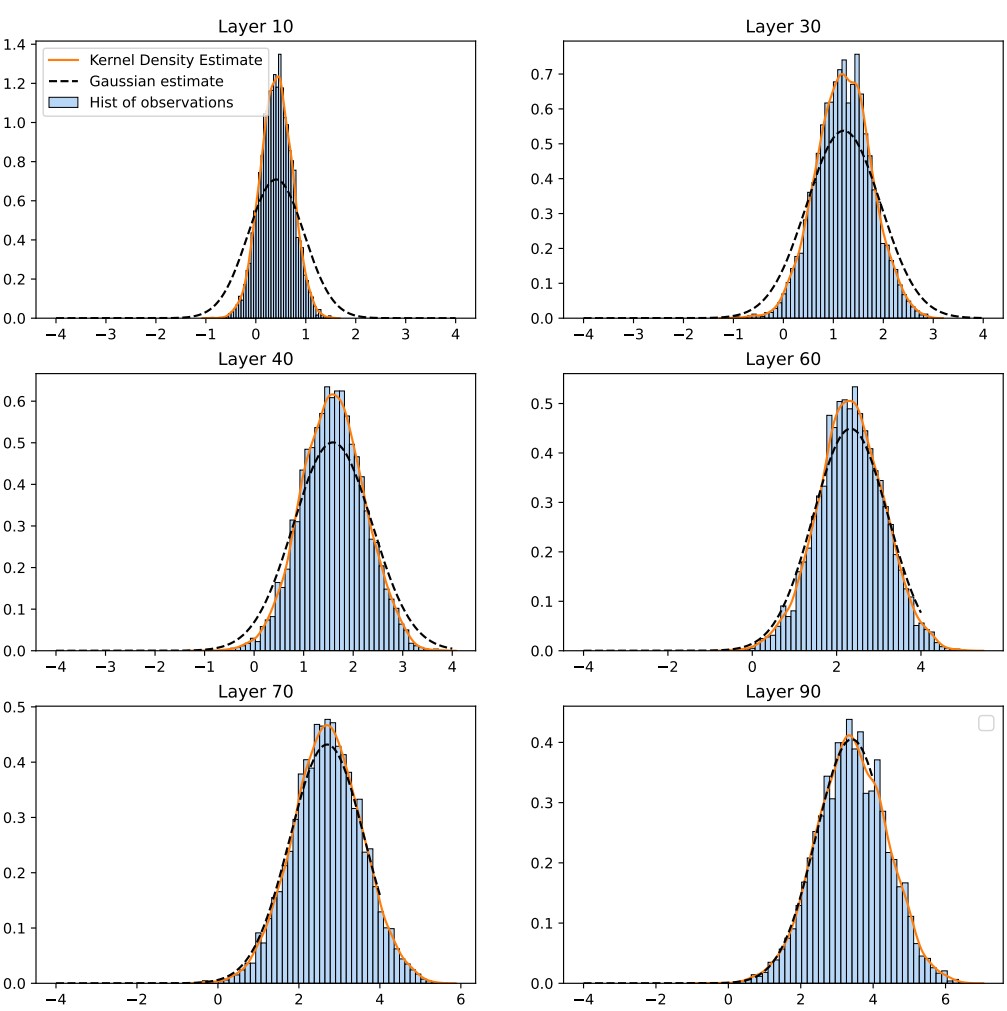

Figure 17: Distribution of $\sqrt{n} \log(\|\phi(Y_l)\|/\|\phi(Y_0)\|)$ (Eq. (1)) for different layer indices, with depth $L = 100$ and width $n = 100$

### L.5 Evolution of $\log(\|\phi(Y_l)\|/\|\phi(Y_0)\|)$ (non-scaled).

In Fig. 18, Fig. 19, Fig. 20, and Fig. 21, we show the non-scaled versions of the histograms from the previous section. We observe that the histogram concentrates around a single value (the distribution converges to a Dirac mass) as $n$ increases. This is a result of the asymptotic behaviour of the ResNet in the infinite-depth-then-infinite-width limit as shown in Theorem 2.

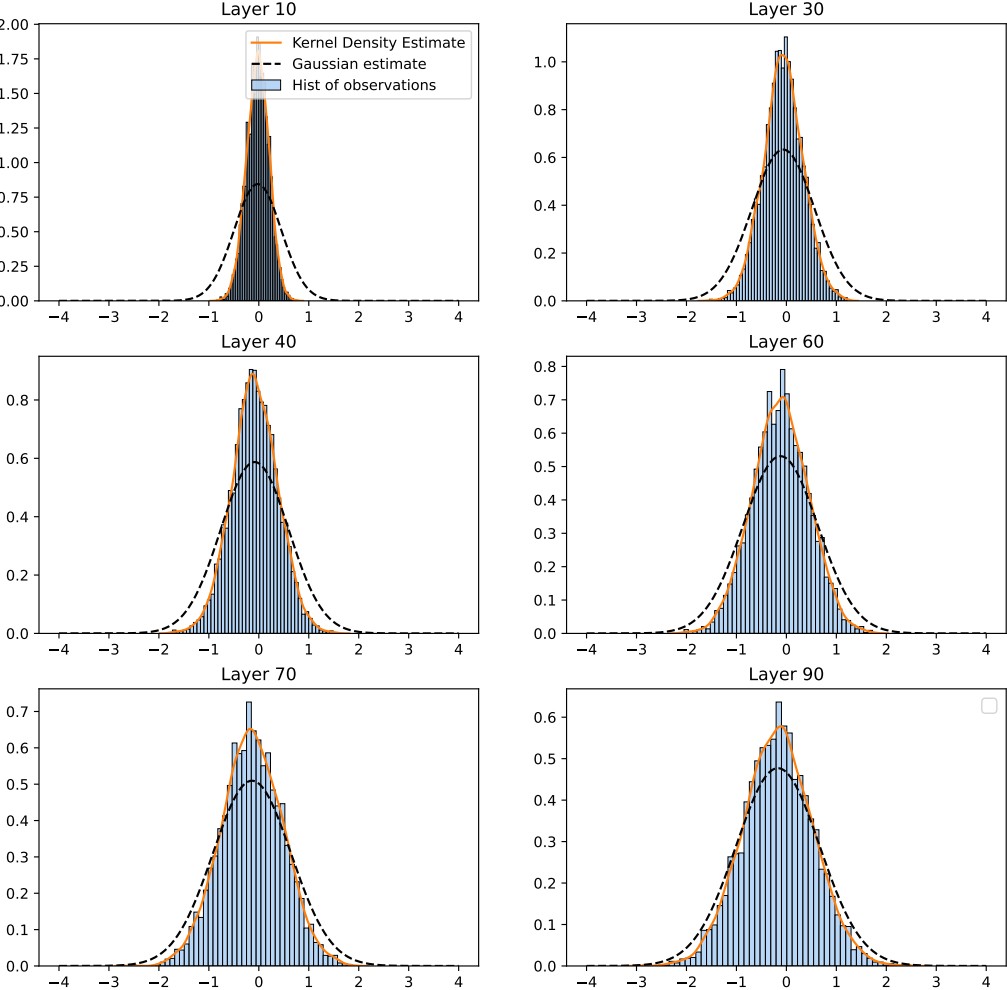

Figure 18: Distribution of $\log(\|\phi(Y_l)\|/\|\phi(Y_0)\|)$ (Eq. (1)) for different layer indices, with depth $L = 100$ and width $n = 2$

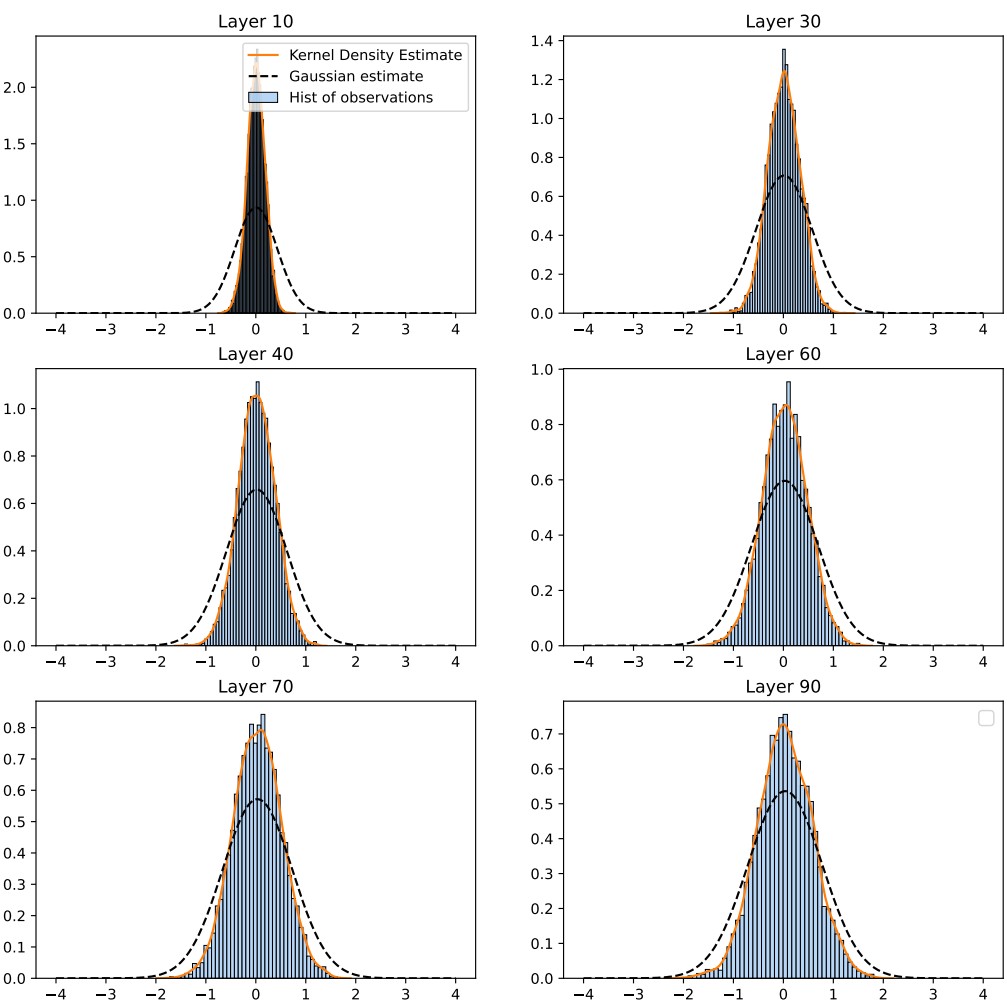

Figure 19: Distribution of $\log(\|\phi(Y_l)\|/\|\phi(Y_0)\|)$ (Eq. (1)) for different layer indices, with depth $L = 100$ and width $n = 3$

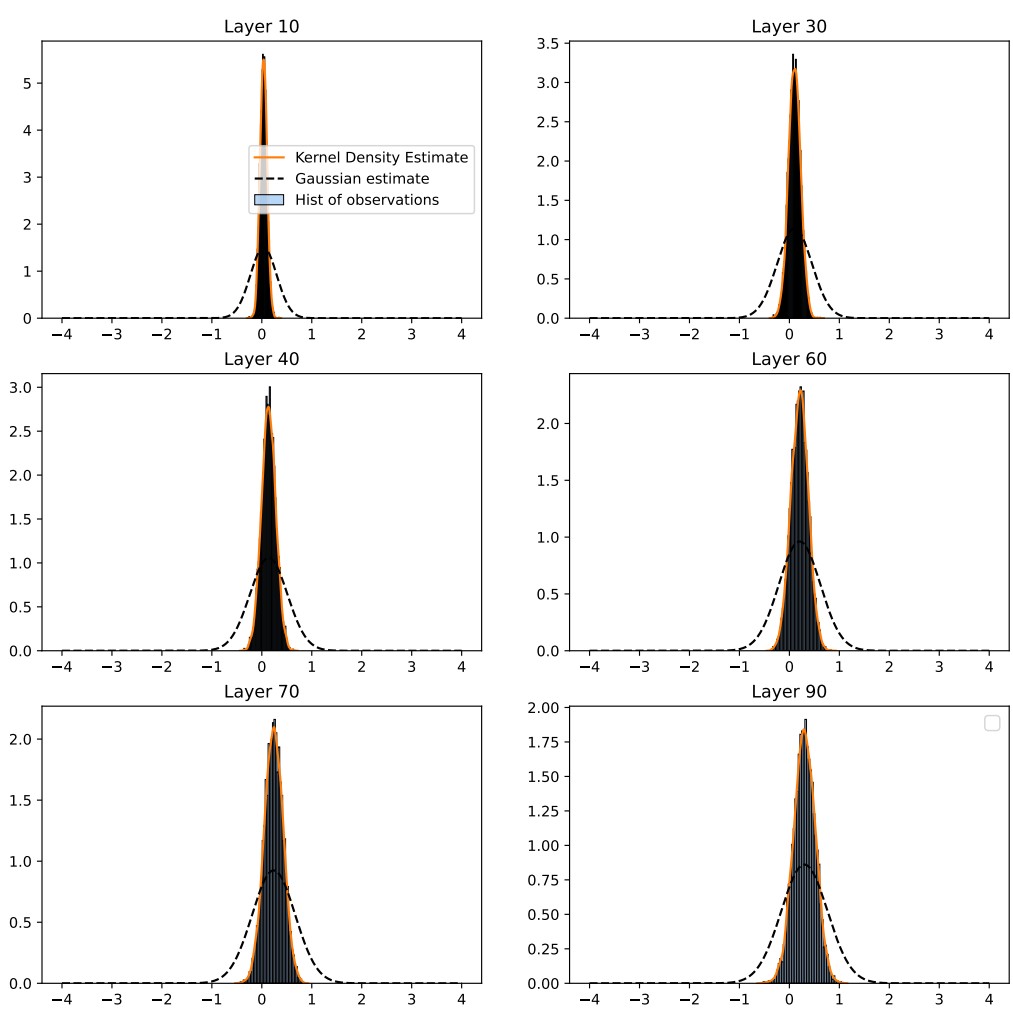

Figure 20: Distribution of $\log(\|\phi(Y_l)\|/\|\phi(Y_0)\|)$ (Eq. (1)) for different layer indices, with depth $L = 100$ and width $n = 20$

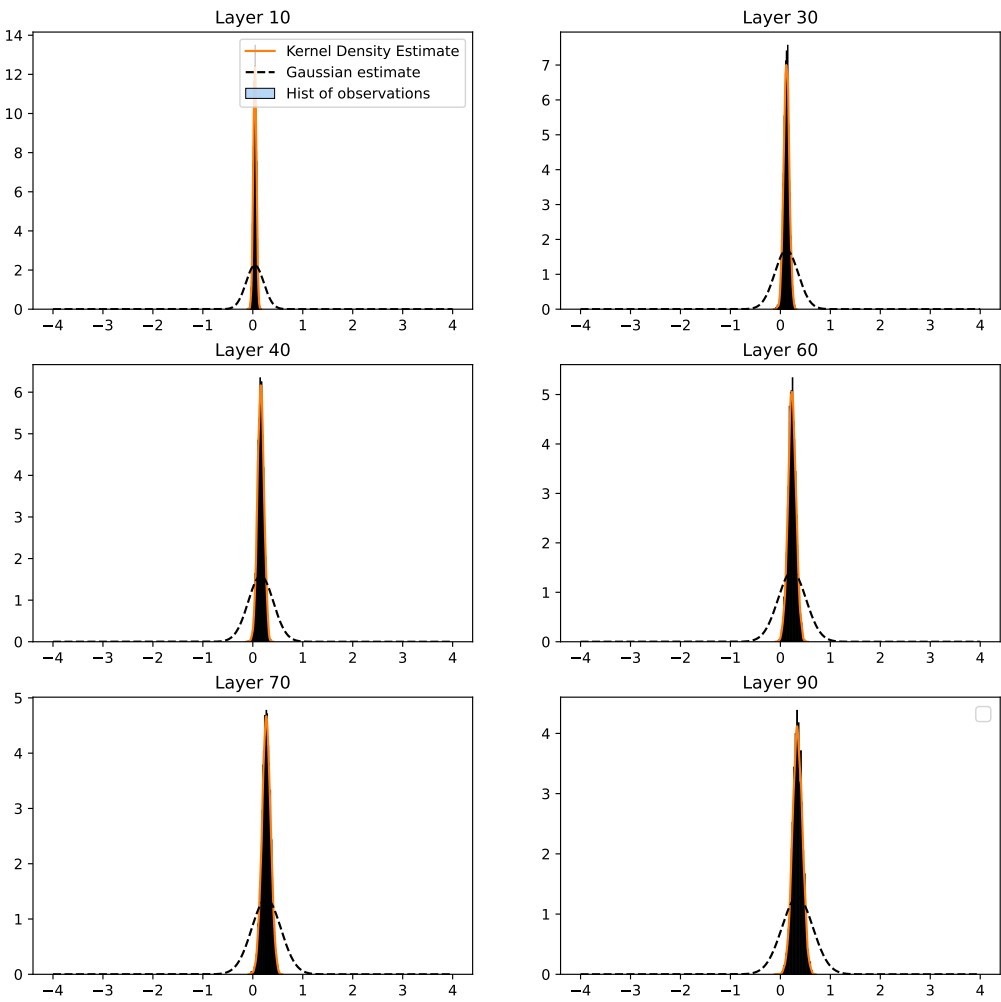

Figure 21: Distribution of $\log(\|\phi(Y_l)\|/\|\phi(Y_0)\|)$ (Eq. (1)) for different layer indices, with depth $L = 100$ and width $n = 100$.

