# OpenReview forum: "On the infinite-depth limit of finite-width neural networks"
_TMLR — Accepted by TMLR_

### Review · Reviewer_aoN3 · 2022-10-05

**Summary Of Contributions:**

The paper studied ResNets at initialization with a particular scaling of $1/\sqrt{L}$ weighting on the fully connected component, and study the limit as depth $L\to\infty$ for some fixed finite width $n$. The main contributions of the paper revolve around the limiting SDE in Proposition 1, and the consequences that follows from this SDE. In particular, the authors showed that the network does not collapse under some basic conditions, the norm of post activations behave approximately like a geometric Brownian motion, and characterized the sequent depth then width limit.

**Audience:**

Yes

**Claims And Evidence:**

Yes

**Requested Changes:**

On the infinite-depth-then-width limit

As the authors described in the introduction, the order of the limits matter for neural networks. In particular, the width-then-depth limit typically loses some information and therefore is not equal to the joint limit. I'm curious if the authors have any thoughts towards whether or not the depth-then-width limit behaves similarly and loses some information compared to the joint limit.

On the degeneracy of C-maps

One common issue of infinite-depth analysis is that the C-maps becomes degenerate in the limit, and this usually leads to poor training stability unless a normalization method or deep kernel shaping is used [1]. In the case of the scaling in this paper, do the authors have any thought on whether or not it leads to degenerate C-maps? As far as I can tell, the contribution of the activation function should be bounded in the limit.

On joint distribution over multiple inputs

Related to the earlier point, can the authors comment on any foreseeable technical challenges for extending towards the multiple input setting? In particular, can techniques from [2] be used towards deriving an SDE for the covariance kernel?

Minor typo in equation 4, I believe the drift is missing a factor of $1/2$.

References

1. Martens, James, Andy Ballard, Guillaume Desjardins, Grzegorz Swirszcz, Valentin Dalibard, Jascha Sohl-Dickstein, and Samuel S. Schoenholz. "Rapid training of deep neural networks without skip connections or normalization layers using deep kernel shaping." arXiv preprint arXiv:2110.01765 (2021).

2. Li, Mufan Bill, Mihai Nica, and Daniel M. Roy. "The Neural Covariance SDE: Shaped Infinite Depth-and-Width Networks at Initialization." arXiv preprint arXiv:2206.02768 (2022).

**Strengths And Weaknesses:**

Strengths
1. The infinite depth limit results are novel, as this regime is under explored.
2. The characterization of the limit is fairly detailed, despite a somewhat intractable limiting SDE.

Weaknesses
1. I would personally prefer some more discussion towards how this work relates to other regimes, and what the authors suspect happens the kernel for multiple inputs. More details in the requested changed section.

---

> ### Author Response · Authors · 2022-10-17
> **Authors' response**
>
> We thank the reviewer for their constructive feedback. We address their questions hereafter.
>
> 1. **Loss of information in the infinite-depth-then-width limit, degeneracy of the C-map and Q-map**
>
> In the infinite-width-then-depth, without proper scaling, the information is lost, meaning that the Q-map and C-map (generally) converge to constant values as depth grows. However, with proper scaling (the $1/\sqrt{L}$ in front of the resnet blocks), the limiting C-map (and Q-map) are not just *not degenerate*, but they are even *universal* as shown in [3], meaning that these kernels are rich enough to approximate any continuous function on a compact set. In the infinite-depth-then-width limit (our paper), we use the same scaling $1/\sqrt{L}$, and intuitively it should be expected that the correlation and covariance kernels do not converge to degenerate limits in this limit. We have added a discussion and a simulation of the exact dynamics of the C-map in section 5.
>
>
> 2. **On joint distribution over multiple inputs**
>
> The result of Proposition 1 can be easily generalized to the multiple input case. We still obtain an SDE in this case (we have added this result in the revised version (proposition 5 in the appendix)).
> However, unlike [2], the correlation and covariance kernels (C-map and Q-map) do not follow an SDE dynamics. This can be shown using ito's lemma: for two inputs $a \neq b$, the correlation $c_t(a, b) \overset{def}{=} \frac{\langle X_t(a), X_t(b) \rangle}{\|X_t(a)\| \|X_t(b)\|}$ has dynamics of the form
> $
>         d  c_t(a,b) = \Psi(X_t(a), X_t(b)) dB_t,
>        $
> for some non-trivial mapping $\Psi$. Unfortunately, this kind of dynamics (the volatility has non-trivial dependence on the process $c_t$) is generally intractable, and we are currently investigating this question for future work. So far, the main challenge is how to transform the dynamics of the correlation to obtain a tractable formula.
>
> 3. **Typo**:
> we would like to thank the reviewer for spotting this typo, which we have fixed in the revised version of the paper.
>
> *In light of your comments, we have added the case of multiple inputs in the revised version (Proposition 5 in the appendix), and added (and extended) the above discussion about the C-map (Section 5 in the main paper).*
>
>
> [1] Martens, James, Andy Ballard, Guillaume Desjardins, Grzegorz Swirszcz, Valentin Dalibard, Jascha Sohl-Dickstein, and Samuel S. Schoenholz. "Rapid training of deep neural networks without skip connections or normalization layers using deep kernel shaping." arXiv preprint arXiv:2110.01765 (2021).
>
> [2] Li, Mufan Bill, Mihai Nica, and Daniel M. Roy. "The Neural Covariance SDE: Shaped Infinite Depth-and-Width Networks at Initialization." arXiv preprint arXiv:2206.02768 (2022).
>
> [3] Soufiane Hayou, Eugenio clerico, Bobby He, et al. "Stable Resnet" (aistats 2021).

---

> > ### Comment · Reviewer_aoN3 · 2022-10-17
> > **Response**
> >
> > Thank you for the detailed response and quick updates to the draft. All of my requested changes have been addressed, and I would recommend this paper for acceptance.

---

### Review · Reviewer_iyP8 · 2022-10-14

**Summary Of Contributions:**

This paper studies random `residual` neural networks in the following limits: depth $L\to \infty$ and the width $n$ are either fixed and finite or go to infinity. Based on the observation that random `residual` neural networks can be viewed as an Euler discretion scheme of SDEs, the authors convert the studies of random residual networks to the studies of continuous time SDEs. As such, tools from stochastic calculus (e.g., Ito's formula) can be applied directly. The authors obtained several results, including
- Single input, $d=n=1$ ($d$: input dimension, $n$ width of the network) case. Characterization of the distribution of the output for some special activations.
-  Single input, $n>1$. The distribution is hard to get. Instead, the authors compute the distribution of the norm of the output, $\|X_t\|$.
- The authors also point out that the distributions of the norm $\|X_t\|$ are not the same under a different order of limits $d\to \infty$ then $n\to\infty$ v.s. $n\to\infty$ then $d\to\infty$.

Overall, the paper contains several interesting statistical insights, and the approaches of the paper seem rigorous. However, I find the results not very relevant, useful, or interesting from an ML perspective. It is unclear to me what this paper's main take-home `ML` message is. In my opinion, this paper may be more suitable for the stat community.

**Audience:**

No

**Broader Impact Concerns:**

NA.

**Claims And Evidence:**

Yes

**Requested Changes:**

I don't have requested changes for this paper. The major concern I have is the lack of relevance to the ML community. I would like to see more insights that could strengthen our theoretical understanding of neural networks.

**Strengths And Weaknesses:**

## Strengths
- Contains several interesting statistical observations.
- Simulations seem to support the authors' insights.

## Weaknesses
The biggest concern is the relevance of the paper to ML. The major connection I can tell is: residual networks can be viewed as an Euler discretion scheme of SDEs. Follow-up results derived from this observation provide limited insights into ML, e.g., insights into
- What is the distribution of the outputs (for multiple inputs), and how does this distribution affect the training and performance of networks? Note that the paper only handles single input.
- How is this limit ($L\to\infty$ first) related to training / initializing of neural networks? My understanding is that this scaling limit is not popular in practice and practitioners often scale the width and depth simultaneously. Moreover, the depth (in tens) << width (in thousands) in practice.
- What do the backward gradients (NTKs) look like, and what is their connection to network training?

I am not convinced that this paper is of great interest to the ML community without such or similar findings. I think the statistics avenue will be more suitable for this paper.

---

> ### Author Response · Authors · 2022-10-17
> **Authors' response (1/2)**
>
> We thank the reviewer for their comments. We believe there is a misunderstanding on some points. We summarize and explain below some key insights and how they are relevant to the ML community.
>
> ### **Relevance to training/initialization**
>
> 1.  **Stability in the large depth limit**: an important factor pertaining to the trainability of neural networks is the behaviour of the neurons. Ensuring that the neurons are well-behaved at initialization is crucial for training since the first step of any gradient-based training algorithm depends on the values of the neurons at initialization. From the existing theory on the infinite-width limit of neural networks, we know  that the $1/\sqrt{L}$ scaling of the residual blocks stabilizes the pre/post-activations (Hayou et al. (2021)) in the large depth limit. One can argue that this (approximately) ensures stability \emph{only} when the width is much larger than the depth ($n \gg L$). What about the other cases when $n \approx L$ and $n \ll L$? the last case can be studied by fixing the width $n$ and taking the depth $L$ to infinity. Wouldn't it be relevant to the ML community to know whether the pre/post-activations are stable in this regime? prior to our work, there was no answer to this question. In our paper, we not \emph{only} show that the neurons remain stable in fixed-width large-depth networks, but we fully characterize their behaviour when the depth is infinite and show that it follows an SDE in this limit.
>
>
> 2.  **Network collapse**: Another issue that could occur in finite-width networks is that of network collapse, i.e. when the pre-activations in a hidden layer are all negative, which causes the post-activations to be all zero. In resnet, this means that increasing depth beyond some level has no effect on the network output. This is problematic since the weights in those "inactive" layers have zero gradient and will not be updated when such event occurs. A simple way to understand such event is to see what happens at initialization. When the width is sufficiently large, one can expect that such event is unlikely to occur. What about small-width neural networks? we offer a simple answer to this question: for finite-width neural networks, increasing depth ensures that such event is unlikely to happen. This is true even for extremely small width, e.g. $n=2,3,$ which is counter-intuitive. Empirical results (Fig. 4) support this theoretical prediction.
>
>
> ### **Other insights relevant to ML community**
>
> 1. **No universal kernel regime**: an intersting application of fixed-depth infinite-width neural network is the so-called Neural Network Gaussian Process (NNGP). This is the Gaussian process limit of neural networks, that can also be used to perform posterior inference and obtain uncertainty estimates (see the introduction and Appendix A for references). The converse limit, i.e. fixed-width infinite-depth, has been however poorly understood, and the question of whether the infinite-depth limit of finite-width networks has some universal behaviour has been an open question since. We address this question in our work and show that the limiting object (in the case of the resnet architecture) does not obey a universal distribution (e.g. Gaussian process in the infinite-width limit). More precisely, this limit is highly sensitive to the activation function. We believe this is an important insight to share with the ML community, both theoreticians and practitioners.
>
> 2. **What about infinite-depth-then-width?**: the infinite-depth limit of infinite-width neural networks has been extensively studied in the literature (see introduction for references). It is known that in this limit, the network behaves as a Gaussian process with a well-defined kernel. What about the converse limit, i.e. infinite-width limit of infinite-depth networks? this has been so far an open question, and our work addresses one part of it. We show that marginal distributions are Gaussians (they follow a Mckean-vlasov process). Characterizing the full covariance kernel is still however an open question. We have added a section (section 5) to provide some insights on why this is a challenging question. We are currently investigating this topic for future work.

---

> ### Author Response · Authors · 2022-10-17
> **Authors' response (2/2)**
>
> ### **Other questions**
>
> 1. **Multiple inputs case, NTK**: our result can be easily generalized to the multiple inputs case. The vector of pre-activations follows an SDE dynamics with a specific volatility matrix. We have added the result in the revised version (Proposition 5 in the appendix). Regarding the NTK, as we mentioned in the conclusion, this is an open question that we are currently investigating.
>
>
> 2. **Should theory follow practice?**: as the reviewer suggests, in practice, the width $n$ is usually much larger than depth $L$, or they are of similar size $n \approx L$. Does this mean theory should only try to provide insights in these two cases? we respectfully disagree, and we believe that theoretical works should also investigate unexplored settings (e.g. finite-width, large-depth) which could yield to interesting practical findings in the future (It is often the case that a mathematical model (e.g. a neural network) is used in practice with a specific setting (e.g. architecture, width , depth, training algorithm etc.) because that is the \emph{empirically tested} setting where this tool yields good empirical results, and not because other settings do not work well).
>
> 3.  **Relevance to TMLR**: we respectfully disagree with the reviewer on this point. As per TMLR's guidelines, we believe our paper fits well in two categories: "experimental and/or theoretical studies yielding new insight into the design and behavior of learning in intelligent systems" and "development of new analytical frameworks that advance theoretical studies of practical learning methods;". Our paper introduces a new theoretical framework that provides new insights on finite-width large-depth neural networks. We submitted to TMLR because it emphasizes technical correctness over subjective significance. This is stated in the homepage of TMLR: "TMLR emphasizes technical correctness over subjective significance, to ensure that we facilitate scientific discourse on topics that are deemed less significant by contemporaries but may be important in the future".
>
>
> We hope this clarifies some misunderstandings and we are happy to address any further questions/comments.

---

### Review · Reviewer_2TWB · 2022-10-18

**Summary Of Contributions:**

The paper studies the distribution of the infinite-depth limit of activations of a fully connected resnet

The authors showed that the distribution is, in general, not tractable and identified a few special cases where the distribution can actually be identified

**Audience:**

Yes

**Broader Impact Concerns:**

Fine

**Claims And Evidence:**

Yes

**Requested Changes:**

1. There are two theoretical cases I would like to see, and I think would improve the comprehensibility of the theory
- The case when the activation is linear. It seems like the distribution is analytically tractable here
- The case when the activation is only perturbatively away from linearity; this would allow us to understand the leading effect of being nonlinear
- having these results, the authors should be able to make much more insightful discussions about the distribution -- for example, what part of the derived results can be attributed to having "depth" and what part can be attributed to being "nonlinear"

2. The authors have indeed demonstrated the correctness of the theory, but I think the authors should probably spend at least a little effort in demonstrating the possible usefulness of the theory, say, suggesting a new initialization trick
- this does not have to be a very large-scale experiment, just suggesting a minor trick that works on a toy example is satisfactory to me

3. I am against using the word "phase transition." The word phase transition comes from statistical physics and has a quite precise definition. To talk about a phase transition, one first needs to define/identify the free energy, and one then needs to identify where it becomes nonanalytic, but this is not what the authors have achieved.
- I would suggest using the word "regime change," "qualitative change in behavior," "regime crossover," or similar wording


After these changes, I would be happy to recommend acceptance


**Strengths And Weaknesses:**

Strength: the results are novel and have some fundamental importance in understanding deep learning, especially at initialization

Weakness:
1. the authors should probably spend at least a little effort in demonstrating the possible usefulness of the theory, say, suggesting a new initialization trick

2. there are a few other problems that I directly outline in the requested change section

---

> ### Author Response · Authors · 2022-10-26
> **Authors' response 1/2**
>
> We thank the reviewer for their constructive feedback. We have incorporated some suggested changes in the revised version of the paper. Details are provided below.
>
> ### I . Linear activation and perturbation analysis.
> Thank you for this interesting question.
> In section 4, we mentioned that the distribution of $X_t$ is generally intractable for $n \geq 2$. This is purely due to *finite width $n\geq 2$* and not the non-linearity of the activation function. The distribution of $X_t$ is intractable even for the identity activation function. In this case the process $X_t$ is solution of the following SDE
>
> $dX_t = \frac{1}{\sqrt{n}} \|X_t\| dB_t.$
>
> When $n=1$, this SDE has a closed-form solution given by the (conditional) GBM distribution (Prop 3). For general $n\geq 2$, the entries of $X_t$ are dependent and the resulting dynamics do not admit closed-form solutions. However, we can obtain closed-form solutions for the norm $\|X_t\|$. Indeed, a simple application of Ito's lemma yields
>
> $$
> \|X_t\| = \|X_0\| \exp\left(\frac{1}{\sqrt{n}} \hat{B}_t + \left(\frac{1}{2} - \frac{1}{n}\right) t\right),  \textrm{(**)}
> $$
>
> where $(\hat{B})_{t \geq 0}$ is a one-dimensional Brownian motion.
>
> **Role of the non-linearity.** By comparing the result of Thm1 and $(**)$, we observe the following: with ReLU, the drift term in $\log(\|\phi(X_t)\|/\|\phi(X_s)\|)$ is given by $\frac{1}{n}\int_{0}^t \mu_s ds$ which is a stochastic term with mean given by  $\left(\frac{1 - 2^{-n}}{4} - \frac{1}{n}\right) t$. With the identity activation, this drift term is \emph{deterministic} and is equal to $\left(\frac{1}{2} - \frac{1}{n}\right) t$. Hence:
>
> * *Non-linearity induces stochastic drift*: the non-linearity of ReLU induces stochasticity in the drift term of $\log(\|X_t\| / \|X_0\|)$, which results in the Quasi-GBM dynamics given by Thm1.
>
> * *Non-linearity induces change of regime*:  with ReLU, the mean drift of $\log(\|\phi(X_t)\| / \|\phi(X_0)\|)$ is given by $\left(\frac{1 - 2^{-n}}{4} - \frac{1}{n}\right) t$ which is negative for $n =1, 2, 3$. This induces the change of regime we discussed after Thm1 (having a negative mean drift implies that there is a significant mass of the distribution of $\|X_t\|/\|X_0\|$ in the regime $(0,1)$). With the identity activation function, the drift term is always non-negative for $n\geq 2$, and negative for $n=1$. Thus, the change of regime cover some values $n \geq 2$ only when there is a non-linearity. We give more details about this observation in the next result.
>
> To capture the effect of the non-linearity on the change-of-regime phenomenon discussed above, we study the dynamics of the post-norm activation for a special class of piece-wise linear activations of the form $\phi_{\alpha, \beta}(z) = \alpha \texttt{ReLU}(z) + \beta \texttt{ReLU}(z)$ (this includes both ReLU and the identity function). The result of Thm1 can be easily extended to the case of general piece-wise linear activation functions using the same proof techniques. We prove a generalized result (Thm10 in Appendix K). Due to page count limitations, we have added all these results (with the discussions) in Appendix K. Using Thm10, we can study the effect of small perturbations of the identity function on the distribution of $\|X_t\|$.
>
> **Perturbation analysis around the identity function**: Consider the case when $\alpha = 1$ and $\beta = 1 - \varepsilon$ for some $\varepsilon \ll 1$. The mean logarithmic growth factor between times $s$ and $t$ ($s < t$) is given by
> $$
> G_{s,t}^n := \left(\frac{1 + (1 - \varepsilon)^2}{4} - \frac{1}{n}\right) (t-s) \approx \left(\frac{1 - \varepsilon}{2} - \frac{1}{n}\right) (t-s).
> $$
> Observe that for $\varepsilon = 0$, we recover the result of the identity activation. Hence, a small perturbation of the identity function has the effect of decreasing the factor $G_{s,t}^n$ which results in having negative values for $G_{s,t}^n$ for certain values of $n$. Indeed, by fixing $\alpha = 1$, notice that the minimum values of $G_{s,t}^n$ is obtained when $\beta \approx 0$, for which $\phi_{1,0} = $ ReLU. Notice that we can also control the change of regime by tuning the parameter $\alpha$. This allows us to control the sign of $G_{s,t}^n$ for any $n$ by tuning the parameter $\alpha$. We leave the analysis of the practical implications of tuning $\alpha$ for future work.

---

> ### Author Response · Authors · 2022-10-26
> **Authors' response 2/2**
>
> ### II. Practical implications.
>
> * **Initialization scheme and stability in the large depth limit**: An important factor pertaining to the trainability of neural networks is the behaviour of the neurons (pre/post-activations). Ensuring that the neurons are well-behaved at initialization is crucial for training since the first step of any gradient-based training algorithm depends on the values of the neurons at initialization. The infinite-width limit of infinite-depth correlations.
> This has led to interesting developments in initialization schemes for MLPs such as the Edge of Chaos (Poole et al., 2016; Schoenholz et al., 2017) which ensures that the variance of the pre-activation does not (exponentially) vanish or explode in the large depth limit. In the case of ResNet, we know from the existing theory on the infinite-width limit of neural networks that scaling of the residual blocks stabilizes the pre/post-activations in the large depth limit (Hayou et al., 2021). Hence, we do not need a special initialization
> scheme in this case. However, one could argue that this (approximately) ensures stability only when the width is much larger than the depth. What about the other cases when $n \approx L$ or $n \ll L$? the last case can be studied by fixing the width and taking the depth to infinity. In our paper, we not only show that the neurons remain stable in fixed-width large-depth networks, but we fully characterize their behaviour when the depth is infinite and show that it
> follows an SDE in this limit. To summarize, we show that initializing ResNet (Eq. (1)) with standard Gaussian random variables and scaling the blocks with $1/\sqrt{L}$ ensures stability inside the network in the large-depth (fixed-width) networks (notice that this is actually equivalent to scaling the variance of the initialization weights with $\sqrt{1/L}$, this can be seen as an initialization scheme). Intuitively, by stabilizing the pre-activations, we also stabilize the gradients. We have added an empirical validation of this result in Fig8. The result shows that the $1/\sqrt{L}$ scaling, along with standard Gaussian initialization, ensure
> well-behaved gradients which is a desirable property for gradient-based training. Another interesting property of the Edge of Chaos initialization scheme for MLPs is that it ensures that the correlation kernel (correlation between the pre-activations for different inputs) does not converge to a degenerate value (constant value). We added a discussion of the correlation kernel in Section 5 and showed empirically
> that with the $1/\sqrt{L}$ scaling, the correlation is well-behaved and does not converge to degenerate values (Fig. 7). We have added a "practical implications" (section 6) to shed light on different practical insights.
>
>
> * **Other minor issues**: after carefully checking the literature on phase-transition, we agree with the reviewer that this wording is not suitable for the kind of phenomenon we are describing. We have changed this wording to "change of regime" or "regime change".
>
>
>
> **Summary of changes**:
>
> We have added the following results in the revised version of the paper:
> 1. Theoretical results for piece-wise linear activations (Appendix K) and a discussion on the effect of the non-linearity on the distribution of $\|X_t\|$
> 2. A section on the practical implication of our work (Section 5) where we highlight the main practical takeaways from our results. Notably, the stability of the scaling $1/\sqrt{L}$ for *finite-width* large depth networks (with empirical evaluations of the gradient norm), and a discussion on the network collapse phenomenon.
> 3. A section on the multiple inputs case, and an empirical evaluation of the correlation kernel showing that the latter does not converge to degenerate values.
>
> We hope these changes address your concerns.

---

> > ### Comment · Reviewer_2TWB · 2022-10-27
> > **thanks for the reply**
> >
> > I am satisfied with the updates, and would like to recommend acceptance

---

### Public Comment · ~Stefano_Peluchetti1 · 2023-01-16
**Clarifying the relationship with prior works**

> In the present paper, we study the infinite-depth limit of finite-width ResNet with random Gaussian weights (an architecture that is different from the one studied in [Peluchetti & Favaro, 2020])

Contrary to the claim, in [Peluchetti & Favaro, 2020] we studied an architecture that includes the one of the present work ([Hayou 2023]).

More in detail, we considered residual blocks of $d$ units of the form

$$
x_{t + Δt} = x_t + \boldsymbol{ϕ}(A_t \boldsymbol{ψ}(x_t) + a_t),\tag{a}
$$

where $\boldsymbol{ϕ}$ and $\boldsymbol{ψ}$ are two activation functions (we use here bold symbols to avoid conflicts with the notation of [Hayou 2023]), $Δt = 1/L$ is the time increment corresponding to $L$ layers, and $A_t$ and $a_t$ are respectively the weights and biases of layer $t$.

The results of [Peluchetti & Favaro, 2020] assume that $A_t$ and $a_t$ follow generic, appropriately scaled, Gaussian distributions (see Assumption 3.1, Theorem 3.1 and Corrollary 3.1).
The particular case of fully i.i.d. parameters (Assumption 3.4) corresponds to $A_t = ε^W_t\frac{σ_w}{\sqrt{L d}}$ and $a_t = ε^b_t\frac{σ_b}{\sqrt{L d}}$ for i.i.d. Gaussian distributions $ε^W_t, ε^b_t$ of appropriate dimensionality.

Setting $σ_w=1$, $σ_b=0$ and $\boldsymbol{ϕ}$ to the identity function, Equation (1) of [Hayou 2023] is recovered by (a).
Accordingly, Corollary 3.3 of [Peluchetti & Favaro, 2020] establishes the same limiting diffusion dynamics of Proposition 1 of [Hayou 2023].

> In the next result (which has been shown for the single input case in [Peluchetti & Favaro, 2020]), ...

The limiting diffusion dynamics for multiple inputs are also established in [Peluchetti & Favaro, 2020].
See Corollary 3.1 for the general case and Corollary 3.3 for the case of i.i.d. parameters.

Finally, it is worth noting that in our followup work ([Peluchetti & Favaro, 2021]) we studied the "Infinite-width limit of infinite-depth networks", which is the subject of Section 4.3 of [Hayou 2023], for the case where $\boldsymbol{ψ}$ of Equation (a) is the identity function.

We do not intend to undermine the accomplishments of the current paper, which presents a vast array of valuable results and advances our understanding of very deep neural networks.

However, we have found the representation of our prior work to be inaccurate and lacking, which prompted this brief comment.

References:

[[Peluchetti & Favaro, 2020]](https://proceedings.mlr.press/v108/peluchetti20a.html) *Infinitely deep neural networks as diffusion processes*

[[Peluchetti & Favaro, 2021]](https://jmlr.org/papers/v22/20-706.html) *Doubly infinite residual neural networks: a diffusion process approach*

---

> ### Author Response · Authors · 2023-01-16
> **Thank you**
>
> Dear Stefano,
>
> Thank your for your comments and for spotting the typo in the intro!
> I forgot to remove the "different architecture" from the intro. As the reader can understand, It is contradictory with the statement before Prop1 (it says clearly that this result was proven in your work). I will definitely update the ArXiv version to fix this typo.
>
> --> I would like to emphasize that the convergence to the SDE (prop1) is not new and is not by any means a contribution of this paper. The main contributions are related the impact of the activation function, and the distribution of post-activations.
>
> Regarding the infinite-width limit of infinite-depth networks, the architecture you are considering is different, and I do not see how it is related to this work.
>
>
> Best regards,
> The author

---

> > ### Public Comment · ~Stefano_Peluchetti1 · 2023-01-18
> > **Doubly infinite limit**
> >
> > > Regarding the infinite-width limit of infinite-depth networks, the architecture you are considering is different, and I do not see how it is related to this work.
> >
> > Section 5.3 of [Peluchetti & Favaro, 2021] considers Equation (a) without $\boldsymbol{ψ}$.
> >
> > Section 4.3 of [Hayou 2023] considers Equation (a) without $\boldsymbol{\phi}$.
> >
> > Both works study the same "infinite depth first, then infinite width" limit, under the same scaled Gaussian i.i.d. parametrization, obtaining the corresponding limiting diffusion dynamics.
> > The architecture is different (depending on where the activation function is placed), but saying that the results are unrelated seems a bit of stretch...

---

> > > ### Author Response · Authors · 2023-01-20
> > > **Response**
> > >
> > > The only intersection between the architecture you're considering and the one I'm considering is when $\phi = \psi = $ identity which is a trivial case. I will have to read carefully your paper, and will cite it in the related works if I find an interesting connection.

---

### Decision · Action_Editors · 2022-12-16

**Recommendation:** Accept as is

**Comment:**

The paper studies the infinite-depth limit of finite-width residual neural networks. The paper discusses several interesting results including characterization of the output distribution for some special cases. The reviewers noted several positive aspects of the paper including:
- The novelty of the infinite depth limit as this regime is relatively under-explored
- The simulations support the theoretical insights.
- Detailed characterization of special cases
- Comparisons of infinite-depth-then-infinite-width with the more popular infinite-width-then-infinite-depth limit.

The authors have also satisfactorily addressed suggested changes proposed by reviewers including:
- linear activation and perturbation analysis
- discussion of practical implications
- discussion of joint distribution over multiple inputs

During the discussion, the consensus decision of the reviewers leaned towards acceptance. There was some discussion around relevance to TMLR and if the paper would be a better fit for stats journal, but I think that this paper is very relevant to TMLR audience (see my justification under "Audience" above).

I recommend accept as is. Congrats!

**Audience:**

The paper is relevant to TMLR's audience as it fits under at least two categories in https://jmlr.org/tmlr/editorial-policies.html#evaluation
- "experimental and/or theoretical studies yielding new insight into the design and behavior of learning in intelligent systems" and
- "development of new analytical frameworks that advance theoretical studies of practical learning methods"

**Claims And Evidence:**

Yes, the claims are convincing and well supported by evidence. All reviewers agree that claims and evidence are well supported "The paper is relevant to theorists of the field, and the results are solid."